# A Bayesian Nonparametric Framework for Private, Fair, and Balanced Tabular Data Synthesis

**Forough Fazeliasl, Linglong Kong & Bei Jiang**
Department of Mathematical
and Statistical Sciences
University of Alberta, AB, Canada
`{fazelias,lkong,bei1}@ualberta.ca`

**Michael Minyi Zhang**
Department of Statistics
and Actuarial Science
University of Hong Kong
Pok Fu Lam, Hong Kong
`mzhang18@hku.hk`

## Abstract

A fundamental challenge in data synthesis is protecting the fairness and privacy of the individual, particularly in data-scarce environments where underrepresented groups are at risk of further marginalization by reproducing the biases inherent in the data modeling process. We introduce a privacy- and fairness-aware generative model, which fuses the conditional generator within the framework of Bayesian nonparametric learning (BNPL). This conditional structure imposes fairness constraints in our generative model by minimizing the mutual information between generated outcomes and protected attributes. Unlike existing methods that primarily focus on sensitive binary-valued attributes, our framework extends seamlessly to non-binary attributes. Moreover, our method provides a systematic solution to class imbalance, ensuring adequate representation of underrepresented protected groups. Our proposed approach offers a scalable, privacy-preserving framework for ethical and equitable data generation, which we demonstrate by theoretical guarantees and extensive experiments on sensitive empirical examples.

## 1 Introduction

Tabular data, which are structured collections of continuous and categorical variables, are critical for building multimodal artificial intelligence (AI) systems such as generative image and language models that integrate structured data with text and images. Synthesizing tabular data is particularly challenging when fairness (equal treatment across protected groups), privacy (individual-level protection, typically through differential privacy), and class balance (uniform representation across categorical attributes) constraints must be jointly ensured while preserving utility. Violations in any of these aspects can reinforce bias, leak sensitive information, or distort learning through imbalance (Shaham et al., 2025).

Although several works separately address these concerns (Chen et al., 2024), unifying them remains an open challenge. Among existing methods, most rely solely on generative adversarial networks (GANs) to model the data, which risk mode collapse and can thereby exacerbate class-diversity issues. Others rely on variational autoencoders (VAEs), which can better promote diversity but often produce outputs that miss fine-grained patterns. Combining VAEs and GANs could jointly leverage the diversity and fidelity of each method for generating data, *yet its application remains underexplored for tabular data*.

To unify fairness and privacy within a single framework for tabular data generation, we propose a novel supervised Bayesian nonparametric method conditioned on protected attributes, which is flexible enough to be applied to any model in the class of generator–decoder models (GDMs). We demonstrate its implementation by applying the proposed scheme to generative model which combines VAEs and conditional GANs.

Our model enforces fairness by using a BNP-based mutual information (MI) regularization term, which offers a computationally efficient procedure for training the generative model. The MI term

facilitates fairness by conditioning on protected attributes and further allows direct control over group proportions to ensure balanced representation of minority classes. In particular, a notable advantage is our ability to support fair data augmentation for non-binary valued protected attributes.

We enforce privacy protection at two levels using BNPL: (1.) Globally, by using a Dirichlet process (DirP) mechanism, which injects structured uncertainty into the sampling process. (2.) Locally, by constructing a copula-based base measure for the DirP to ensure that continuous and categorical features receive tailored privacy protections at the marginal level while preserving feature dependencies. The limitations of standard training methods in privacy preservation make BNPL a promising alternative, as we will discuss in later sections.

Our paper is structured as follows: In Section 2, we briefly provide an overview of fair and private generative networks. In Section 3, we define fairness and privacy in the context of this paper. Later, we present the technical details for our proposed idea in Section 4. Then, we provide experimental results in Section 5 where we will validate our proposed model on real datasets which exhibit sensitive features and demand privacy protection. Finally, we conclude our paper in Section 6 and provide directions for future work. A notational glossary, a definition of terms, proofs, additional experiments, and algorithms are provided in the appendix.

## 2 BACKGROUND WORK

We position our work in the intersection of privacy preservation and fairness-aware data synthesis. While others have focused on, at most, one aspect of this problem, we seek to fill a critical gap in tabular generative models to address both of these issues. Previous work in synthetic generation of tabular data is primarily based on generative adversarial networks (Goodfellow et al., 2014). One notable example is the CTGAN, which aims to preserve the modes of the population to ensure high utility by conditioning on one discrete attribute per step (Xu et al., 2019). However, CTGAN lacks the ability to set category ratios arbitrarily at generation which hinders our ability to fairly balance classes. Moreover, GANs in general do not inherently have a mechanism to protect privacy.

In privacy-preserving tabular data synthesis (Truda, 2023, for example), most approaches build on either the differentially private stochastic gradient descent (Song et al., 2013), which can exacerbate class imbalance by disproportionately affecting underrepresented groups. Or private aggregation of teacher ensembles methods (Papernot et al., 2017), which faces scalability issues due to its reliance on multiple teacher discriminators. VAE-based methods, like OVAE, introduce a framework with differentiable oblivious decision trees to embed inductive bias (Vardhan & Kok, 2020), but also lacks formal privacy guarantees and does not support a tunable privacy budget.

Fairness-aware models like TabFairGAN, DECAF, and FairGAN, for example (Rajabi & Garibay, 2022; Van Breugel et al., 2021; Xu et al., 2018), impose fairness constraints through modifications of the GAN architecture. However, these models struggle to handle non-binary protected attributes during training. Currently, there exists some related works which use MI for classifier-based fairness (Cho et al., 2020; Roh et al., 2020), though these models ignore the marginal entropy of the protected attributes, which reduces its capacity to represent the data.

Privacy and fairness interact in ways that make the joint problem more challenging than enforcing either one alone. Fairness relies on accurate group-level statistics, yet privacy noise can distort these estimates, especially for small sensitive groups. At the same time, fairness constraints further restrict an optimization process already affected by noise. Consequently, the two constraints can reinforce each other's difficulty. Existing approaches at this intersection include PF-WGAN and PreFair for differentially private fair synthetic data generation (Sarmin et al., 2025; Pujol et al., 2023).

Previous methods to remedy unbalanced classes rely on techniques like oversampling of the minority class (Chawla et al., 2002), or generating synthetic examples through linear interpolations and generative networks (He et al., 2008; Chen et al., 2024). However, many of these methods are rather simplistic in their design to handle unbalanced data or are complex to train.

## 3 FAIRNESS AND PRIVACY

Consider a dataset $\mathcal{D} = (\mathbf{X}, Y, S) \in \mathbb{R}^d$, where $\mathbf{X}$ contains unprotected attributes (both continuous and categorical), $S \in \{0, 1\}$ is a binary protected attribute, and $Y \in \{0, 1\}$ is a binary outcome variable. A positive outcome is denoted by $Y = 1$, and $S = 1$ indicates membership in a protected group (e.g., gender or ethnicity). In this paper, we focus on the following concepts of fairness and privacy:

**Fairness via Statistical Parity (SP):** SP requires the decision $Y$ to be independent of the protected attribute $S$, i.e., the probability of a positive outcome must be the same across groups:

$$\Pr(Y = 1 \mid S = 0) = \Pr(Y = 1 \mid S = 1). \tag{1}$$

**Equivalence via Mutual Information:** While SP extends to categorical $S \in [\![K]\!]$ with $K > 2$, it becomes complex in many generative models due to pairwise comparisons. Instead, SP can be formulated via MI, holding if and only if

$$\mathrm{MI}(Y, S) = \mathrm{D}_{\mathrm{KL}}(F_{Y,S}, F_Y \otimes F_S) = H(Y) - H(Y \mid S) = 0, \tag{2}$$

with $\mathrm{D}_{\mathrm{KL}}$ denoting the KL divergence, $\otimes$ the product of marginals, and $H(\cdot)$, $H(\cdot \mid \cdot)$ the Shannon and conditional entropy, respectively.

$(\epsilon, \delta)$**-Differential Privacy (DP) Mechanism:** DP ensures that the inclusion or exclusion of a single data point does not significantly alter the output distribution of a randomized algorithm. A mechanism $\mathcal{M} : \mathfrak{X} \to \mathbb{R}^{m'}$ satisfies $(\epsilon, \delta)$-differential privacy if for any neighboring datasets $\mathbf{X}, \mathbf{X}' \in \mathbb{R}^m$ and measurable set $E \subseteq \mathbb{R}^{m'}$,

$$\Pr[\mathcal{M}(\mathbf{X}; \epsilon, \delta) \in E] \le e^\epsilon \Pr[\mathcal{M}(\mathbf{X}'; \epsilon, \delta) \in E] + \delta$$

where $\epsilon$ controls the privacy-utility trade-off, and $\delta$ allows a small failure probability. If $\mathcal{M}$ satisfies $(\epsilon, 0)$-differential privacy, it is referred to as $\epsilon$-differential privacy.

**Classical Gaussian Mechanism (CGM):** A mechanism that satisfies $(\epsilon, \delta)$-DP defines the CGM as $\mathcal{M}_{\mathrm{CGM}}(\mathbf{X}; \epsilon, \delta) = s(\mathbf{X}) + \mathbf{Z}$ where $s : \mathfrak{X} \to \mathbb{R}^{m'}$ is a deterministic function, and $\mathbf{Z} \sim \mathcal{N}(\mathbf{0}, \sigma^2 \boldsymbol{I}_{m'})$. The noise variance is calibrated using a tail bound approximation, leading to the variance $\sigma^2 = \frac{2\Delta_s^2 \ln(1.25/\delta)}{\epsilon^2}$, which tends to overestimate the noise, especially in high and low privacy regimes. Here, $\Delta_s = \sup_{\mathbf{X}, \mathbf{X}'} \|s(\mathbf{X}) - s(\mathbf{X}')\|$ is the global sensitivity of $s$ measuring the maximum change in the output of $s$ between neighboring inputs.

**Analytic Gaussian Mechanism (AGM):** The AGM improves the CGM by tightening the noise variance calibration for continuous datasets (Balle & Wang, 2018). The AGM, denoted by $\mathcal{M}_{\mathrm{AGM}}(\cdot; \epsilon, \delta)$, improves the CGM by computing the noise variance $\sigma$ that satisfies the exact privacy condition: $\Phi\left(\frac{\Delta_s}{2\sigma} - \frac{\epsilon\sigma}{\Delta_s}\right) - e^\epsilon \Phi\left(-\frac{\Delta_s}{2\sigma} + \frac{\epsilon\sigma}{\Delta_s}\right) \le \delta$, where $\Phi$ is the standard normal cumulative distribution function (CDF). AGM reduces variance, improving utility while maintaining privacy.

**Randomized Response Mechanism (RRM):** Randomized response is a privacy-preserving mechanism used to privatize categorical data (Wang et al., 2016). Let $X$ be a categorical random variable taking values from a discrete set $[\![K]\!]$, where $K$ is the number of categories. The RRM, denoted by $\mathcal{M}_{\mathrm{RRM}}(X; \epsilon)$, perturbs the original value $X$ according to a privacy budget $\epsilon$, which controls the trade-off between privacy and accuracy. The $\epsilon$-differential privacy mechanism is given as

$$\Pr(\mathcal{M}_{\mathrm{RRM}}(X; \epsilon) = x) = \begin{cases} \frac{e^\epsilon}{e^\epsilon + K - 1} & \text{if } x \text{ is the true value of } X, \\ \frac{1}{e^\epsilon + K - 1} & \text{otherwise.} \end{cases}$$

## 4 PRIVACY AND FAIRNESS PRESERVATION WITH BAYESIAN NONPARAMETRIC LEARNING

Our proposed generative model uses BNPL (Fong et al., 2019) as a method of ensuring privacy and fairness protection by resampling the data from a Dirichlet process (DirP) posterior, which we will first introduce in this section.

## 4.1 THE DIRICHLET PROCESS

The Dirichlet process, denoted by $F \sim \mathrm{DP}(a, H)$, is a distribution over probability measures on a measurable space $(\mathfrak{X}, \mathcal{A})$, where for any measurable partition $A_1, \ldots, A_k$, the vector $(F(A_1), \ldots, F(A_k)) \sim \mathrm{Dir}(aH(A_1), \ldots, aH(A_k))$ (Ferguson, 1973). The base measure $H$ encodes prior knowledge, and $a > 0$ controls its strength. As a conjugate prior, the posterior is again a DirP: $F^{\mathrm{Pos}} \sim \mathrm{DP}(a + n, H^*)$, where $H^* = \frac{a}{a+n} H + \frac{n}{a+n} F_{\mathcal{D}_{1:n}}$ and $F_{\mathcal{D}_{1:n}}$ is the empirical CDF of the sample, $\mathcal{D}_{1:n} = (\mathbf{X}_{1:n}, \mathbf{Y}_{1:n}, \mathbf{S}_{1:n})$. To facilitate sampling, we use the finite approximation of the posterior DirP from Ishwaran & Zarepour (2002), avoiding the need to truncate the infinite stick-breaking process (Sethuraman, 1994; Zarepour & Al-Labadi, 2012):

$$F_{\mathcal{D}_{1:N}^{\mathrm{Pos}}}^{\mathrm{Pos}}(\cdot) := \sum_{i=1}^N J_{i,N-1}^{(a+n)} \mathbb{I}_{\mathcal{D}_i^{\mathrm{Pos}}}(\cdot), \tag{3}$$

where $(J_{1,N-1}^{(a+n)}, \ldots, J_{N,N-1}^{(a+n)}) \sim \mathrm{Dir}((a+n)/N, \ldots, (a+n)/N)$, and $\mathcal{D}_i^{\mathrm{Pos}} \overset{\mathrm{IID}}{\sim} H^*$. Here, $J_{i,N-1}^{(a+n)}$ and $\mathcal{D}_i^{\mathrm{Pos}}$ represent weights and locations, respectively. In the subsequent sections, we investigate the efficacy of this approximation within a regularization method in a BNP generative model. To generate $(J_{i,N-1}^{(a+n)})_{1 \leq i \leq N}$, we first sample $\Gamma_{i,N} \overset{\mathrm{IID}}{\sim} \mathrm{Gamma}((a+n)/N, 1)$ independently from $\mathcal{D}_i^{\mathrm{Pos}}$ and then set $J_{i,N-1}^{(a+n)} = \Gamma_{i,N} / \sum_{i=1}^N \Gamma_{i,N}$.

**Remark 1** *To balance accuracy and efficiency in approximating Eq. 3, we adopt the adaptive truncation rule from Zarepour & Al-Labadi (2012), stopping at $N = \inf \left\{ j : \frac{\Gamma_{j,j}}{\sum_{i=1}^j \Gamma_{i,j}} < \rho \right\}$ for threshold $\rho \in (0, 1)$, when the incremental weight becomes negligible.*

## 4.2 THE DIRICHLET PROCESS SATISFIES $(\epsilon, \delta)$-DIFFERENTIAL PRIVACY

The next proposition shows how the randomized weights in Eq. 3 contribute to the privatization of the distribution; see Appendix G for further clarification. The top panel of Fig. 1 illustrates its effect in perturbing the shape of the empirical distribution around the base measure.

**Proposition 1 (Global Privacy Guarantee)** *For a fixed $\varepsilon \in (0, 1]$ and parameters $\eta, \bar{\eta} \in (0, 1)$, let $W \subseteq [\![N-1]\!]$ and define $b = \min\{1, a^{-2}\}$. Consider two $b$-adjacent empirical distributions of the resampled dataset $\mathcal{D}_{1:N}$, given by $F_{\mathcal{D}_{1:N}} = \sum_{i=1}^N p_i \mathbb{I}_{\mathcal{D}_i}$ and $F'_{\mathcal{D}_{1:N}} = \sum_{i=1}^N p'_i \mathbb{I}_{\mathcal{D}_i}$, where $p_{1:N}, p'_{1:N} \in \Delta_{\eta, \bar{\eta}}$, $N \geq 2/b$, and there exist indices $i, j \in W$ such that $p_{-(i,j)} = p'_{-(i,j)} = 1/N$, with $p'_i = (1+\varepsilon)/N$, $p'_j = (1-\varepsilon)/N$, and $\|p_{(i,j)} - p'_{(i,j)}\|_1 \leq b$. Then, given regularity assumptions (Eq. 3-Eq. 5), the DirP weights, $\mathcal{M}_{\mathrm{DirP}}^{(a+n)}(p_{1:N}; \epsilon_{glo}, \delta_{glo}) := \left( J_{1,p_1}^{(a+n)}, \ldots, J_{N,p_N}^{(a+n)} \right)$, define a Dirichlet mechanism with parameter $a + n$, meaning that for all measurable sets $E \subset \Delta$, it satisfies $(\epsilon_{glo}, \delta_{glo})$-differential privacy:*

$$\Pr[\mathcal{M}_{\mathrm{DirP}}^{(a+n)}(p_{1:N}; \epsilon_{glo}, \delta_{glo}) \in E] \leq e^{\epsilon_{glo}} \Pr[\mathcal{M}_{\mathrm{DirP}}^{(a+n)}(p'_{1:N}; \epsilon_{glo}, \delta_{glo}) \in E] + \delta_{glo}.$$

*The privacy parameters are given by*

$$\epsilon_{glo} = \ln \left( \frac{B\left((a+n)\eta, (a+n)(1-\bar{\eta}-\eta)\right)}{B\left((a+n)(\eta + \frac{b}{2}), (a+n)(1-\bar{\eta}-\eta-\frac{b}{2})\right)} \right) + \frac{b(a+n)}{2} \ln \left( \frac{1 - (|W|-1)\gamma}{\gamma} \right),$$

*and*

$$\delta_{glo} = 1 - \min_{p_{1:N}, p'_{1:N}} \Pr \left[ \left( J_{1,N-1}^{(a+n)}, \ldots, J_{N,N-1}^{(a+n)} \right) \in \Omega_\gamma \right],$$

*where $\Omega_\gamma = \{\boldsymbol{x} \in \Delta \mid x_i \geq \gamma, \forall i \in W\}$, $\Delta = \{\boldsymbol{x} \in \mathbb{R}^N \mid \sum_{i=1}^N x_i = 1, x_i \geq 0, \forall i \in [\![N]\!]\}$, $\Delta_{\eta, \bar{\eta}} = \{p_{1:N} \mid \sum_{i \in W} p_i \leq 1 - \bar{\eta}, p_i \geq \eta, \forall i \in W\}$, and $B(\cdot, \cdot)$ is the beta function.*

**Corollary 1 (Global Perfect Privacy)** *Under the conditions of Proposition 1, as $a \to \infty$, we have (i) $\epsilon_{glo} \to 0$; moreover, (ii) $\delta_{glo} \overset{p}{\to} 0$ for fixed $|W| = N - 1$.*

## 4.3 Localized Privacy via Copula-Based Base Measure

Proposition 1 inherently implies a default level of privacy that is always obtained from the weights of the DirP approximation in Eq. 3. However, the locations in the DirP approximation are not inherently protected by this mechanism. To enhance privacy while incorporating regularization, we model the base measure of DirP prior as a copula-based measure. The copula applies a suitable privacy mechanism separately to each continuous and categorical column of the tabular dataset while preserving the dependency structure among variables.

Specifically, let the tabular dataset $\mathcal{D}_{1:n}$ have $N_C$ continuous columns and $N_D$ discrete columns, with $d = N_C + N_D$. Let $\mathcal{D}_{1:n,i_C}^{(C)}$ and $\mathcal{D}_{1:n,i_D}^{(D)}$ be the $i_C$th continuous and $i_D$th discrete columns with $n$ samples, where $i_C \in [\![N_C]\!]$ and $i_D \in [\![N_D]\!]$. Consider the representation of the dataset as

$$\mathcal{D}_{1:n} = \Big( \overbrace{\underbrace{\mathcal{D}_{1:n,1}^{(C)}, \cdots, \mathcal{D}_{1:n,N_C}^{(C)}}_{}, \overbrace{\underbrace{\mathcal{D}_{1:n,1}^{(D)}, \cdots, \mathcal{D}_{1:n,N_D-2}^{(D)}}_{\text{Unprotected attributes: } \mathbf{X}_{1:n}}, \underbrace{\mathcal{D}_{1:n,N_D-1}^{(D)}}_{\text{Outcomes: } \mathbf{Y}_{1:n}}, \underbrace{\mathcal{D}_{1:n,N_D}^{(D)}}_{\text{Protected attributes: } \mathbf{S}_{1:n}}}^{\mathcal{D}_{1:n,1:N_D}^{(D)} \in \mathbb{Z}_+^{n \times N_D}} \Big). \quad (4)$$

where the continuous block is $\mathcal{D}_{1:n,1:N_C}^{(C)} \in \mathbb{R}^{n \times N_C}$.

For each $i_C$, consider the sample mean and variance given by $\hat{\mu}_{i_C}^{(C)} = n^{-1} \sum_{i=1}^n \mathcal{D}_{i,i_C}^{(C)}$ and $\hat{\sigma}_{i_C}^{2(C)} = n^{-1} \sum_{i=1}^n \left( \mathcal{D}_{i,i_C}^{(C)} - \hat{\mu}_{i_C}^{(C)} \right)^2$, respectively. Likewise, consider the estimated probability of observing category $j$ in $\mathcal{D}_{1:n,i_D}^{(D)}$ as $\hat{p}_{i_D j} = \frac{n_{i_D j}}{n}$, where $n_{i_D j}$ is the number of samples in category $j$ of $\mathcal{D}_{1:n,i_D}^{(D)}$, $j \in [\![K_{i_D}]\!]$, with $K_{i_D}$ being the number of categories in $\mathcal{D}_{1:n,i_D}^{(D)}$. The estimated probability vector is $\hat{\boldsymbol{p}}_{i_D} = (\hat{p}_{i_D 1}, \ldots, \hat{p}_{i_D K_{i_D}})$ with $\sum_{j=1}^{K_{i_D}} \hat{p}_{i_D j} = 1$. We then construct the base measure of DirP prior through the following steps:

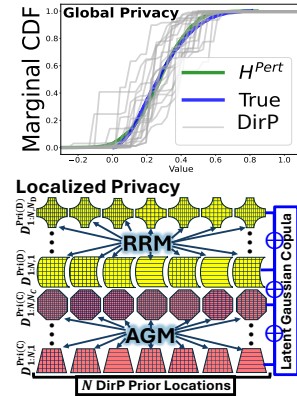

Figure 1: Privacy Scheme.

**Step 1. Regularizing marginals:** We first assign a normal distribution $H_{i_C}^{(C)} := N(\hat{\mu}_{i_C}^{(C)}, \hat{\sigma}_{i_C}^{2(C)})$ and a categorical distribution $H_{i_D}^{(D)} := \text{Cat}(K_{i_D}, \hat{\boldsymbol{p}}_{i_D})$ as marginal priors to each $F_{i_C}^{(C)}$ and $F_{i_D}^{(D)}$, respectively, where $F_{i_C}^{(C)}$ and $F_{i_D}^{(D)}$ represent the true distributions of the $i_C$th continuous and $i_D$th discrete columns in Eq. 4. These marginal priors will be used to construct the joint base measure of DirP prior. This step anchors the sampled distribution around the marginal prior and acts a regularizer on the marginals, thereby robustifying model training.

Furthermore, it prevents *prior data conflict* (Evans & Moshonov, 2006; Mutsvari et al., 2016)—when the prior contradicts the data—by employing the *empirical Bayes* framework (Maritz, 2018), where the prior's hyperparameters are estimated from the data and treated separately in inference to stabilize the model. In the BNP framework, empirical Bayes estimates the base measure of the DirP, which serves as its hyperparameter, using data (McAuliffe et al., 2006).

**Step 2. Perturbing marginals:** For given $\epsilon_{\text{loc}_{i_C}}^{(C)}, \delta_{\text{loc}_{i_C}}^{(C)}, \epsilon_{\text{loc}_{i_D}}^{(D)}$, let $\mathcal{D}_{i_C}^{\text{Pri(C)}} \sim H_{i_C}^{(C)}$ and $\mathcal{D}_{i_D}^{\text{Pri(D)}} \sim H_{i_D}^{(D)}$. Using the identity function for $s(\cdot)$ in the AGM, we perturb each prior marginal by $\mathcal{M}_{\text{AGM}}(\mathcal{D}_{i_C}^{\text{Pri(C)}}; \epsilon_{\text{loc}_{i_C}}^{(C)}, \delta_{\text{loc}_{i_C}}^{(C)}) \sim F_{\text{AGM},i_C}$ and $\mathcal{M}_{\text{RRM}}(\mathcal{D}_{i_D}^{\text{Pri(D)}}; \epsilon_{\text{loc}_{i_D}}^{(D)}) \sim F_{\text{RRM},i_D}$, respectively.

**Step 3. Capturing dependencies between marginals:** Then, we estimate the correlation between continuous features using Pearson correlation, denoted by $\hat{R}_{CC}$. For dependencies involving discrete features, use Spearman rank correlation for both discrete-discrete and discrete-continuous pairs, resulting in the total estimated correlation matrix $\hat{R} = [\hat{R}_{CC}, \hat{R}_{CD}; \hat{R}_{DC}, \hat{R}_{DD}]$. This estimation process follows the procedure described in Mendes et al. (2023) for tabular dataset. Given $\mathbf{t} \in \overline{\mathbb{R}}^d$, apply $\hat{R}, F_{\text{AGM},i_C}, F_{\text{AGM},i_D}$ in Eq. 10 (Appendix D) to model the joint perturbed base measure using copula-based model Eq. 5.

$$H^{\text{Pert}}(\mathbf{t}) = C_{\hat{R}}(F_{\text{AGM},1}(t_1), \ldots, F_{\text{AGM},N_C}(t_{N_C}), F_{\text{RRM},1}(t_{N_C+1}), \ldots, F_{\text{RRM},N_D}(t_d)). \quad (5)$$

The final two steps ensure that both continuous and categorical attributes receive tailored privacy treatment through the prior marginals while capturing the covariance structure between the variables.

This framework enables *localized privacy* by allowing marginal privatization only on selected continuous and discrete columns indexed by $I^{(\mathrm{C})} \subseteq [\![N_{\mathrm{C}}]\!]$ and $I^{(\mathrm{D})} \subseteq [\![N_{\mathrm{D}}]\!]$.

This is achieved by assigning large enough values to $\epsilon_{\mathrm{loc}_{i_{\mathrm{C}}}}^{(\mathrm{C})}, \epsilon_{\mathrm{loc}_{i_{\mathrm{D}}}}^{(\mathrm{D})}$, and setting $\delta_{\mathrm{loc}_{i_{\mathrm{C}}}}^{(\mathrm{C})} = 0$ for all $i_{\mathrm{C}} \notin I^{(\mathrm{C})}, i_{\mathrm{D}} \notin I^{(\mathrm{D})}$. See the bottom panel of Fig. 1 for an illustration, and Appendix H for the posterior privacy scheme. However, generating samples from $H^{\mathrm{Pert}}$ is challenging, as it jointly models dependencies across both continuous and discrete marginals. Algorithm 1 in the Appendix outlines a sampling procedure inspired by the *latent Gaussian copula approach* (Fan et al., 2017).

**Proposition 2** *Consider $H^{\mathrm{Pert}}$ defined by Eq. 5, and let $H$ be the joint CDF modeled by a semi-Gaussian copula with estimated correlation matrix $\hat{R}$ and marginals $H_{i_{\mathrm{C}}}^{(\mathrm{C})}, H_{i_{\mathrm{D}}}^{(\mathrm{D})}$. If $\delta_{\mathrm{loc}_{i_{\mathrm{C}}}}^{(\mathrm{C})} = 0$, and $\epsilon_{\mathrm{loc}_{i_{\mathrm{C}}}}^{(\mathrm{C})} \to \infty, \epsilon_{\mathrm{loc}_{i_{\mathrm{D}}}}^{(\mathrm{D})} \to \infty$ for all $i_{\mathrm{C}} \in [\![N_{\mathrm{C}}]\!] \setminus I^{(\mathrm{C})}, i_{\mathrm{D}} \in [\![N_{\mathrm{D}}]\!] \setminus I^{(\mathrm{D})}$, then $\lim \left| H^{\mathrm{Pert}}(\mathbf{t}) - H(\mathbf{t}) \right| \leq \sum_{i_{\mathrm{C}} \in I^{(\mathrm{C})}} \left| F_{\mathrm{AGM}, i_{\mathrm{C}}}(t_{i_{\mathrm{C}}}) - H_{i_{\mathrm{C}}}^{(\mathrm{C})}(t_{i_{\mathrm{C}}}) \right| + \sum_{i_{\mathrm{D}} \in I^{(\mathrm{D})}} \left| F_{\mathrm{RRM}, i_{\mathrm{D}}}(t_{N_{\mathrm{C}}+i_{\mathrm{D}}}) - H_{i_{\mathrm{D}}}^{(\mathrm{D})}(t_{N_{\mathrm{C}}+i_{\mathrm{D}}}) \right|$. Moreover, in the unperturbed case where $\delta_{\mathrm{loc}_{i_{\mathrm{C}}}}^{(\mathrm{C})} = 0$ and $\epsilon_{\mathrm{loc}_{i_{\mathrm{C}}}}^{(\mathrm{C})}, \epsilon_{\mathrm{loc}_{i_{\mathrm{D}}}}^{(\mathrm{D})} \to \infty$ for all $i_{\mathrm{C}} \in [\![N_{\mathrm{C}}]\!], i_{\mathrm{D}} \in [\![N_{\mathrm{D}}]\!]$, we have $H^{\mathrm{Pert}} \xrightarrow{d} H$.*

Proposition 2(ii) confirms that, when $\delta_{\mathrm{loc}_{i_{\mathrm{C}}}}^{(\mathrm{C})} = 0$ and $\epsilon_{\mathrm{loc}_{i_{\mathrm{C}}}}^{(\mathrm{C})}, \epsilon_{\mathrm{loc}_{i_{\mathrm{D}}}}^{(\mathrm{D})} \to \infty$ for all relevant data columns, the copula-based base measure imposes no perturbation and solely enhances generalization.

**Remark 2** *Following Steinke (2022, Theorem 3.22), for any $\delta > 0$, the composition of differentially private mechanisms per-continuous attribute privatization is $(\epsilon, \ \delta + \delta_{\mathrm{glo}} + \delta_{\mathrm{loc}_{i_{\mathrm{C}}}}^{(\mathrm{C})})$-differential private, where $\epsilon = \min \left\{ \epsilon_{\mathrm{glo}} + \epsilon_{\mathrm{loc}_{i_{\mathrm{C}}}}^{(\mathrm{C})}, \ \frac{1}{2}(\epsilon_{\mathrm{glo}}^2 + (\epsilon_{\mathrm{loc}_{i_{\mathrm{C}}}}^{(\mathrm{C})})^2) + \sqrt{2 \ln(1/\delta)(\epsilon_{\mathrm{glo}}^2 + (\epsilon_{\mathrm{loc}_{i_{\mathrm{C}}}}^{(\mathrm{C})})^2)} \right\}$. A similar expression applies per-discrete attribute privatization.*

### 4.4 TABULAR DATA SYNTHESIS

To train our generative model, we first must preprocess the posterior samples drawn from the DirP. Given these samples, they are fed into a BNP conditional VAE-GAN network whose details we will describe in this section.

#### 4.4.1 CONDITIONING LEARNING ON PROTECTED ATTRIBUTES

**Preprocessing:** Given a tabular dataset $\mathcal{D}_{1:n} \sim F$ with $F \sim \mathrm{DP}(a, H^{\mathrm{Pert}})$, let $\mathcal{D}_{1:N}^{\mathrm{Pos}} = (\mathbf{X}_{1:N}^{\mathrm{Pos}}, \mathbf{Y}_{1:N}^{\mathrm{Pos}}, \mathbf{S}_{1:N}^{\mathrm{Pos}})$ denote $N$ realizations from the DirP approximation in Eq. 3, corresponding to $\mathcal{D}^{\mathrm{Pos}} \sim F^{\mathrm{Pos}}$. The structure of $\mathcal{D}_{1:N}^{\mathrm{Pos}}$ follows the same representation as in Eq. 4, except that $N$ replaces $n$, and each term carries the superscript "Pos" to indicate that it is the posterior distribution. To preprocess the posterior sample for the subsequent network architecture, the continuous columns are quantile-transformed via $\breve{\mathcal{D}}_{1:N,i_{\mathrm{C}}}^{\mathrm{Pos}(\mathrm{C})} = \Phi_{\mathrm{U}}^{-1}(\mathcal{D}_{1:N,i_{\mathrm{C}}}^{\mathrm{Pos}(\mathrm{C})})$, where $\Phi_{\mathrm{U}}$ is the CDF of the standard uniform distribution. This step addresses the multi-modal of numerical features that hinder GAN training, improving stability and sample quality without relying on mixture-based techniques (Rajabi & Garibay, 2022). Each discrete variable $\mathcal{D}_{1:N,i_{\mathrm{D}}}^{\mathrm{Pos}(\mathrm{D})} \in [\![K_{i_{\mathrm{D}}}]\!]$ is encoded as a one-hot vector $\dot{\mathcal{D}}_{1:N,i_{\mathrm{D}}}^{\mathrm{Pos}(\mathrm{D})} \in \mathbb{R}^{K_{i_{\mathrm{D}}}}$, with 1 at the observed category and 0 elsewhere. The complete transformed dataset is shown in Eq. 6. Hereafter, one-hot representations are marked by a "superscript dot".

$$\breve{\dot{\mathcal{D}}}_{1:N}^{\mathrm{Pos}} = \Big( \overbrace{\underbrace{\breve{\mathcal{D}}_{1:N,1}^{\mathrm{Pos}(\mathrm{C})}, \ldots, \breve{\mathcal{D}}_{1:N,N_{\mathrm{C}}}^{\mathrm{Pos}(\mathrm{C})}, \dot{\mathcal{D}}_{1:N,1}^{\mathrm{Pos}(\mathrm{D})}, \ldots, \dot{\mathcal{D}}_{1:N,N_{\mathrm{D}}-2}^{\mathrm{Pos}(\mathrm{D})}}_{\breve{\dot{\mathbf{X}}}_{1:N}^{\mathrm{Pos}}}}^{\breve{\dot{\mathbf{X}}}\dot{\mathbf{Y}}_{1:N}^{\mathrm{Pos}} := (\breve{\dot{\mathbf{X}}}_{1:N}^{\mathrm{Pos}}, \dot{\mathbf{Y}}_{1:N}^{\mathrm{Pos}}) \text{ with total feature dimension } d_1' = N_{\mathrm{C}} + \sum_{i_{\mathrm{D}}=1}^{N_{\mathrm{D}}-1} K_{i_{\mathrm{D}}}}, \underbrace{\dot{\mathcal{D}}_{1:N,N_{\mathrm{D}}-1}^{\mathrm{Pos}(\mathrm{D})}}_{\dot{\mathbf{Y}}_{1:N}^{\mathrm{Pos}}}, \overbrace{\underbrace{\dot{\mathcal{D}}_{1:N,N_{\mathrm{D}}}^{\mathrm{Pos}(\mathrm{D})}}_{\dot{\mathbf{S}}_{1:N}^{\mathrm{Pos}}}}^{d_2' = K_{N_{\mathrm{D}}}} \Big). \tag{6}$$

**BNPL:** Let $G_{\boldsymbol{\omega}}$ be a neural network generator in the class of GDMs, parameterized by $\boldsymbol{\omega} \in \boldsymbol{\Omega}$. Given a distance $\triangle$, a robust posterior approximation of $\boldsymbol{\omega}$ can be obtained by optimizing Eq. 7 to enforce $G_{\boldsymbol{\omega}}$ to generate realistic tabular samples (Dellaporta et al., 2022):

$$\boldsymbol{\omega}^*(F^{\mathrm{Pos}}) := \arg \min_{\boldsymbol{\omega} \in \boldsymbol{\Omega}} \triangle(F_{\breve{\dot{\mathcal{D}}}_{1:N}^{\mathrm{Pos}}}^{\mathrm{Pos}}, F_{G_{\boldsymbol{\omega}}}). \tag{7}$$

**BNP-VAE$\mathcal{C}$GAN:** We use a generative model that learns a latent representation by minimizing reconstruction and regularization errors in a VAE composed of an encoder mapping the data to the latent space and a decoder which reconstructs the data (Fazeli-Asl & Zhang, 2023). The $\mathcal{C}$ denotes a code-GAN that includes only a code-generator and uses MMD as a discriminator to explore underrepresented code regions. The GAN comprises of a generator that maps latent variables $\ell \in \mathbb{R}^p$ to synthetic samples. The decoder is then shared with the generator to enable joint optimization.

**Architecture:** We define our NN-based generator, encoder, code-generator, and discriminator functions conditioned on $\dot{\mathbf{S}}_{1:N}^{\text{Pos}}$ as $G_{\boldsymbol{\omega}} : \mathbb{R}^{p \times d_2'} \to \mathbb{R}^{d_1'}$, $E_{\boldsymbol{\eta}} : \mathbb{R}^{d_1' \times d_2'} \to \mathbb{R}^p$, $CG_{\boldsymbol{\tau}} : \mathbb{R}^{q \times d_2'} \to \mathbb{R}^p$, and $D_{\boldsymbol{\theta}} : \mathbb{R}^{d_1' \times d_2'} \to \mathbb{R}$, with parameters $\boldsymbol{\omega} \in \boldsymbol{\Omega}$, $\boldsymbol{\eta} \in \mathcal{H}$, $\boldsymbol{\tau} \in \mathcal{T}$, and $\boldsymbol{\theta} \in \boldsymbol{\Theta}$, respectively. We train the CBNP-VAE$\mathcal{C}$GAN by minimizing the composite objective (Eq. 8) with respect to its respective network parameters:

$$\mathcal{L}_{\text{Utility}}(G_{\boldsymbol{\omega}}, E_{\boldsymbol{\eta}}, D_{\boldsymbol{\theta}}, CG_{\boldsymbol{\tau}}) = \mathcal{L}(G_{\boldsymbol{\omega}}, E_{\boldsymbol{\eta}} \mid \dot{\mathbf{S}}^{\text{Pos}}) + \mathcal{L}(D_{\boldsymbol{\theta}} \mid \dot{\mathbf{S}}^{\text{Pos}}) + \mathcal{L}(CG_{\boldsymbol{\tau}} \mid \dot{\mathbf{S}}^{\text{Pos}}). \tag{8}$$

The conditional generator is fed with $(\boldsymbol{\ell}_{1:N}, \dot{\mathbf{S}}_{1:N}^{\text{Pos}})$ to produce synthetic samples $\widetilde{\check{\mathbf{X}}\dot{\mathbf{Y}}}_{1:N}^{\text{Pos}}$. Since sensitive attributes are never generated, the framework allows omitting their privacy constraint $(\epsilon_{\text{N}_{\text{D}}} = \infty)$ in equation 5, preventing the removal of minority sensitive groups during perturbation.

### 4.4.2 Employing DirPMINE as a Fairness Regularizer

The structure of CBNP-VAE$\mathcal{C}$GAN flexibly enforces SP fairness by regularizing the loss function in Eq. 8 with an MI penalty, $\text{MI}(\widetilde{\mathbf{Y}}^{\text{Pos}}, \dot{\mathbf{S}}^{\text{Pos}})$, to reduce dependence on protected attributes. However, this remains challenging due to intractable likelihoods and the curse of dimensionality in high-dimensional spaces. The following definition shows that MI can be approximated by maximizing a variational form of the DV lower bound[1] (DVLB, Donsker & Varadhan, 1975), derived from the KL representation of MI and claimed as a scalable estimator (Fazeli-Asl et al., 2026, see Figs. 3-4 and Table 1). It is used here as a proxy for fairness training in high-dimensional settings.

**Definition 1 (DirPMINE (Fazeli-Asl et al., 2026))** *Let $T_{\boldsymbol{\upsilon}} : \mathbb{R}^{d_1' \times d_2'} \to \mathbb{R}$ be an NN over compact $\boldsymbol{\Upsilon}$. Then $\text{MI}(\widetilde{\mathbf{Y}}^{\text{Pos}}, \dot{\mathbf{S}}^{\text{Pos}})$ is approximated as follows using a random permutation $[\![\pi(N)]\!]$ of $[\![N]\!]$.*

$$\overbrace{\text{MI}^{DirPDV}(\widetilde{\mathbf{Y}}_{1:N}^{\text{Pos}}, \dot{\mathbf{S}}_{1:N}^{\text{Pos}})}^{\mathcal{L}_{Fair}(G_{\boldsymbol{\omega}})} = \max_{\boldsymbol{\upsilon} \in \boldsymbol{\Upsilon}} \Big\{ \overbrace{\sum_{r=1}^{N} J_{r,N-1}^{(a+n)} T_{\boldsymbol{\upsilon}}(\widetilde{\mathbf{Y}}_r^{\text{Pos}}, \dot{\mathbf{S}}_r^{\text{Pos}}) - \ln \sum_{r=1}^{N} J_{r,N-1}^{(a+n)} e^{T_{\boldsymbol{\upsilon}}(\widetilde{\mathbf{Y}}_r^{\text{Pos}}, \dot{\mathbf{S}}_{\pi(r)}^{\text{Pos}})}}^{\text{The BNP Approximation of DVLB}} \Big\}.$$

The framework manages the privacy–fairness tension by applying the localized privacy mechanism once before fairness optimization, and its manually set per-attribute budgets, light for outcomes and none for sensitive attributes, help prevent localized noise from distorting the group-conditioned statistics needed for fairness, especially for small sensitive groups.

### 4.4.3 Class Balancing: A Potential Driver of Fairness

CBNP-VAE$\mathcal{C}$GAN ensures class balance across protected attributes via generator conditioning. To extend balancing to unprotected discrete features $i_{\text{D}} \in [\![N_{\text{D}}-2]\!]$, we define $U_{i_{\text{D}}} \sim \text{Cat}(K_{i_{\text{D}}}, 1/K_{i_{\text{D}}})$ and minimize $\text{D}_{\text{KL}}(\widetilde{\boldsymbol{\mathcal{D}}}_{i_{\text{D}}}^{\text{Pos(D)}}, U_{i_{\text{D}}}) = \sum_{j=1}^{K_{i_{\text{D}}}} \tilde{p}_{i_{\text{D}}j}^{\text{Pos}} \ln(\tilde{p}_{i_{\text{D}}j}^{\text{Pos}} K_{i_{\text{D}}})$, where $\tilde{p}_{i_{\text{D}}j}^{\text{Pos}}$ is the generated class proportion. This divergence is estimated via a semi-BNP DVLB with $N$ one-hot samples $\dot{\mathbf{U}}_{1:N,i_{\text{D}}}$ and auxiliary functions $T_{\boldsymbol{\upsilon}_{i_{\text{D}}}} : \mathbb{R}^{K_{i_{\text{D}}}} \to \mathbb{R}$:

$$\overbrace{\text{D}_{\text{KL}}^{\text{DirPDV}}(\widetilde{\boldsymbol{\mathcal{D}}}_{1:N,i_{\text{D}}}^{\text{Pos(D)}}, \dot{\mathbf{U}}_{1:N,i_{\text{D}}})}^{\mathcal{L}_{\text{Balance}_{i_{\text{D}}}}(G_{\boldsymbol{\omega}})} = \max_{\boldsymbol{\upsilon}_{i_{\text{D}}} \in \boldsymbol{\Upsilon}_{i_{\text{D}}}} \Big\{ \sum_{r=1}^{N} J_{r,N-1}^{(a+n)} T_{\boldsymbol{\upsilon}_{i_{\text{D}}}}(\widetilde{\boldsymbol{\mathcal{D}}}_{i_{\text{D}},r}^{\text{Pos(D)}}) - \ln \sum_{r=1}^{N} \frac{1}{N} e^{T_{\boldsymbol{\upsilon}_{i_{\text{D}}}}(\dot{\mathbf{U}}_{i_{\text{D}},r})} \Big\}.$$

The final loss, scaled by $\lambda_{\text{F}}, \lambda_{\text{B}_{i_{\text{D}}}} \in [0, 1]$, is then given by Eq. 9. The generator's outputs are eventually reverted by inverse QT for continuous and argmax decoding for categorical columns.

$$\mathcal{L}_{\text{Utility}}(G_{\boldsymbol{\omega}}, E_{\boldsymbol{\eta}}, D_{\boldsymbol{\theta}}, CG_{\boldsymbol{\tau}}) + \lambda_{\text{F}} \mathcal{L}_{\text{Fair}}(G_{\boldsymbol{\omega}}) + \sum_{i_{\text{D}}=1}^{N_{\text{D}}-2} \lambda_{\text{B}_{i_{\text{D}}}} \mathcal{L}_{\text{Balance}_{i_{\text{D}}}}(G_{\boldsymbol{\omega}}). \tag{9}$$

---

[1] For CDFs $F_{\mathbf{X}}, F_{\mathbf{Z}}$, and $T_{\boldsymbol{\upsilon}} : \mathbb{R}^{\dim(\mathbf{X})} \to \mathbb{R}$, DVLB $= \mathbb{E}_{F_{\mathbf{X}}}[T_{\boldsymbol{\upsilon}}(\mathbf{X})] - \ln \mathbb{E}_{F_{\mathbf{Z}}}[e^{T_{\boldsymbol{\upsilon}}(\mathbf{Z})}] \leq \text{D}_{\text{KL}}(F_{\mathbf{X}}, F_{\mathbf{Z}})$.

We provide a strong theoretical guarantee that the key result—showing that utility, fairness, and class imbalance are jointly addressed by the method—is preserved, as stated in Theorem 2 in the Appendix G. This guarantee is further validated by the results in Appendix I. Together, these credible findings make the proposed framework clearly distinguishable from other works that lack a truly joint treatment of these aspects.

## 5 EXPERIMENTAL RESULTS

We assess the utility of our proposed model by using MMD and three classifiers (DTS, LR, MLP) (Hastie et al., 2009), each trained on 80% of the original data and validated on the remaining 20% as a true baseline, with utility measured using accuracy and F1 scores (Rainio et al., 2024). The classifiers are then trained on synthetic data generated by the learned conditional generator with **r**eal **s**ensitive **r**atio inputs ($\mathbf{S}_{rsr}$) and tested on the same 20% original, with each score averaged over 10 runs.

**Adult Dataset :** It includes 6 continuous attributes and 9 discrete attributes with $\mathbf{S}_{rsr}$ consisting of 67.5% males and 32.5% females. (Appendix I.3.1). It is known for its `income` (**I**) bias (>50K vs. $\leq$50K) across `sex` (**S**). Here, $\mathbf{Y}$ = "`income`" and $\mathbf{S}$ = "`sex`".

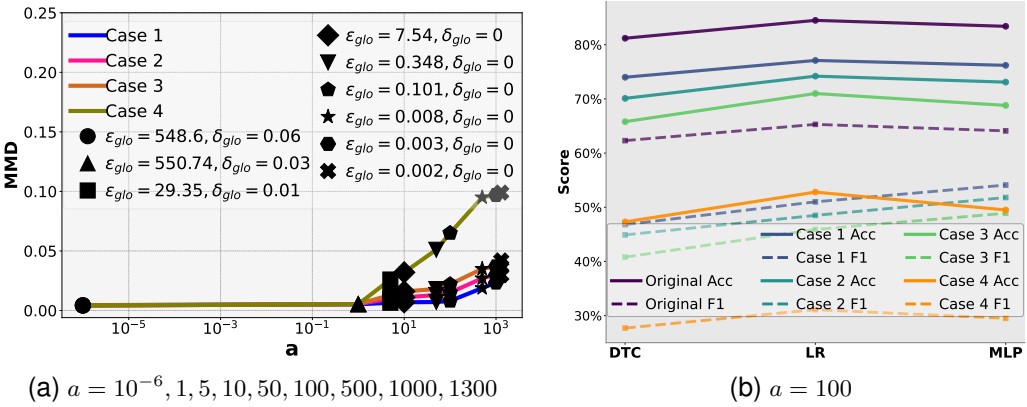

(a) $a = 10^{-6}, 1, 5, 10, 50, 100, 500, 1000, 1300$      (b) $a = 100$

Figure 2: Privacy investigation under different global and local budgets when $\lambda_F = 0$.

**Privacy-Preserving:** Four designs are considered for different $a$ values: **(1)** No local privacy ($\epsilon_{loc} = \infty$), **(2)** (2.5, 0.05)-local privacy on continuous attributes 'A' and '**HPW**', **(3)** the same budget on discrete attributes $\overline{\mathbf{MS}}$ and $\overline{\mathbf{Ra}}$ in addition to the two continuous attributes in Case 2 (four attributes in total), and **(4)** the same budget on all attributes. Fig. 2a shows global privacy budgets from Proposition 1 using $|W| = 3, \eta = \bar{\eta} = 0.2, \gamma = 0.01$ as recommended in Gohari et al. (2021), illustrating a trade-off where increasing $a$ and the number of privatized attributes reduces utility, validating Corollary 1 and Proposition 2. We also note that global privacy based on parameter $a$ refers to the proportion of posterior samples generated from $H^{\text{Pert}}$ because of the mixture form of the posterior base measure $H^*$, whereas local privacy refers to the intensity of noise injected into the marginals of the samples generated from $H^{\text{Pert}}$. Hence, for sufficiently small values of $a$ (e.g., $a = 10^{-6}$), even a strong local privacy budget does not lead to meaningful perturbation of the marginals, effectively leading a clean model (see Appendix H). This clearly shows how our privacy mechanism maintains good utility for small values of $a$ even under a strong local privacy budget. Fig. 2b shows that adding more privatized attributes under high global privacy ($a = 100$) leads to worse accuracy and F1.

**Fairness-Aware:** Given $a = 5$ for light global privacy budget (29.35, 0.01), experiments now proceed with $\lambda_F = 1$ for Case 4, comparing ours with FairGAN (Xu et al., 2018), DECAF, and TabFair GAN. Fairness in ours is evaluated using SP = $|\Pr(\widetilde{\mathbf{Y}} > 50K \mid \mathbf{S}_{rsr} = \text{Male}) - \Pr(\widetilde{\mathbf{Y}} > 50K \mid \mathbf{S}_{rsr} = \text{Female})|$, and MI($\widetilde{\mathbf{Y}}, \mathbf{S}_{rsr}$). In others, $\mathbf{S}_{rsr}$ is replaced with the corresponding synthesis $\widetilde{\mathbf{S}}$. Fig. 3 shows that ours outperforms others in faithfulness and fairness, particularly the state-of-the-art DECAF, while being a strong competitor with imposed uncertainty.

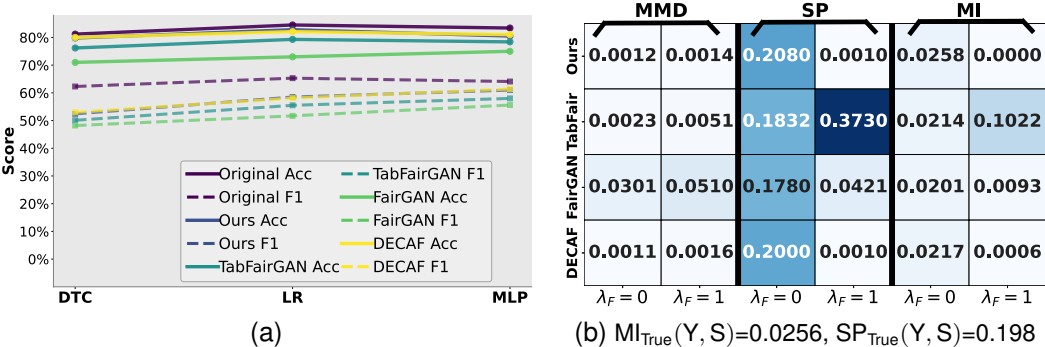

Figure 3: **Adult**: Fairness-Aware investigation for Case 4 given $a = 5$ over 10 runs.

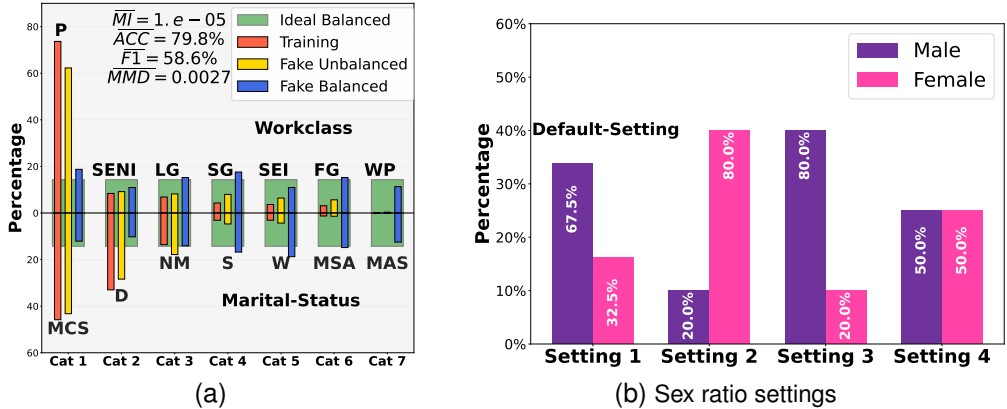

Figure 4: **Adult**: Class balanced investigation for Case 4 given $a = 5$ and $\lambda_F = 1$ over 10 runs.

**Class-Balancing:** Completing our investigation, Fig. 4a shows category frequencies in real and synthetic data, averaged across sex ratio settings from Fig. 4b, input to the learned conditional generator and balanced on 'W' and 'MS' with $\lambda_F = 1$ and $a = 5$ for Case 4. Despite this balancing, utility and fairness remain strong.

COMPAS Dataset : It includes 3 continuous and 15 discrete attributes. As shown in Table 13 and Eq. 35 (Appendix I.4.3), the score_text attribute exhibits disparities in risk classification across ethnic groups. The corresponding MI is 0.0259, underscoring the disparity. Any generative model should aim to minimize these disparities. Here, $\mathbf{Y}$ = "score_text" and $\mathbf{S}$ = "ethnic" have 3 and 8 categories, respectively, making standard fairness baselines inapplicable. This highlights a limitation of existing methods and motivates our approach. As shown in Table 14 (Appendix I.4.3), several groups (e.g., the Oriental category) appear as minorities in the original dataset, increasing the risk of mode mixing. To address this, we set the protected attribute $\mathbf{S}$ to have equal proportions across all ethnic groups and use it as an input to the conditional generator for supervised learning.

For privacy, we jointly apply a $(2.5, 0.05)$-local (across all attributes) together with a $(29.35, 0.01)$-global privacy budget, allowing us to evaluate performance under combined privacy constraints.

The conditional distribution of score_text across different ethnic categories for the generated samples is shown in Table 1, with orange indicating generation without fairness constraints and blue indicating generation with fairness constraints applied. Comparing the orange part with the corresponding values from the original dataset in Table 13 confirms that the model preserves the underlying distribution (i.e., perfect utility).

Table 1: $\Pr(\mathbf{Y} = \text{``Score Text''} \mid \mathbf{S} = \text{``Ethnic''})$ for the generated samples.

| Ethnic Group | Score Text ($\lambda_F = 0$) | | | Score Text ($\lambda_F = 1$) | | |
|---|---|---|---|---|---|---|
| | Low | Medium | High | Low | Medium | High |
| African-American | 0.603 | 0.251 | 0.146 | 0.621 | 0.276 | 0.103 |
| Arabic | 0.759 | 0.164 | 0.077 | 0.631 | 0.230 | 0.139 |
| Asian | 0.826 | 0.116 | 0.058 | 0.635 | 0.264 | 0.101 |
| Caucasian | 0.734 | 0.185 | 0.081 | 0.629 | 0.234 | 0.137 |
| Hispanic | 0.773 | 0.162 | 0.065 | 0.624 | 0.253 | 0.123 |
| Native American | 0.673 | 0.212 | 0.115 | 0.618 | 0.211 | 0.171 |
| Oriental | 0.817 | 0.121 | 0.062 | 0.676 | 0.214 | 0.110 |
| Other | 0.809 | 0.143 | 0.048 | 0.683 | 0.202 | 0.115 |

As shown in red, the generated data exhibits disparities in `score_text` across `ethnic` groups without a fairness policy. For instance, African-Americans are classified as "High" risk with probability 0.146, compared to only 0.058 for Asians. These probabilities become nearly equal in the blue portions.

Table 2: COMPAS: $\text{MI}_{\text{True}}(\mathbf{Y}, \mathbf{S}) = 0.0259$, $\text{F1}_{\text{True}} = 0.913$, and $\text{Acc}_{\text{True}} = 0.901$ over 10 runs.

| $\lambda_B = 1$ | $\lambda_F$ | MI | MMD | F1 (DTC) | Acc (DTC) |
|---|---|---|---|---|---|
| None | 0 | 0.0263 | 0.0011 | 0.921 | 0.894 |
| | 1 | $1e-5$ | 0.0018 | 0.903 | 0.886 |
| Sex Marital Language | 1 | $1e-5$ | 0.0024 | 0.917 | 0.883 |

Table 2 analyzes model performance under fairness policies. The first block reports the proposed model without fairness and its improvement when fairness drives MI near zero. The second block activates three categorical variables (Sex, Marital, Language) to balance generation across all their categories. In all cases, MMD between generated and original samples remains negligible, indicating strong data utility, and F1 and Accuracy on the generated data stay close to their values on the original data, confirming that predictive performance is preserved.

Despite the widespread use of the Adult and COMPAS datasets in algorithmic fairness research, recent work has noted limitations in relying on them for evaluating fairness guarantees (Bao et al., 2021; Ding et al., 2021); see the Appendix for a brief review. To address this, we include additional experiments in the Appendix using the Bank Marketing dataset. We also provide a controlled toy example that enables a more isolated analysis of specific components of the proposed approach.

## 6 CONCLUDING REMARKS

We propose a conditional BNPL framework for private, fair, and balanced tabular data synthesis. By conditioning on sensitive attributes, we can effectively address challenges in training generative models, like preserving faithfulness and preventing mode collapse, with theoretical guarantees validated through numerical results. Our fairness term is based on regular MI, the same structure can be extended to conditional MI, allowing compatibility with fairness notions such as equalized odds, equality of opportunity, predictive equality, and conditional SP, though several of these are inherently classifier-oriented rather than generator-oriented. Although the basic structure of the GAN is different from large language models (LLMs) like ChatGPT, we believe that our proposed approach can still inspire private and fair language models. In future work, we plan to extend this framework by incorporating Dirichlet processes into LLMs, as a mechanism to provide stronger privacy and fairness guarantees.

ACKNOWLEDGMENTS

Bei Jiang and Linglong Kong were partially supported by grants from the Canada CIFAR AI Chairs program, the Alberta Machine Intelligence Institute (AMII), and Natural Sciences and Engineering Council of Canada (NSERC), and Linglong Kong was also partially supported by grants from the Canada Research Chair program from NSERC. Michael Zhang was supported by the University of Hong Kong Seed Fund for PI Research #2402101367. The authors would also like to thank the anonymous reviewers for their constructive comments that improved the quality of this article.

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

## A  NOTATION

### Numbers and Arrays

| | |
|---|---|
| $a$ | A scalar (integer or real) |
| $\boldsymbol{a}$ | A vector |
| $\boldsymbol{I}_n$ | Identity matrix with $n$ rows and $n$ columns |
| $X$ | A scalar random variable |
| $\mathbf{X}$ | A vector-valued random variable |
| $\mathbf{X}_{1:n}$ | A matrix containing $n$ realizations of the random vector $\mathbf{X}$ (for a $d$-dimensional $\mathbf{X}$, the matrix has $n$ rows and $d$ columns). |
| $\mathbf{X}_{1:n,i}$ | Column $i$ of $\mathbf{X}_{1:n}$. |
| $\mathbf{X}_{1:n,1:m}$ | Columns 1 through $m$ of $\mathbf{X}_{1:n}$. |

### Sets and Graphs

| | |
|---|---|
| $\mathbb{R}$ | The set of real numbers |
| $\mathbb{R}^d$ | The set of all $d$-dimensional real-valued vectors. |
| $\mathbb{R}^{n \times N_C}$ | The set of all $n \times N_C$ matrices with entries in $\mathbb{R}$. |
| $\mathbb{Z}_+$ | The set of non-negative integers. |
| $\mathbb{Z}_+^{n \times N_D}$ | The set of all $n \times N_D$ matrices with entries in $\mathbb{Z}_+$. |
| $|W|$ | The cardinality of the set $W$, representing the number of elements in $W$. |
| $\{1,\dots,N\}$ | The set containing $1,\dots,N$. |
| $[\![N]\!]$ | The set of all integers between 1 and $N$, i.e., $\{1,\dots,N\}$. |
| $[a,b]$ | The real interval including $a$ and $b$ |
| $(a,b]$ | The real interval excluding $a$ but including $b$ |
| $\mathbb{A}\backslash\mathbb{B}$ | Set subtraction, i.e., the set containing the elements of $\mathbb{A}$ that are not in $\mathbb{B}$ |

### Indexing

| | |
|---|---|
| $a_i$ | Element $i$ of vector $\boldsymbol{a}$, with indexing starting at 1 |
| $\mathbf{A}_{i,j}$ | Element $i, j$ of matrix $\mathbf{A}$ |
| $\dim(\mathbf{A})$ | The dimensions of the matrix $\mathbf{A}$, typically given as (number of rows $\times$ number of columns). |
| $\mathbf{A}^{(\mathrm{D})}$ | The categorical-valued columns of $\mathbf{A}$. |
| $\mathbf{A}^{(\mathrm{C})}$ | The continuous-valued columns of $\mathbf{A}$. |
| $N_{\mathrm{D}}$ | The number of categorical-valued columns in matrix $\mathbf{A}$. |
| $N_{\mathrm{C}}$ | The number of continuous-valued columns in matrix $\mathbf{A}$. |
| $K_{i_{\mathrm{D}}}$ | The number of categories in the $i_{\mathrm{D}}$-th categorical-valued column of $\mathbf{A}$, where $i_{\mathrm{D}} \in \{1, \ldots, N_{\mathrm{D}}\}$. |
| $K_{i_{\mathrm{C}}}$ | The number of categories in the $i_{\mathrm{C}}$-th continuous-valued column of $\mathbf{A}$, where $i_{\mathrm{C}} \in \{1, \ldots, N_{\mathrm{C}}\}$. |
| $(\epsilon_{\mathrm{glo}}, \delta_{\mathrm{glo}})$ | Global privacy parameters of the Dirichlet Process mechanism applied to the entire underlying data distribution. |
| $(\epsilon_{\mathrm{loc}_{i_{\mathrm{C}}}}^{(\mathrm{C})}, \delta_{\mathrm{loc}_{i_{\mathrm{C}}}}^{(\mathrm{C})})$ | Localized privacy parameters of a differentially private mechanism applied to the $i_{\mathrm{C}}$-th continuous column. |
| $\epsilon_{\mathrm{loc}_{i_{\mathrm{D}}}}^{(\mathrm{D})}$ | Localized privacy parameters of a differentially private mechanism applied to the $i_{\mathrm{D}}$-th continuous column. |
| $\dot{\mathbf{A}}$ | One-hot matrix. For a categorical-valued vector $\boldsymbol{a} = (a_1, \ldots, a_n)$, the one-hot representation is a matrix whose $i$-th row is a one-hot vector of the form $[0, \ldots, 0, 1, 0, \ldots, 0]$, with a single 1 indicating the active category of $a_i$ and 0s elsewhere. |
| $\widetilde{\dot{\mathbf{A}}}$ | The one-hot–encoded columns of the matrix generated to match the one-hot training matrix $\dot{\mathbf{A}}$, produced by a generator function. |
| $\breve{\mathbf{A}}$ | The quantile-transformed version of the continuous-valued matrix/vector $\mathbf{A}$. |
| $\mathring{\mathbf{A}}$ | A tabular-valued matrix containing a mix of categorical and continuous variables, where "$\dot{}$" indicates the one-hot encoded categorical columns and "$\breve{}$" indicates the quantile-transformed continuous columns. |
| $\widetilde{\mathring{\mathbf{A}}}$ | The generated tabular matrix, produced by a generator function, matching the tabular training matrix $\mathring{\mathbf{A}}$. |
| $\mathring{\mathbf{A}}\dot{\mathbf{B}}$ | The column-wise concatenation of the tabular matrix $\mathring{\mathbf{A}}$ with the one-hot matrix $\dot{\mathbf{B}}$. |
| $\mathbf{S}_{\mathrm{rsr}}$ | Randomized sensitive attribute with a sample size equal to the mini-batch, constructed so that the proportion of each label matches that of the true sensitive attribute $\mathbf{S}$. Used as the conditional input to the generator in the experimental section. |

**Calculus**

| | |
|---|---|
| $\nabla_{\boldsymbol{x}}$ | Gradient with respect to $\boldsymbol{x}$ |
| $\int f(\boldsymbol{x})d\boldsymbol{x}$ | Definite integral over the entire domain of $\boldsymbol{x}$ |
| $\int_{\mathbb{S}} f(\boldsymbol{x})d\boldsymbol{x}$ | Definite integral with respect to $\boldsymbol{x}$ over the set $\mathbb{S}$ |

### Probability and Information Theory

| | |
|---|---|
| $F(A)$ | A probability of set $A$ under $F$. variable |
| $X \sim F$ | Random variable $X$ has distribution $F$ |
| $\mathbb{I}_X(A)$ | Indicator function taking value 1 if $X \in A$ and 0 otherwise |
| $F_{X_{1:n}}$ | Empirical distribution based on the sample $X_{1:n}$ |
| $\mathrm{DP}(a, H)$ | Dirichlet Process prior, with $a > 0$ the concentration parameter and $H$ the base measure. |
| $F^{\mathrm{Pri}}$ | A random probability measure drawn from a Dirichlet Process prior, i.e., $F^{\mathrm{pri}} := (F \sim \mathrm{DP}(a, H))$. |
| $F^{\mathrm{Pos}}$ | A random probability measure drawn from a Dirichlet Process posterior, i.e., $F^{\mathrm{Pos}} := (F \sim \mathrm{DP}(a + n, H^*))$. |
| $H^*$ | Mixture probability measure combining the prior base measure $H$ and the empirical distribution $F_{X_{1:n}}$, given by $H^* = \dfrac{a}{a+n}H + \dfrac{n}{a+n}F_{X_{1:n}}$. |
| $\mathbb{E}_F[h(x)]$ or $\mathbb{E}h(x)$ | Expectation of $h(x)$ with respect to $F(\mathrm{x})$ |
| $\mathrm{Var}_F(h(x))$ | Variance of $h(x)$ under $F(\mathrm{x})$ |
| $d_{\mathrm{W}}(P, Q)$ | Wasserstein distance between two probability measures $P$ and $Q$ |
| $d_{\mathrm{MMD}}(P, Q)$ | Maximum mean discrepancy distance between two probability measures $P$ and $Q$ |
| $D_{\mathrm{KL}}(P, Q)$ | Kullback-Leibler divergence between two probability measures $P$ and $Q$ |
| $\mathbb{A}(P, Q)$ | A statistical distance between $P$ and $Q$. |
| $\mathrm{Dir}(\alpha_1, \ldots, \alpha_N)$ | Dirichlet distribution with parameter $(\alpha_1, \ldots, \alpha_N)$ |
| $N(\boldsymbol{\mu}, \boldsymbol{\Sigma})$ | Gaussian distribution with mean $\boldsymbol{\mu}$ and covariance $\boldsymbol{\Sigma}$ |
| $\mathrm{Cat}(K, \boldsymbol{p})$ | Categorical distribution over $K$ categories with probability vector $\boldsymbol{p} = (p_1, \ldots, p_j, \ldots, p_K)$. Each component $p_j$ denotes the probability of observing category $j$ among the $K$ possible categories of the categorical variable |
| $\xrightarrow{\text{a.s.}}$ | Almost sure convergence |
| $\xrightarrow{p}$ | Convergence in probability |
| $\xrightarrow{d}$ | Convergence in distribution |

### Functions

| | |
|---|---|
| $f : \mathbb{A} \to \mathbb{B}$ | The function $f$ with domain $\mathbb{A}$ and range $\mathbb{B}$ |
| $f(\boldsymbol{x}; \boldsymbol{\theta})$ | A function of $\boldsymbol{x}$ parametrized by $\boldsymbol{\theta}$. (Sometimes we write $f(\boldsymbol{x})$ and omit the argument $\boldsymbol{\theta}$ to lighten notation) |

| | |
|---|---|
| $\ln x$ | Natural logarithm of $x$ |
| $\|\boldsymbol{x}\|_p$ | $L^p$ norm of $\boldsymbol{x}$ |
| $\|\boldsymbol{x}\|$ | $L^2$ norm of $\boldsymbol{x}$ |
| $k_{\mathrm{RBF}}(\mathbf{X}, \mathbf{Y})$ | Radial Basis Function (RBF) kernel: $k_{\mathrm{RBF}}(\mathbf{X}, \mathbf{Y}) = \exp\!\big(-\gamma\|\mathbf{X} - \mathbf{Y}\|^2\big)$, where $\gamma > 0$ is the kernel bandwidth parameter. |

### Network Architecture

| | |
|---|---|
| $G_{\boldsymbol{\omega}}$ | A generator function parameterized by a neural network with parameters $\boldsymbol{\omega} \in \boldsymbol{\Omega}$. |
| $\boldsymbol{\xi}$ | A generator input (latent noise). |
| $\overset{\mathrm{Pos}}{\overset{\frown}{\mathbf{\mathring{X}}\mathbf{\mathring{Y}}}}$ | A generator output (generated posterior sample). |
| $E_{\boldsymbol{\eta}}$ | An encoder function parameterized by a neural network with parameters $\boldsymbol{\eta} \in \mathcal{H}$. |
| $\boldsymbol{c}$ | An encoder output. |
| $CG_{\boldsymbol{\tau}}$ | A code-generator function parameterized by a neural network with parameters $\boldsymbol{\tau} \in \mathcal{T}$. |
| $\boldsymbol{\zeta}$ | A code-generator input (sub-latent noise). |
| $\widetilde{\boldsymbol{c}}$ | A code-generator output (generated code). |
| $D_{\boldsymbol{\theta}}$ | A discriminator function parameterized by a neural network with parameters $\boldsymbol{\theta} \in \boldsymbol{\Theta}$. |
| $T_{\boldsymbol{\nu}}$ | A continuous function parameterized by a neural network with parameters $\boldsymbol{\nu} \in \boldsymbol{\Upsilon}$. |

### Acronyms

| | |
|---|---|
| AGM | Analytic Gaussian Mechanism |
| BNPL | Bayesian Nonparametric Learning |
| CBNP-VAE$\mathcal{C}$GAN | Conditional Bayesian Nonparametric framework with a triple generative model including a VAE, a Code-GAN, and a GAN |
| CDF | Cumulated Distribution Function |
| CGM | Classical Gaussian Mechanism |
| DirP | Dirichlet Process |
| DirPDV | Dirichlet Process approximation of Donsker-Varadhan representation of Kullback-Leibler Divergence |
| DirMINE | Dirichlet Process Mutual Information Neural Estimation |
| DTS | Decision Trees classifier |
| DV | Donsker-Varadhan representation of Kullback-Leibler Divergence |
| DVLB | Donsker-Varadhan Lower Bound of Kullback-Leibler Divergence |
| GAN | Generative Adversarial Network |

| | |
|---|---|
| GDM | Generator-decoder-based Models |
| IID | Independent and identically distributed. Indicates that random variables share the same distribution and are mutually independent. |
| KL | Kullback-Leibler Divergence |
| LLM | Large Language Model |
| LR | Logistic Regression classifier |
| MI | Mutual Information |
| MLP | Multilayer Perceptron classifier |
| NN | Neural Network |
| QT | Quantile-Transformed representation of continuous variables (maps data to a uniform or normal distribution using their empirical quantiles). |
| Semi-BNP DVLB | A semi–Bayesian nonparametric representation of the DVLB of the KL divergence, where one distribution in the KL is unknown and the other is known |
| SP | Statistical Parity |
| VAE | Variational Autoencoder |
| WMMD | Combined Wasserstein and Maximum Mean Discrepancy (MMD) distance |

## B   REVIEW OF KEY CONTRIBUTIONS

We begin our appendix discussion by noting that the contributions of this paper are novel and not derived from Fazeli-Asl & Zhang (2023), which proposed a VAE$\mathcal{C}$GAN under a BNPL framework focused solely on improving generation accuracy, without addressing privacy or fairness. In contrast, our work introduces a *private–fair–balanced* generative framework that is broadly applicable to any model within the class of Generator–Decoder-based Models (GDMs)—that is, generative models that include at least a generator or a decoder component—within the BNPL setting, by conditioning on sensitive attributes. This class encompasses a wide range of models, including GANs (Dellaporta et al., 2022; Fazeli-Asl et al., 2023), VAEs (Kingma et al., 2016), VAEGANs (Kwon et al., 2019; Fazeli-Asl & Zhang, 2023), Diffusion–GAN hybrids (DiffGANs) (Zhang et al., 2023), and Diffusion–VAE models (Della Maggiora et al., 2024; Kingma et al., 2021). Over this class of GDMs, our contributions can be summarized as follows:

1. **Privacy via DirP:** We are the first to introduce the Dirichlet Process (DirP) as an $(\epsilon, \delta)$-differential privacy mechanism within the BNPL setting. We provide theoretical guarantees for privacy budget control through perturbation of DirP samples. In addition to global privacy, a novel local privacy mechanism is proposed by modeling the base measure of the DirP using copula models, allowing structured perturbation at the distributional level. None of these privacy mechanisms were introduced or discussed in Fazeli-Asl & Zhang (2023).

2. **Fairness via Conditional Generator:** We propose a conditional generation strategy that minimizes the DirPMINE between generated outputs and sensitive attributes. Using DirP-MINE provides a scalable and effective fairness mechanism applicable to any sensitive attribute with more than two groups, an aspect not addressed in the existing literature.

3. **Class Balancing via DirPKL:** We introduce a class-balancing procedure using a DirP-based KL estimator. This supports both fairness and distributional privacy by preventing disproportionate representation across groups, another contribution not present in the literature.

We emphasize that the main contribution of this paper is not to provide a comprehensive comparison across members of GDMs, as such comparisons are already widely addressed in the literature, primarily for utility benchmarking, and could be revisited in a dedicated survey study. Instead, the focus of this work is on developing a principled approach to jointly enforce privacy on categorical and continuous features in tabular data while simultaneously preserving fairness and class balance. Here, class balance is treated as a complementary notion of fairness, reinforcing the fairness achieved through conditional generation with respect to sensitive attributes.

We chose to implement this triple policy approach using the VAE$\mathcal{C}$GAN architecture from Fazeli-Asl & Zhang (2023) to demonstrate improved utility and reduced mode collapse, but the methodology itself is general and original.

If the proposed triple-policy (fair–private–balanced) framework succeeds, practitioners could flexibly adopt any desired differential privacy mechanism for the categorical and continuous features of a tabular dataset though their relevant generative model in GDMs within our general framework.

We also note that the experimental results reported in this paper include only those baselines most closely aligned with our objectives—namely, methods for tabular data generation that incorporate statistical parity–based fairness within VAE-, GAN-, or hybrid-based architectures. Among these, FairGAN (Xu et al., 2018), DECAF (Van Breugel et al., 2021), and TabFairGAN (Rajabi & Garibay, 2022) are the most representative of this intersection, and their inclusion underscores our positioning as a next-step contribution in this line of research.

## C  CBNP-VAE$\mathcal{C}$GAN Loss Terms

Given $d'_1 = N_{\mathrm{C}} + \sum_{i_{\mathrm{D}}=1}^{N_{\mathrm{D}}-1}$, and $d'_2 = K_{N_{\mathrm{D}}}$, the NN-based generator, encoder, code-generator, and discriminator functions conditioned on $\dot{\mathbf{S}}^{\mathrm{Pos}}_{1:N}$ as $G_{\boldsymbol{\omega}} : \mathbb{R}^{p \times d'_2} \to \mathbb{R}^{d'_1}$, $E_{\boldsymbol{\eta}} : \mathbb{R}^{d'_1 \times d'_2} \to \mathbb{R}^p$, $CG_{\boldsymbol{\tau}} : \mathbb{R}^{q \times d'_2} \to \mathbb{R}^p$, and $D_{\boldsymbol{\theta}} : \mathbb{R}^{d'_1 \times d'_2} \to \mathbb{R}$, with parameters $\boldsymbol{\omega} \in \boldsymbol{\Omega}, \boldsymbol{\eta} \in \mathcal{H}, \boldsymbol{\tau} \in \mathcal{T}$, and $\boldsymbol{\theta} \in \boldsymbol{\Theta}$, respectively.

The code space is defined as $\boldsymbol{c}_{1:N} := E_{\boldsymbol{\eta}}(\dot{\check{\mathbf{X}}}\dot{\mathbf{Y}}^{\mathrm{Pos}}_{1:N}, \dot{\mathbf{S}}^{\mathrm{Pos}}_{1:N})$, and the generated codes are denoted as $\widetilde{\boldsymbol{c}}_{1:N} := CG_{\boldsymbol{\tau}}(\boldsymbol{\zeta}_{1:N}, \dot{\mathbf{S}}^{\mathrm{Pos}}_{1:N})$, with sub-latent noise $\boldsymbol{\zeta}_{1:N} \sim N(\mathbf{0}, \boldsymbol{I}_q)$. Given latent vectors $\boldsymbol{\xi}_{1:N} \sim N(\mathbf{0}, \boldsymbol{I}_p)$, where $q < p$ and considering $\boldsymbol{\ell}_{1:N} := (\boldsymbol{\xi}_{1:N}, \boldsymbol{c}_{1:N}, \widetilde{\boldsymbol{c}}_{1:N})$, we train CBNP-VAE$\mathcal{C}$GAN by minimizing the composite objective in Eq. 8 over relevant parameter spaces.

Recall the general utility loss function of VAE$\mathcal{C}$GAN in 4.4.1:

$$\mathcal{L}_{\mathrm{Utility}}(G_{\boldsymbol{\omega}}, E_{\boldsymbol{\eta}}, D_{\boldsymbol{\theta}}, CG_{\boldsymbol{\tau}}) = \mathcal{L}(G_{\boldsymbol{\omega}}, E_{\boldsymbol{\eta}} \mid \dot{\mathbf{S}}^{\mathrm{Pos}}) + \mathcal{L}(D_{\boldsymbol{\theta}} \mid \dot{\mathbf{S}}^{\mathrm{Pos}}) + \mathcal{L}(CG_{\boldsymbol{\tau}} \mid \dot{\mathbf{S}}^{\mathrm{Pos}});$$

Loss components are defined as:

$$
\begin{cases}
\mathcal{L}(G_{\boldsymbol{\omega}}, E_{\boldsymbol{\eta}} \mid \dot{\mathbf{S}}^{\mathrm{Pos}}) = \underbrace{-\dfrac{1}{N}\sum_{i=1}^{N} D_{\boldsymbol{\theta}}(G_{\boldsymbol{\omega}}(\boldsymbol{\ell}_i, \dot{\mathbf{S}}^{\mathrm{Pos}}_i), \dot{\mathbf{S}}^{\mathrm{Pos}}_i)}_{\mathcal{I}_{1,1}} + \underbrace{\mathrm{MMD}(F^{\mathrm{Pos}}_{\widetilde{\dot{\check{\mathbf{X}}}\dot{\mathbf{Y}}}_i}_{\dot{\check{\mathbf{X}}}\dot{\mathbf{Y}}^{\mathrm{Pos}}_{1:N}}, F_{G_{\boldsymbol{\omega}}(\boldsymbol{\ell}_{1:N}, \dot{\mathbf{S}}^{\mathrm{Pos}}_{1:N})})}_{\text{Posterior reconstruction error: } \mathcal{I}_{1,2}} \\
\qquad\qquad + \underbrace{\mathrm{MMD}(F_{\boldsymbol{\xi}_{1:N}}, F_{\boldsymbol{c}_{1:N}})}_{\text{Regularization error: } \mathcal{I}_2}, \\[2mm]
\mathcal{L}(D_{\boldsymbol{\theta}} \mid \dot{\mathbf{S}}^{\mathrm{Pos}}) = \underbrace{\dfrac{1}{N}\sum_{i=1}^{N}\Big[D_{\boldsymbol{\theta}}(G_{\boldsymbol{\omega}}(\boldsymbol{\ell}_i, \dot{\mathbf{S}}^{\mathrm{Pos}}_i), \dot{\mathbf{S}}^{\mathrm{Pos}}_i) - 3J^{(a+n)}_{i,N-1} D_{\boldsymbol{\theta}}(\dot{\check{\mathbf{X}}}\dot{\mathbf{Y}}^{\mathrm{Pos}}_{1:N}, \dot{\mathbf{S}}^{\mathrm{Pos}}_i)\Big]}_{\mathcal{J}} + \lambda_{\mathrm{GP}}L_{\mathrm{GP}}, \\[2mm]
\mathcal{L}(CG_{\boldsymbol{\tau}} \mid \dot{\mathbf{S}}^{\mathrm{Pos}}) = \mathrm{MMD}(F_{\boldsymbol{c}_{1:N}}, F_{\widetilde{\boldsymbol{c}}_{1:N}}).
\end{cases}
$$

The terms $\mathcal{I}_{1,1} + \mathcal{I}_{1,2}$ and $\mathcal{J}$ correspond to minimizing the combined distance

$$d_{\mathrm{WMMD}}\big(F^{\mathrm{Pos}}_{\dot{\check{\mathbf{X}}}\dot{\mathbf{Y}}^{\mathrm{Pos}}_{1:N} \mid \dot{\mathbf{S}}^{\mathrm{Pos}}_{1:N}}, F_{G_{\boldsymbol{\omega}}(\boldsymbol{\xi}, \dot{\mathbf{S}}^{\mathrm{Pos}}_{1:N})}\big).$$

The gradient penalty, scaled by $\lambda_{\text{GP}}$, is defined as

$$L_{\text{GP}} = \frac{1}{N} \sum_{i=1}^{N} (\|\nabla_{\widehat{\check{\mathbf{X}}\dot{\mathbf{Y}}}_i^{\text{Pos}}} D_{\boldsymbol{\theta}}(\widehat{\check{\mathbf{X}}\dot{\mathbf{Y}}}_i^{\text{Pos}})\| - 1)^2,$$

and enforces the 1-Lipschitz constraint to stabilize discriminator training, where

$$\widehat{\check{\mathbf{X}}\dot{\mathbf{Y}}}_i^{\text{Pos}} = u\widetilde{\check{\mathbf{X}}\dot{\mathbf{Y}}}_i^{\text{Pos}} + (1-u)\check{\mathbf{X}}\dot{\mathbf{Y}}_i^{\text{Pos}},$$

with $u \in [0, 1]$. See Diagram 21 in Appendix K for the model structure.

## D    COPULA-BASED MODEL

Copula models are widely used to describe multivariate distributions Sklar (1959). A $d$-dimensional copula is a nondecreasing, right-continuous function $C : [0,1]^d \rightarrow [0,1]$ that is grounded, has margins, and is $d$-increasing, making it uniformly continuous on $[0,1]^d$ Nelsen (2006). Sklar's theorem establishes the link between copulas and multivariate distributions:

**Theorem 1** *(Sklar's theorem (Sklar, 1959)) Let $F$ be a $d$-variate distribution function with marginals $F_1, \ldots, F_d$. There exists a d-copula $C$ such that for all $\mathbf{t} \in \overline{\mathbb{R}}^d$, $F(t_1, \ldots, t_d) = C(F_1(t_1), \ldots, F_d(t_d))$.*

*If the marginals are continuous, $C$ is unique and given by $C(\mathbf{u}) = F(F_1^{-1}(u_1), \ldots, F_d^{-1}(u_d))$, where $u_i \in [0, 1]$, $F_i^{-1}(u_i) = \inf\{t \in \mathbb{R} : F_i(t) \geq u_i\}$, and $\overline{\mathbb{R}}^d$ is d product of the extended real line $[-\infty, \infty]$. Otherwise, $C$ is uniquely determined on the range of the marginals.*

Copulas capture the dependence structure of multivariate distributions Joe (1997). A well-known example is the Gaussian copula. Let $\Phi_R$ be the CDF of the multivariate normal distribution with zero mean and correlation matrix $R$. The family of Gaussian copulas is $C_R(\mathbf{u}) = \Phi_R(\Phi^{-1}(u_1), \ldots, \Phi^{-1}(u_d))$, for $\mathbf{u} \in [0,1]^d$. Following Chen et al. Chen et al. (2006), the multivariate distribution $F$ is of a semi-parametric Gaussian copula model with unknown correlation matrix $R$, taking the form

$$F(t_1, \ldots, t_d) = C_R(F_1(t_1), \ldots, F_d(t_d)). \tag{10}$$

## E    DEFINITION OF MMD AND WASSERSTEIN DISTANCES: DIRP AND EMPIRICAL REPRESENTATIONS

Selecting an appropriate statistical distance is crucial for effective generative model training. Here, we focus on the DirP representation of two popular distances used in BNP deep learning and we will briefly mention their frequentist counterparts.

### E.1    MAXIMUM MEAN DISCREPANCY DISTANCE (FEATURE-MATCHING COMPARISON)

The MMD distance was initially introduced in Gretton et al. (2012) for frequentist two-sample comparisons. Recently, a DirP-based version of this distance has been proposed for BNP hypothesis testing Fazeli-Asl et al. (2023). Consider a set of functions $\{G_{\boldsymbol{\omega}}\}_{\boldsymbol{\omega} \in \boldsymbol{\Omega}}$ parameterized by an NN that can generate $n$ IID random variables $(\mathbf{Y}_{1:n})$, where the likelihood function is intractable and not accessible. Given a kernel function $k(\cdot, \cdot)$ satisfying Assumption 1 and a sample $(\boldsymbol{\mathcal{D}}_{1:n}) \overset{\text{IID}}{\sim} F$, the MMD distance between $F^{\text{Pos}}$ and $F_{\mathbf{Y}_{1:n}}$ is approximated as:

$$\text{MMD}^2(F_{\boldsymbol{\mathcal{D}}_{1:N}^{\text{Pos}}}^{\text{Pos}}, F_{\mathbf{Y}_{1:n}}) := \sum_{r,t=1}^{N} J_{r,N-1}^{(a+n)} J_{t,N-1}^{(a+n)} k(\boldsymbol{\mathcal{D}}_r^{\text{Pos}}, \boldsymbol{\mathcal{D}}_t^{\text{Pos}})$$

$$- \frac{2}{n} \sum_{r=1}^{N} \sum_{t=1}^{n} J_{r,N-1}^{(a+n)} k(\boldsymbol{\mathcal{D}}_r^{\text{Pos}}, \mathbf{Y}_t) + \frac{1}{n^2} \sum_{r,t=1}^{n} k(\mathbf{Y}_r, \mathbf{Y}_t). \tag{11}$$

In the frequentist version of the MMD distance, as defined in Gretton et al. (2012), the empirical distribution $F_{\mathcal{D}_{1:n}}$ is considered. This is denoted as $\mathrm{MMD}^2(F_{\mathcal{D}_{1:n}}, F_{\mathbf{Y}_{1:n}})$, which is obtained by replacing $N$, $J_{1:N,N-1}^{(a+n))}$, and $\mathcal{D}_{1:N}^{\mathrm{Pos}}$ with $n$, $1/n$, and $\mathcal{D}_{1:n}$, respectively, in Eq. 11.

**Assumption 1.** The kernel $k(\cdot, \cdot)$ is continuous and defined on a compact metric space $\mathfrak{X}$, with a feature space in a universal reproducing kernel Hilbert space (RKHS). Additionally, $|k(t, t')| < K$ for all $t, t' \in \mathbb{R}^d$.

**Remark 3** *The Radial Basis Function (RBF) kernel, denoted by $k_{\mathrm{RBF}}(\cdot, \cdot)$, satisfies Assumption 1 and is therefore used throughout the implementation in this paper, with the same parameter settings as given in Fazeli-Asl & Zhang (2023).*

### E.2 WASSERSTEIN DISTANCE (OVERALL DISTRIBUTION COMPARISON)

The frequentist version of the Wasserstein distance is completely discussed in (Villani, 2009, Part I6). Fazeli-Asl et al. Fazeli-Asl & Zhang (2023) proposed a BNP version of this distance through its Kantorovich-Rubinstein dual representation. Let $\{D_{\boldsymbol{\theta}}\}_{\boldsymbol{\theta} \in \Theta}$ be a parametrized family of continuous functions that all are 1-Lipschitz. Then the Wasserstein distance between $F^{\mathrm{Pos}}$ and $F_{\mathbf{Y}_{1:n}}$ is approximated as:

$$\mathrm{W}(F_{\mathcal{D}_{1:N}^{\mathrm{Pos}}}^{\mathrm{Pos}}, F_{\mathbf{Y}_{1:n}}) := \max_{\Theta} \left( \sum_{i=1}^{N} J_{i,N-1}^{(a+n)} D_\theta(\mathcal{D}_i^{\mathrm{Pos}}) - \sum_{i=1}^{n} \frac{D_\theta(\mathbf{Y}_i)}{n} \right). \tag{12}$$

By utilizing modifications in Eq. 12 similar to those described for the MMD distance, the empirical representation of the Wasserstein distance can also be obtained.

**Assumption 2.** There exists a constant $M > 0$ such that for any $n, N \in \mathbb{N}$,

$$\max \left( \mathrm{W}(F_{\dot{\mathbf{X}}\dot{\mathbf{Y}}|\dot{\mathbf{S}}}, F_{\dot{\mathbf{X}}\dot{\mathbf{Y}}_{1:n}|\dot{\mathbf{S}}_{1:n}}), \mathrm{W}(F_{\dot{\mathbf{X}}\dot{\mathbf{Y}}_{1:N}^{\mathrm{Pos}}|\dot{\mathbf{S}}_{1:N}^{\mathrm{Pos}}}^{\mathrm{Pos}}, H^{*\prime}), \mathrm{W}(F_{\dot{\mathbf{X}}\dot{\mathbf{Y}}|\dot{\mathbf{S}}}, H^{*\prime}) \right) < M, \tag{13}$$

where $H^{*\prime}$ denotes the conditional $d_1'$-variate CDF derived from the full $d_1' + d_2'$-variate base measure associated with $\mathcal{D}_{1:N}^{\mathrm{Pos}\prime}$, conditioning on the last $d_2'$ marginals. Moreover,

$$\lim_{n \to \infty} \mathrm{W}(F_{\dot{\mathbf{X}}\dot{\mathbf{Y}}|\dot{\mathbf{S}}}, F_{\dot{\mathbf{X}}\dot{\mathbf{Y}}_{1:n}|\dot{\mathbf{S}}_{1:n}}) = \lim_{n,N \to \infty} \mathrm{W}(F_{\dot{\mathbf{X}}\dot{\mathbf{Y}}_{1:N}^{\mathrm{Pos}}|\dot{\mathbf{S}}_{1:N}^{\mathrm{Pos}}}^{\mathrm{Pos}}, H^{*\prime}) = \lim_{n \to \infty} \mathrm{W}(F_{\dot{\mathbf{X}}\dot{\mathbf{Y}}|\dot{\mathbf{S}}}, H^{*\prime}) = 0.$$

## F DIRICHLET MECHANISM: ASSUMPTIONS AND DEFINITION GOHARI ET AL. (2021)

REGULARITY ASSUMPTION

**Assumption 3.** In $\mathbf{\Delta}_{\eta,\bar{\eta}}$, $\eta > 0$, $\bar{\eta} > 0$, and $\eta + \bar{\eta} < 1/2$.

**Assumption 4.** For the Dirichlet mechanism $M(k)_{\mathcal{D}}$, the parameter $k$ satisfies

$$a + n \geq \max \left( \frac{1}{\eta}, \frac{1}{1 - \eta - \bar{\eta}} \right).$$

**Assumption 5.** Fix $W \subseteq [\![N - 1]\!]$, and, then $\gamma \leq \frac{1}{|W|+1}$.

The assumption in Eq. 5 ensures that the argument of the final logarithm term in the global privacy budget $\epsilon_{\mathrm{glo}}$ in Proposition 1 is positive.

**Definition 2** *A Dirichlet mechanism with parameter* $(a + n) \in \mathbb{R}^+$*, denoted by* $\mathcal{M}_{\mathrm{DirP}}^{(a+n)}(p_{1:N}; \epsilon_{glo}, \delta_{glo})$*, outputs* $\mathbf{x} \in \Delta_N$ *according to the Dirichlet probability distribution function centered on* $p_{1:N}$*, i.e.,*

$$\Pr\left[ \mathcal{M}_{\mathrm{DirP}}^{(a+n)}(p_{1:N}; \epsilon_{glo}, \delta_{glo}) = x_{1:N} \right] = \frac{1}{B((a+n)p_{1:N})} \prod_{i=1}^{N-1} x_i^{(a+n)p_i - 1} \left( 1 - \sum_{i=1}^{N-1} x_i \right)^{(a+n)p_N - 1},$$

*where*

$$B((a+n)p_{1:N}) := \frac{\prod_{i=1}^{N} \Gamma((a+n)p_i)}{\Gamma\left((a+n)\sum_{i=1}^{N} p_i\right)},$$

*is the multi-variate beta function.*

## G   THEORETICAL PROOFS

**Proof of Proposition 1** *Gohari et al. introduced a general mechanism that guarantees differential privacy for vectors on the unit simplex. Proposition 1 identifies a practically relevant realization of that mechanism by showing how it naturally arises in the context of resampling within a Bayesian nonparametric framework.*

*Let the fixed dataset be*

$$\mathcal{D} = \{\mathcal{D}_1, \ldots, \mathcal{D}_n\},$$

*and consider two size-$N$ resampled vectors,*

$$\mathcal{D}_{1:N} = (\mathcal{D}_1, \ldots, \mathcal{D}_{i-1}, \mathcal{D}_i, \mathcal{D}_{i+1}, \ldots, \mathcal{D}_{j-1}, \mathcal{D}_j, \mathcal{D}_{j+1}, \ldots, \mathcal{D}_N),$$
$$\mathcal{D}'_{1:N} = (\mathcal{D}_1, \ldots, \mathcal{D}_{i-1}, \mathcal{D}_i, \mathcal{D}_{i+1}, \ldots, \mathcal{D}_{j-1}, \mathcal{D}_i, \mathcal{D}_{j+1}, \ldots, \mathcal{D}_N),$$

*where in $\mathcal{D}'_{1:N}$ the observation $\mathcal{D}_i$ appears twice, while $\mathcal{D}_j$ $(j \neq i)$ is omitted.*

*The corresponding empirical distributions are*

$$F_{\mathcal{D}_{1:N}} = \sum_{\ell=1}^{N} p_\ell \, \mathbb{1}_{\{\mathcal{D}_\ell\}}, \qquad F_{\mathcal{D}'_{1:N}} = \sum_{\ell=1}^{N} p'_\ell \, \mathbb{1}_{\{\mathcal{D}_\ell\}},$$

*where the weights satisfy*

$$p_\ell = \tfrac{1}{N} \text{ for all } \ell \in [\![N]\!], \qquad p'_i = \tfrac{2}{N}, \quad p'_j = 0, \quad p'_\ell = \tfrac{1}{N} \text{ for } \ell \notin \{i,j\}.$$

*These two empirical distributions are defined to be neighbors, as they differ in exactly one resampled observation. For $N \geq 2/b$, the corresponding weight vectors satisfy*

$$\|p - p'\|_1 = |p_i - p'_i| + |p_j - p'_j| = \left|\tfrac{1}{N} - \tfrac{2}{N}\right| + \left|\tfrac{1}{N} - 0\right| = \frac{2}{N} \leq b,$$

*which matches the b-adjacency condition required by the Dirichlet mechanism in Gohari et al. (2021).*

*Therefore, the randomized weights in equation 3 coincide exactly with the Dirichlet mechanism of Gohari et al. (2021) when applied to the classical resampling weights. This establishes a direct theoretical connection between classical resampling and the DirP resampling scheme in equation 3. In particular, the additional uncertainty introduced by the randomized weights constitutes a differentially private perturbation of standard resampling, with an explicit privacy budget and formal privacy guarantees.*

*The remaining argument follows directly from (Gohari et al., 2021, Theorem 1) and is therefore omitted.* □

**Proof of Corollary 1** *Recall that $\epsilon_{glo}$ in Proposition 1 is given by*

$$\epsilon_{glo} = \underbrace{\ln\left(\frac{B\left((a+n)\eta, (a+n)(1-\bar{\eta}-\eta)\right)}{B\left((a+n)(\eta+\frac{b}{2}), (a+n)(1-\bar{\eta}-\eta-\frac{b}{2})\right)}\right)}_{\mathcal{I}_1(a)} + \underbrace{\frac{b(a+n)}{2}}_{\mathcal{I}_2(a)} \ln\left(\frac{1-(|W|-1)\gamma}{\gamma}\right).$$

*To prove (i), we utilize the Beta-Gamma relation to rewrite $\mathcal{I}_1(a)$ as*

$$\mathcal{I}_1(a) = \underbrace{\ln\left(\frac{\Gamma((a+n)\eta)}{\Gamma((a+n)\eta + \mathcal{I}_2(a))}\right)}_{\mathcal{I}_{1,1}(a)} + \underbrace{\ln\left(\frac{\Gamma((a+n)(1-\bar{\eta}-\eta))}{\Gamma((a+n)(1-\bar{\eta}-\eta) - \mathcal{I}_2(a))}\right)}_{\mathcal{I}_{1,2}(a)}.$$

*Applying the fundamental theorem of calculus, we obtain*

$$-\mathcal{I}_{1,1}(a) = \ln\Gamma\left((a+n)\eta + \mathcal{I}_2(a)\right) - \ln\Gamma\left((a+n)\eta\right)$$
$$= \int_{(a+n)\eta}^{(a+n)\eta+\mathcal{I}_2(a)} \psi(t)\,dt,$$

*where $\psi(t) = \frac{d}{dt}\ln\Gamma(t)$ is the Digamma function.*

*Since $\psi(\cdot)$ is continuous on $[(a+n)\eta, (a+n)\eta + \mathcal{I}_2(a)]$, the mean value theorem for integrals ensures the existence of $t^* \in [(a+n)\eta, (a+n)\eta + \mathcal{I}_2(a)]$ such that*

$$-\mathcal{I}_{1,1}(a) = \psi(t^*)\mathcal{I}_2(a).$$

*For sufficiently large $a$, we approximate*

$$\psi\left((a+n)\eta + \mathcal{I}_2(a)\right) \approx \ln\left((a+n)\eta + \mathcal{I}_2(a)\right) - \frac{1}{2(a+n)\eta + \mathcal{I}_2(a)} \tag{14}$$

$$= \ln\left((a+n)\eta\right) + \ln\left(1 + \frac{1}{2a^2\eta}\right) - \frac{1}{2(a+n)\eta + \mathcal{I}_2(a)} \tag{15}$$

$$\approx \ln\left((a+n)\eta\right) \pm \frac{1}{(a+n)\eta} + \frac{1}{2a^2\eta} - \frac{1}{2(a+n)\eta + \mathcal{I}_2(a)} \tag{16}$$

$$= \psi\left((a+n)\eta\right) + \underbrace{\frac{1}{(a+n)\eta} + \frac{1}{2a^2\eta} - \frac{1}{2(a+n)\eta + \mathcal{I}_2(a)}}_{h(a)}. \tag{17}$$

*Eq. 14 follows from the asymptotic expansion of the Digamma function. The value of $b$ is chosen based on the cutoff $a = 1$, and to analyze the asymptotic behavior of $\epsilon_{glo}$ as $a \to \infty$, we consider $a \geq 1$, leading to the substitution $b = a^{-2}$ in $\mathcal{I}_2(a)$ across all equations. Eq. 15 is derived by factorizing $(a+n)\eta$ inside the logarithms and applying the logarithm product rule. Eq. 16 uses the approximation $\ln(1+x) \approx x$ for small $x$, obtained from the Maclaurin series expansion. Finally, using the asymptotic expansion of the Digamma function again concludes Eq. 17.*

*Considering Eq. 14-Eq. 17 and the continuity of the Digamma function, we obtain*

$$\psi\left((a+n)\eta\right)\mathcal{I}_2(a) \leq -\mathcal{I}_{1,1}(a) \leq \psi\left((a+n)\eta\right)\mathcal{I}_2(a) + h(a)\mathcal{I}_2(a). \tag{18}$$

*Since $\lim_{a\to\infty} h(a)\mathcal{I}_2(a) = 0$, letting $a \to \infty$ in all terms of inequality in Eq. 18 yields*

$$\lim_{a\to\infty} -\mathcal{I}_{1,1}(a) = \lim_{a\to\infty} \psi\left((a+n)\eta\right)\mathcal{I}_2(a). \tag{19}$$

*Applying the asymptotic expansion of $\psi(\cdot)$ in Eq. 19 and simplifying, we obtain*

$$\lim_{a\to\infty} -\mathcal{I}_{1,1}(a) \approx \lim_{a\to\infty} \mathcal{I}_2(a)\ln\left((a+n)\eta\right) - \lim_{a\to\infty}\frac{1}{2a^2\eta}$$
$$\overset{\text{H}}{=} \lim_{a\to\infty}\frac{a+n}{2a^2+4n} - \lim_{a\to\infty}\frac{1}{2a^2\eta} \qquad\qquad \text{(Applying L'Hôpital's rule)}$$
$$= 0.$$

*Similarly, we find that $\lim_{a\to\infty}\mathcal{I}_{1,2}(a) = 0$. Since it is evident that $\lim_{a\to\infty}\mathcal{I}_2(a) = 0$, the proof of (i) is complete.*

*To establish (ii), we first demonstrate that*

$$\left(J_{1,N-1}^{(a+n)}, \ldots, J_{N,N-1}^{(a+n)}\right) \overset{p}{\to} \left(\frac{1}{N}, \ldots, \frac{1}{N}\right) \quad as \quad a \to \infty.$$

*Since the expectation under $F^{Pos}$ satisfies $E_{F^{Pos}}(J_{i,N-1}^{(a+n)}) = \frac{1}{N}$ for all $i \in \{1, \ldots, N\}$, we apply Chebyshev's inequality:*

$$\Pr\left(\left|J_{i,N^{-1}}^{(a+n)} - \frac{1}{N}\right| \geq \epsilon\right) \leq \frac{\mathrm{Var}_{F^{Pos}}(J_{i,N^{-1}}^{(a+n)})}{\epsilon^2}, \tag{20}$$

*for any $\epsilon > 0$. Substituting $\mathrm{Var}_{F^{Pos}}(J_{i,N^{-1}}^{(a+n)}) = \frac{N-1}{N^2(a+n+1)}$ into Eq. 20, we obtain*

$$\Pr\left(\left|J_{i,N^{-1}}^{(a+n)} - \frac{1}{N}\right| \geq \epsilon\right) \leq \frac{N-1}{N^2(a+n+1)\epsilon^2}. \tag{21}$$

*Taking the limit as $a \to \infty$ in Eq. 21 implies that*

$$J_{i,N^{-1}}^{(a+n)} \xrightarrow{p} \frac{1}{N}.$$

*Since convergence in probability implies convergence in distribution, it follows that*

$$\left(J_{1,N^{-1}}^{(a+n)}, \ldots, J_{N,N^{-1}}^{(a+n)}\right) \xrightarrow{d} \left(\frac{1}{N}, \ldots, \frac{1}{N}\right) \quad as\ a \to \infty.$$

*By the definition of convergence in distribution, for any Borel set $A \subset \Delta$ whose boundary (denoted by $\partial A$) satisfies*

$$\Pr\left(\left(\frac{1}{N}, \ldots, \frac{1}{N}\right) \in \partial A\right) = 0, \tag{22}$$

*it holds that*

$$\lim_{a \to \infty} \Pr\left(\left(J_{1,N^{-1}}^{(a+n)}, \ldots, J_{N,N^{-1}}^{(a+n)}\right) \in A\right) = \Pr\left(\left(\frac{1}{N}, \ldots, \frac{1}{N}\right) \in A\right).$$

*To clarify the boundary condition, note that $\partial A$ is the boundary of the set $A$, defined as*

$$\partial A = \bar{A} \cap \overline{A^c},$$

*where $\bar{A}$ is the closure of $A$ and $\overline{A^c}$ is the closure of the complement of $A$. The boundary consists of points that can be approached both from inside and outside of $A$. The condition in Eq. 22 ensures that the limiting measure does not concentrate on the boundary of $A$. This requirement is necessary for the definition of convergence in distribution to hold, as convergence in distribution is defined in terms of the measure of sets where the limit point lies away from the boundary.*

*Since the limiting distribution is degenerate at $\left(\frac{1}{N}, \ldots, \frac{1}{N}\right)$, the probability measure is concentrated at this single point, meaning the distribution becomes deterministic in the limit. Therefore, the boundary condition is trivially satisfied. Consequently, the convergence holds for any Borel set $A \subset \Delta$.*

*Furthermore, since $\mathbf{\Omega}_\gamma \subset \Delta$ and, by Assumption 5, $\gamma \leq \frac{1}{N}$, the set $\mathbf{\Omega}_\gamma$ is a Borel set whose boundary does not contain the limiting point $\left(\frac{1}{N}, \ldots, \frac{1}{N}\right)$ with positive probability. Therefore, the convergence equation holds for $\mathbf{\Omega}_\gamma$, implying that*

$$\Pr\left(\left(\frac{1}{N}, \ldots, \frac{1}{N}\right) \in \Omega_\gamma\right) = 1.$$

*Finally, the continuity of the $\min$ function completes the proof.* $\qquad\square$

**Proof of Proposition 2** *Given $H^{\mathrm{Pert}}$ and $H$ modeled by a semi-Gaussian copula with the estimated correlation matrix $\hat{R}$, for $\mathbf{t} \in \overline{\mathbb{R}}^d$ with $d = N_{\mathrm{C}} + N_{\mathrm{D}}$, we have*

$$\left| H^{\mathrm{Pert}} - H \right| = \left| C_{\hat{R}}(F_{\mathrm{AGM},1}(t_1), \ldots, F_{\mathrm{AGM},N_{\mathrm{C}}}(t_{N_{\mathrm{C}}}), F_{\mathrm{RRM},1}(t_{N_{\mathrm{C}}+1}), \ldots, F_{\mathrm{RRM},N_{\mathrm{D}}}(t_d)) \right.$$

$$\left. - C_{\hat{R}}(H_1^{(\mathrm{C})}(t_1), \ldots, H_{N_{\mathrm{C}}}^{(\mathrm{C})}(t_{N_{\mathrm{C}}}), H_1^{(\mathrm{D})}(t_{N_{\mathrm{C}}+1}), \ldots, H_{N_{\mathrm{D}}}^{(\mathrm{D})}(t_d)) \right|$$

$$\overset{1-\mathrm{Lipschitz}}{\leq} \sum_{i_{\mathrm{C}}=1}^{N_{\mathrm{C}}} \left| F_{\mathrm{AGM},i_{\mathrm{C}}}(t_{i_{\mathrm{C}}}) - H_{i_{\mathrm{C}}}^{(\mathrm{C})}(t_{i_{\mathrm{C}}}) \right| + \sum_{i_{\mathrm{D}}=1}^{N_{\mathrm{D}}} \left| F_{\mathrm{RRM},i_{\mathrm{D}}}(t_{N_{\mathrm{C}}+i_{\mathrm{D}}}) - H_{i_{\mathrm{D}}}^{(\mathrm{D})}(t_{N_{\mathrm{C}}+i_{\mathrm{D}}}) \right|$$

$$= \sum_{i_{\mathrm{C}} \in I^{(\mathrm{C})}} \left| F_{\mathrm{AGM},i_{\mathrm{C}}}(t_{i_{\mathrm{C}}}) - H_{i_{\mathrm{C}}}^{(\mathrm{C})}(t_{i_{\mathrm{C}}}) \right| + \sum_{i_{\mathrm{D}} \in I^{(\mathrm{D})}} \left| F_{\mathrm{RRM},i_{\mathrm{D}}}(t_{N_{\mathrm{C}}+i_{\mathrm{D}}}) - H_{i_{\mathrm{D}}}^{(\mathrm{D})}(t_{N_{\mathrm{C}}+i_{\mathrm{D}}}) \right|$$

$$+ \sum_{i_{\mathrm{C}} \in [\![N_{\mathrm{C}}]\!] \setminus I^{(\mathrm{C})}} \left| F_{\mathrm{AGM},i_{\mathrm{C}}}(t_{i_{\mathrm{C}}}) - H_{i_{\mathrm{C}}}^{(\mathrm{C})}(t_{i_{\mathrm{C}}}) \right| + \sum_{i_{\mathrm{D}} \in [\![N_{\mathrm{D}}]\!] \setminus I^{(\mathrm{D})}} \left| F_{\mathrm{RRM},i_{\mathrm{D}}}(t_{N_{\mathrm{C}}+i_{\mathrm{D}}}) - H_{i_{\mathrm{D}}}^{(\mathrm{D})}(t_{N_{\mathrm{C}}+i_{\mathrm{D}}}) \right|$$

*The last inequality follows from the 1-Lipschitz property of copula functions, which ensures that the difference between two copula values is bounded by the sum of the differences between their marginal inputs.*

*Given $\delta_{loc_{i_{\mathrm{C}}}}^{(\mathrm{C})} = 0$, $F_{\mathrm{AGM},i_{\mathrm{C}}}$ corresponds to the CDF of an $\epsilon$-differential privacy mechanism. For two differential privacy mechanisms, AGM and RRM, we have*

$$F_{\mathrm{AGM},i_{\mathrm{C}}} \overset{d}{\to} H_{i_{\mathrm{C}}}^{(\mathrm{C})}(t_{i_{\mathrm{C}}}) \quad and \quad F_{\mathrm{RRM},i_{\mathrm{D}}} \overset{d}{\to} H_{i_{\mathrm{D}}}^{(\mathrm{D})}(t_{i_{\mathrm{D}}})$$

*as $\epsilon_{loc_{i_{\mathrm{C}}}}^{(\mathrm{C})} \to \infty, \epsilon_{loc_{i_{\mathrm{D}}}}^{(\mathrm{D})} \to \infty$ since increasing the privacy budget reduces the added noise, causing the perturbed marginals to converge to the original CDFs. This completes the proof.* $\square$

The following theorem establishes the asymptotic guarantees of the proposed framework. It shows that, under mild regularity conditions, the generator converges in probability to the true conditional distribution in the utility-only case and jointly achieves utility, fairness, and balance in the full-loss setting. This result provides a rigorous theoretical foundation for the method and highlights its ability to simultaneously address multiple objectives.

**Theorem 2 (Consistency and Joint Guarantees)** *Given choosing $\triangle(\cdot, \cdot)$ as $d_{\mathrm{WMMD}}(\cdot, \cdot)$ in Eq. 7, define*
$$M_{d_{\mathrm{WMMD}}}(\mathbb{R}^d) := \left( \mathcal{P}(\mathbb{R}^d), d_{\mathrm{WMMD}} \right).$$
*Under assumptions in Eq. 1–Eq. 2, as $n, N \to \infty$:*

*(i) If $\lambda_{\mathrm{F}} = \lambda_{\mathrm{B}_{i_{\mathrm{D}}}} = 0$ for $i_{\mathrm{D}} \in [\![N_{\mathrm{D}}-2]\!]$, and $\boldsymbol{\omega}^{\circledcirc}$ minimizes the utility-only loss from Eq. 8, then in the well-specified case where*
$$d_{\mathrm{WMMD}}\left( F_{\mathring{\mathbf{X}}\dot{\mathbf{Y}}|\dot{\mathbf{S}}}, F_{G_{\boldsymbol{\omega}^{\circ}}} \right) = 0 \quad for \ some \ true \ minimizer \ \boldsymbol{\omega}^{\circ},$$
*we have*
$$F_{G_{\boldsymbol{\omega}^{\circledcirc}}} \overset{\mathbb{P}}{\to} F_{\mathring{\mathbf{X}}\dot{\mathbf{Y}}|\dot{\mathbf{S}}} \qquad \text{(perfect utility)},$$
*where $\mathring{\mathbf{X}}\dot{\mathbf{Y}} := (\mathring{\mathbf{X}}, \dot{\mathbf{Y}})$.*

*(ii) If $\lambda_{\mathrm{F}} = \lambda_{\mathrm{B}_{i_{\mathrm{D}}}} = 1$ and $\boldsymbol{\omega}^{\odot}$ minimizes the full loss in Eq. 9, then in the well-specified case where all target discrepancies are zero for some $\boldsymbol{\omega}^{\bullet}$,*
$$F_{G_{\boldsymbol{\omega}^{\odot}}} \overset{p}{\to} F_{\mathring{\mathbf{X}}\dot{\mathbf{Y}}|\dot{\mathbf{S}}} \quad \text{(utility)},$$
$$\mathrm{MI}\left( \left( G_{\boldsymbol{\omega}^{\odot}}(\boldsymbol{\ell}, \dot{\mathbf{S}}) \right)_{(C_{N_{\mathrm{D}}-2}:C_{N_{\mathrm{D}}-2}^+)}, \dot{\mathbf{S}} \right) \overset{d}{\to} 0 \quad \text{(fairness)},$$
$$\mathrm{D_{KL}}\left( \left( G_{\boldsymbol{\omega}^{\odot}}(\boldsymbol{\ell}, \dot{\mathbf{S}}) \right)_{(C_{i_{\mathrm{D}}}:C_{i_{\mathrm{D}}}^+)}, \dot{U}_{i_{\mathrm{D}}} \right) \overset{d}{\to} 0 \quad \text{(balance)},$$
*where*
$$C_{i_{\mathrm{D}}} = N_{\mathrm{C}} + i_{\mathrm{D}}, \qquad C_{i_{\mathrm{D}}}^+ = C_{i_{\mathrm{D}}} + K_{i_{\mathrm{D}}},$$

denote the column indices $(C_{i_{\mathrm{D}}} : C_{i_{\mathrm{D}}}^+)$ in the generator output corresponding to the one-hot–encoded representation of the $i_{\mathrm{D}}$-th unprotected attribute in the original data. This block spans $K_{i_{\mathrm{D}}} = C_{i_{\mathrm{D}}}^+ - C_{i_{\mathrm{D}}}$ columns, each being $0$ except for a single $1$ indicating the active category.

**Proof of Theorem 2** When $\lambda_{\mathrm{F}} = \lambda_{\mathrm{B}_{i_{\mathrm{D}}}} = 0$, the loss function in Eq. 9 reduces to the utility loss $\mathcal{L}_{\mathrm{Utility}}$ defined in Eq. 8. In this case, the optimized value $\boldsymbol{\omega}^{\circledcirc}$ is determined solely by minimizing $\mathcal{L}_{\mathrm{Utility}}$, as the fairness and balance constraints have no effect. For such $\boldsymbol{\omega}^{\circledcirc}$ and $\epsilon > 0$, Markov's inequality implies

$$\Pr\left(d_{\mathrm{WMMD}}(F_{\check{\mathbf{X}}\dot{\mathbf{Y}}|\dot{\mathbf{S}}}, F_{G_{\boldsymbol{\omega}^{\circledcirc}}}) \geq \epsilon\right) \leq \frac{\mathbb{E}\left(d_{\mathrm{WMMD}}(F_{\check{\mathbf{X}}\dot{\mathbf{Y}}|\dot{\mathbf{S}}}, F_{G_{\boldsymbol{\omega}^{\circledcirc}}})\right)}{\epsilon}. \tag{23}$$

The triangular inequality implies:

$$d_{\mathrm{WMMD}}(F_{\check{\mathbf{X}}\dot{\mathbf{Y}}|\dot{\mathbf{S}}}, F_{G_{\boldsymbol{\omega}^{\circledcirc}}}) \leq d_{\mathrm{WMMD}}(F_{\check{\mathbf{X}}\dot{\mathbf{Y}}|\dot{\mathbf{S}}}, F_{\boldsymbol{\mathcal{D}}_{1:N}^{\mathrm{Pos}}}^{\mathrm{Pos}}) + d_{\mathrm{WMMD}}(F_{\boldsymbol{\mathcal{D}}_{1:N}^{\mathrm{Pos}}}^{\mathrm{Pos}}, F_{G_{\boldsymbol{\omega}^{\circledcirc}}}) = \mathcal{I}_1. \tag{24}$$

For any $\omega \in \Omega$, within the context of minimizing $\mathcal{L}_{Utility}$, applying the triangular inequality yields:

$$
\begin{aligned}
d_{\mathrm{WMMD}}(F_{\check{\mathbf{X}}\dot{\mathbf{Y}}|\dot{\mathbf{S}}}, F_{G_{\boldsymbol{\omega}^{\circledcirc}}}) &\leq d_{\mathrm{WMMD}}(F_{\check{\mathbf{X}}\dot{\mathbf{Y}}|\dot{\mathbf{S}}}, F_{\boldsymbol{\mathcal{D}}_{1:N}^{\mathrm{Pos}}}^{\mathrm{Pos}}) + d_{\mathrm{WMMD}}(F_{\boldsymbol{\mathcal{D}}_{1:N}^{\mathrm{Pos}}}^{\mathrm{Pos}}, F_{G_{\boldsymbol{\omega}^{\circledcirc}}}) \quad \text{(Triangle inequality)}\\
&\leq d_{\mathrm{WMMD}}(F_{\check{\mathbf{X}}\dot{\mathbf{Y}}|\dot{\mathbf{S}}}, F_{\boldsymbol{\mathcal{D}}_{1:N}^{\mathrm{Pos}}}^{\mathrm{Pos}}) + d_{\mathrm{WMMD}}(F_{\boldsymbol{\mathcal{D}}_{1:N}^{\mathrm{Pos}}}^{\mathrm{Pos}}, F_{G_{\boldsymbol{\omega}}}) \quad \text{(Definition of } \boldsymbol{\omega}^{\circledcirc})\\
&\leq d_{\mathrm{WMMD}}(F_{\check{\mathbf{X}}\dot{\mathbf{Y}}|\dot{\mathbf{S}}}, F_{\boldsymbol{\mathcal{D}}_{1:N}^{\mathrm{Pos}}}^{\mathrm{Pos}}) + d_{\mathrm{WMMD}}(F_{G_{\boldsymbol{\omega}}}, F_{\check{\mathbf{X}}\dot{\mathbf{Y}}|\dot{\mathbf{S}}})\\
&\qquad + d_{\mathrm{WMMD}}(F_{\check{\mathbf{X}}\dot{\mathbf{Y}}|\dot{\mathbf{S}}}, F_{\boldsymbol{\mathcal{D}}_{1:N}^{\mathrm{Pos}}}^{\mathrm{Pos}}) \quad \text{(Triangle inequality)}\\
&\leq 2\left[d_{\mathrm{WMMD}}(F_{\check{\mathbf{X}}\dot{\mathbf{Y}}|\dot{\mathbf{S}}}, F_{\check{\mathbf{X}}\dot{\mathbf{Y}}_{1:n}|\dot{\mathbf{S}}_{1:n}}) + d_{\mathrm{WMMD}}(F_{\check{\mathbf{X}}\dot{\mathbf{Y}}_{1:n}|\dot{\mathbf{S}}_{1:n}}, F_{\boldsymbol{\mathcal{D}}_{1:N}^{\mathrm{Pos}}}^{\mathrm{Pos}})\right]\\
&\qquad + d_{\mathrm{WMMD}}(F_{G_{\boldsymbol{\omega}}}, F_{\check{\mathbf{X}}\dot{\mathbf{Y}}|\dot{\mathbf{S}}}) \quad \text{(Triangle inequality)}\\
&\leq 2\big[d_{\mathrm{WMMD}}(F_{\check{\mathbf{X}}\dot{\mathbf{Y}}|\dot{\mathbf{S}}}, F_{\check{\mathbf{X}}\dot{\mathbf{Y}}_{1:n}|\dot{\mathbf{S}}_{1:n}}) + d_{\mathrm{WMMD}}(F_{\check{\mathbf{X}}\dot{\mathbf{Y}}_{1:n}|\dot{\mathbf{S}}_{1:n}}, H^{*\prime})\\
&\qquad + d_{\mathrm{WMMD}}(H^{*\prime}, F^{\mathrm{Pos}})\big] + d_{\mathrm{WMMD}}(F_{G_{\boldsymbol{\omega}}}, F_{\check{\mathbf{X}}\dot{\mathbf{Y}}|\dot{\mathbf{S}}})\\
&\qquad\qquad\qquad\qquad\qquad\qquad\qquad\qquad \text{(Triangle inequality)}\\
&= 2\underbrace{\big[d_{\mathrm{MMD}}(F_{\check{\mathbf{X}}\dot{\mathbf{Y}}|\dot{\mathbf{S}}}, F_{\check{\mathbf{X}}\dot{\mathbf{Y}}_{1:n}|\dot{\mathbf{S}}_{1:n}}) + d_{\mathrm{MMD}}(F_{\check{\mathbf{X}}\dot{\mathbf{Y}}_{1:n}|\dot{\mathbf{S}}_{1:n}}, H^{*\prime})}_{}\\
&\qquad\qquad\qquad\qquad\qquad\qquad \underbrace{+ d_{\mathrm{MMD}}(H^{*\prime}, F^{\mathrm{Pos}})\big]}_{\mathcal{I}}\\
&\qquad + 2\big[d_{\mathrm{W}}(F_{\check{\mathbf{X}}\dot{\mathbf{Y}}|\dot{\mathbf{S}}}, F_{\check{\mathbf{X}}\dot{\mathbf{Y}}_{1:n}|\dot{\mathbf{S}}_{1:n}}) + d_{\mathrm{W}}(F_{\check{\mathbf{X}}\dot{\mathbf{Y}}_{1:n}|\dot{\mathbf{S}}_{1:n}}, H^{*\prime})\\
&\qquad\qquad\qquad\qquad\qquad\qquad + d_{\mathrm{W}}(H^{*\prime}, F^{\mathrm{Pos}})\big]\\
&\qquad\qquad\qquad\qquad\qquad\qquad\qquad + d_{\mathrm{WMMD}}(F_{G_{\boldsymbol{\omega}}}, F_{\check{\mathbf{X}}\dot{\mathbf{Y}}|\dot{\mathbf{S}}})\\
&\qquad\qquad\qquad\qquad\qquad\qquad\qquad\qquad \text{(Definition of WMMD)}
\end{aligned}
\tag{25}
$$

Since the inequality in Eq. 25 holds for all $\boldsymbol{\omega} \in \Omega$, it follows that

$$
\begin{aligned}
d_{\mathrm{WMMD}}(F_{\check{\mathbf{X}}\dot{\mathbf{Y}}|\dot{\mathbf{S}}}, F_{G_{\boldsymbol{\omega}^{\circledcirc}}}) &\leq 2\big[d_{\mathrm{MMD}}(F_{\check{\mathbf{X}}\dot{\mathbf{Y}}|\dot{\mathbf{S}}}, F_{\check{\mathbf{X}}\dot{\mathbf{Y}}_{1:n}|\dot{\mathbf{S}}_{1:n}}) + d_{\mathrm{MMD}}(F_{\check{\mathbf{X}}\dot{\mathbf{Y}}_{1:n}|\dot{\mathbf{S}}_{1:n}}, H^{*\prime})\\
&\qquad + d_{\mathrm{MMD}}(H^{*\prime}, F^{\mathrm{Pos}})\big] + 2\mathcal{I} + \min_{\boldsymbol{\omega} \in \Omega} d_{\mathrm{WMMD}}(F_{G_{\boldsymbol{\omega}}}, F_{\check{\mathbf{X}}\dot{\mathbf{Y}}|\dot{\mathbf{S}}}). \tag{26}
\end{aligned}
$$

Taking the expectation of both sides of Eq. 26, we get:

$$
\mathbb{E}\left(d_{\text{WMMD}}(F_{\check{\mathbf{X}}\dot{\mathbf{Y}}|\dot{\mathbf{S}}}, F_{G_{\boldsymbol{\omega}^\circledcirc}})\right) = \mathbb{E}_{F_{\check{\mathbf{X}}\dot{\mathbf{Y}}|\dot{\mathbf{S}}}}\left(\mathbb{E}_{F^{\text{Pos}}}\left(d_{\text{WMMD}}(F_{\check{\mathbf{X}}\dot{\mathbf{Y}}|\dot{\mathbf{S}}}, F_{G_{\boldsymbol{\omega}^\circledcirc}})\right)\right)
$$

$$
\leq 2\left[2\mathbb{E}_{\check{\mathbf{X}}\dot{\mathbf{Y}}|\dot{\mathbf{S}}}\left(\text{W}(F_{\check{\mathbf{X}}\dot{\mathbf{Y}}|\dot{\mathbf{S}}}, F_{\check{\mathbf{X}}\dot{\mathbf{Y}}_{1:n}|\dot{\mathbf{S}}_{1:n}})\right) + 2\mathbb{E}_{F_{\check{\mathbf{X}}\dot{\mathbf{Y}}|\dot{\mathbf{S}}}}\left(\mathbb{E}_{F^{\text{Pos}}}\left(\text{W}(F_{\check{\mathbf{X}}\dot{\mathbf{Y}}|\dot{\mathbf{S}}}, H^{*\prime})\right)\right)\right.
$$

$$
\left. + \mathbb{E}_{F_{\check{\mathbf{X}}\dot{\mathbf{Y}}|\dot{\mathbf{S}}}}\left(\mathbb{E}_{F^{\text{Pos}}}\left(\text{W}(F_{\boldsymbol{\mathcal{D}}^{\text{Pos}}_{1:N}}^{\text{Pos}}, F_{\check{\mathbf{X}}\dot{\mathbf{Y}}|\dot{\mathbf{S}}})\right)\right)\right] \tag{27a}
$$

$$
+ \frac{2K}{\sqrt{n}} + \frac{4aK}{a+n} + 2\sqrt{\frac{(a+n+N)K}{(a+n+1)N}} \tag{27b}
$$

$$
+ d_{\text{WMMD}}(F_{G_{\boldsymbol{\omega}^\circ}}, F_{\check{\mathbf{X}}\dot{\mathbf{Y}}|\dot{\mathbf{S}}}), \tag{27c}
$$

where Eq. 27b refers to $2\mathbb{E}(\mathcal{I})$, directly derived from the proof of (Fazeli-Asl et al., 2023, Lemma 4), and Eq. 27c is due to assuming $\omega^\circ$ is the true parameter minimizing $d_{\text{WMMD}}(F_{G_{\boldsymbol{\omega}}}, F_{\check{\mathbf{X}}\dot{\mathbf{Y}}|\dot{\mathbf{S}}})$.

To complete the proof of (i), we need to analyze the behavior of the expressions in Eq. 27a and Eq. 27b as $n$ and $N$ approach infinity. Considering Assumption 2 and then applying the dominated convergence theorem, we conclude that all expectations in Eq. 27a converge to zero as $n, N \to \infty$. Finally, taking the limit as $n$ and $N$ approach infinity on both sides of Eq. 27 yields:

$$
\lim_{n,N\to\infty} \mathbb{E}\left(d_{\text{WMMD}}(F_{\check{\mathbf{X}}\dot{\mathbf{Y}}|\dot{\mathbf{S}}}, F_{G_{\boldsymbol{\omega}^\circledcirc}})\right) \leq d_{\text{WMMD}}(F_{G_{\boldsymbol{\omega}^\circ}}, F_{\check{\mathbf{X}}\dot{\mathbf{Y}}|\dot{\mathbf{S}}}),
$$

which establishes the utility guarantee in the well-specified case.

For proving (ii), when $\lambda_{\text{F}} = \lambda_{\text{B}_{i_{\text{D}}}} = 1$, the objective in Eq. 9 becomes the full loss function:

$$
\mathcal{L}_{\text{Utility}} + \mathcal{L}_{\text{Fair}} + \sum_{i_{\text{D}}=1}^{N_{\text{D}}} \mathcal{L}_{\text{Balance}_{i_{\text{D}}}}, \tag{28}
$$

where the learned parameter $\boldsymbol{\omega}^\circledcirc$ minimizes all three terms simultaneously. In this case, the well-specified scenario corresponds to the setting where the optimal minimizer $\boldsymbol{\omega}^\bullet$ satisfies the following:

$$
\begin{cases}
d_{\text{WMMD}}(F_{\check{\mathbf{X}}\dot{\mathbf{Y}}|\dot{\mathbf{S}}}, F_{G_{\boldsymbol{\omega}^\bullet}}) = 0 & \text{(29a)} \\[2mm]
\text{MI}((G_{\boldsymbol{\omega}^\bullet}(\boldsymbol{\ell}, \dot{\mathbf{S}}))_{(C_{N_{\text{D}}-2}:C^+_{N_{\text{D}}-2})}, \dot{\mathbf{S}}) = 0 & \text{(29b)} \\[2mm]
\text{D}_{\text{KL}}((G_{\boldsymbol{\omega}^\bullet}(\boldsymbol{\ell}, \dot{\mathbf{S}}))_{(C_{i_{\text{D}}}:C^+_{i_{\text{D}}})}, U_{i_{\text{D}}}) = 0 & \text{(29c)}
\end{cases}
$$

As a result, the utility guarantee

$$
d_{\text{WMMD}}(F_{G_{\boldsymbol{\omega}^\circledcirc}}, F_{\check{\mathbf{X}}\dot{\mathbf{Y}}|\dot{\mathbf{S}}}) \xrightarrow{p} 0, \tag{30}
$$

directly follows from Eq. 29a, similar to the reasoning in part (i), now in the context of the full loss formulation Eq. 28.

To establish the fairness guarantee, we begin with the following inequality:

$$\left| \text{MI}((G_{\boldsymbol{\omega}\odot}(\boldsymbol{\ell}, \dot{\mathbf{S}}))_{(C_{N_{\mathrm{D}}-2}:C^+_{N_{\mathrm{D}}-2})}, \dot{\mathbf{S}}) - \text{MI}((G_{\boldsymbol{\omega}\bullet}(\boldsymbol{\ell}, \dot{\mathbf{S}}))_{(C_{N_{\mathrm{D}}-2}:C^+_{N_{\mathrm{D}}-2})}, \dot{\mathbf{S}}) \right|$$

$$\leq \left| \text{MI}((G_{\boldsymbol{\omega}\odot}(\boldsymbol{\ell}, \dot{\mathbf{S}}))_{(C_{N_{\mathrm{D}}-2}:C^+_{N_{\mathrm{D}}-2})}, \dot{\mathbf{S}}) - \text{MI}^{\text{DV}}((G_{\boldsymbol{\omega}\bullet}(\boldsymbol{\ell}, \dot{\mathbf{S}}))_{(C_{N_{\mathrm{D}}-2}:C^+_{N_{\mathrm{D}}-2})}, \dot{\mathbf{S}}) \right|$$

$$+ \left| \text{MI}^{\text{DV}}((G_{\boldsymbol{\omega}\bullet}(\boldsymbol{\ell}, \dot{\mathbf{S}}^{)}))_{(C_{N_{\mathrm{D}}-2}:C^+_{N_{\mathrm{D}}-2})}, \dot{\mathbf{S}}^{)} - \text{MI}((G_{\boldsymbol{\omega}\bullet}(\boldsymbol{\ell}, \dot{\mathbf{S}}^{)}))_{(C_{N_{\mathrm{D}}-2}:C^+_{N_{\mathrm{D}}-2})}, \dot{\mathbf{S}}^{)} \right|$$

$$\leq \left| \text{MI}((G_{\boldsymbol{\omega}\odot}(\boldsymbol{\ell}, \dot{\mathbf{S}}))_{(C_{N_{\mathrm{D}}-2}:C^+_{N_{\mathrm{D}}-2})}, \dot{\mathbf{S}}) - \text{MI}^{\text{DV}}((G_{\boldsymbol{\omega}\odot}(\boldsymbol{\ell}, \dot{\mathbf{S}}))_{(C_{N_{\mathrm{D}}-2}:C^+_{N_{\mathrm{D}}-2})}, \dot{\mathbf{S}}) \right|$$

$$+ \left| \text{MI}^{\text{DV}}((G_{\boldsymbol{\omega}\odot}(\boldsymbol{\ell}, \dot{\mathbf{S}}))_{(C_{N_{\mathrm{D}}-2}:C^+_{N_{\mathrm{D}}-2})}, \dot{\mathbf{S}}) - \text{MI}^{\text{DV}}((G_{\boldsymbol{\omega}\bullet}(\boldsymbol{\ell}, \dot{\mathbf{S}}))_{(C_{N_{\mathrm{D}}-2}:C^+_{N_{\mathrm{D}}-2})}, \dot{\mathbf{S}}) \right|$$

$$+ \left| \text{MI}^{\text{DV}}((G_{\boldsymbol{\omega}\bullet}(\boldsymbol{\ell}, \dot{\mathbf{S}}))_{(C_{N_{\mathrm{D}}-2}:C^+_{N_{\mathrm{D}}-2})}, \dot{\mathbf{S}}) - \text{MI}((G_{\boldsymbol{\omega}\bullet}(\boldsymbol{\ell}, \dot{\mathbf{S}}))_{(C_{N_{\mathrm{D}}-2}:C^+_{N_{\mathrm{D}}-2})}, \dot{\mathbf{S}}) \right|$$

$$\leq \left| \text{MI}^{\text{DPDV}}((G_{\boldsymbol{\omega}\odot}(\boldsymbol{\ell}_{1:N}, \dot{\mathbf{S}}^{\text{Pos}}_{1:N}))_{(C_{N_{\mathrm{D}}-2}:C^+_{N_{\mathrm{D}}-2})}, \dot{\mathbf{S}}^{\text{Pos}}_{1:N}) \right.$$
$$\left. - \text{MI}^{\text{DV}}((G_{\boldsymbol{\omega}\odot}(\boldsymbol{\ell}, \dot{\mathbf{S}}))_{(C_{N_{\mathrm{D}}-2}:C^+_{N_{\mathrm{D}}-2})}, \dot{\mathbf{S}}) \right| \tag{31a}$$

$$+ \left| \text{MI}((G_{\boldsymbol{\omega}\odot}(\boldsymbol{\ell}, \dot{\mathbf{S}}))_{(C_{N_{\mathrm{D}}-2}:C^+_{N_{\mathrm{D}}-2})}, \dot{\mathbf{S}}) \right.$$
$$\left. - \text{MI}^{\text{DirPDV}}((G_{\boldsymbol{\omega}\odot}(\boldsymbol{\ell}_{1:N}, \dot{\mathbf{S}}^{\text{Pos}}_{1:N}))_{(C_{N_{\mathrm{D}}-2}:C^+_{N_{\mathrm{D}}-2})}, \dot{\mathbf{S}}^{\text{Pos}}_{1:N}) \right| \tag{31b}$$

$$+ \left| \text{MI}^{\text{DV}}((G_{\boldsymbol{\omega}\odot}(\boldsymbol{\ell}, \mathbf{dotS}))_{(C_{N_{\mathrm{D}}-2}:C^+_{N_{\mathrm{D}}-2})}, \mathbf{dotS}) - \text{MI}^{\text{DV}}((G_{\boldsymbol{\omega}\bullet}(\boldsymbol{\ell}, \dot{\mathbf{S}}))_{(C_{N_{\mathrm{D}}-2}:C^+_{N_{\mathrm{D}}-2})}, \dot{\mathbf{S}}) \right| \tag{31c}$$

$$+ \left| \text{MI}^{\text{DV}}((G_{\boldsymbol{\omega}\bullet}(\boldsymbol{\ell}, \dot{\mathbf{S}}))_{(C_{N_{\mathrm{D}}-2}:C^+_{N_{\mathrm{D}}-2})}, \dot{\mathbf{S}}) - \text{MI}((G_{\boldsymbol{\omega}\bullet}(\boldsymbol{\ell}, \dot{\mathbf{S}}))_{(C_{N_{\mathrm{D}}-2}:C^+_{N_{\mathrm{D}}-2})}, \dot{\mathbf{S}}) \right| \tag{31d}$$

To conclude the proof it is enough to assess the asymptotical behavior of the right hand side of the inequality in Eq. 31. Applying (Fazeli-Asl et al., 2026, Theorem 5(i,ii)) yields:

$$Eq.\ 31a \xrightarrow{a.s.} 0, Eq.\ 31b \xrightarrow{a.s.} 0, \quad \text{as,} \quad n, N \to \infty. \tag{32}$$

To handle Eq. 31c, consider triangle inequality:

$$| d_{\text{WMMD}}(F_{G_{\boldsymbol{\omega}\odot}}, F_{\boldsymbol{\omega}\bullet}) | \leq \underbrace{| d_{\text{WMMD}}(F_{G_{\boldsymbol{\omega}\odot}}, F_{\check{\mathbf{X}}\dot{\mathbf{Y}}|\dot{\mathbf{S}}}) |}_{\mathcal{J}_1} + \underbrace{| d_{\text{WMMD}}(F_{\check{\mathbf{X}}\dot{\mathbf{Y}}|\dot{\mathbf{S}}}, F_{G_{\boldsymbol{\omega}\bullet}}) |}_{\mathcal{J}_2},$$

By Eq. 30 and Eq. 29a, we have $\mathcal{J}_1 \xrightarrow{p} 0$ and $\mathcal{J}_2 = 0$, hence $F_{G_{\boldsymbol{\omega}\odot}} \xrightarrow{p} F_{\boldsymbol{\omega}\bullet}$ as $n, N \to \infty$, which further implies that $G_{\boldsymbol{\omega}\odot} \xrightarrow{d} G_{\boldsymbol{\omega}\bullet}$. Applying the continuous mapping theorem:

$$T_{\boldsymbol{\gamma}}(G_{\boldsymbol{\omega}\odot}(\boldsymbol{\ell}, \dot{\mathbf{S}}))_{(C_{N_{\mathrm{D}}-2}:C^+_{N_{\mathrm{D}}-2})}, \mathbf{S}') \xrightarrow{d} T_{\boldsymbol{\gamma}}((G_{\boldsymbol{\omega}\bullet}(\boldsymbol{\ell}, \dot{\mathbf{S}}))_{(C_{N_{\mathrm{D}}-2}:C^+_{N_{\mathrm{D}}-2})}, \dot{\mathbf{S}}),$$

and again via continuity of the $\max(\cdot)$ function:

$$Eq.\ 31c \xrightarrow{d} 0. \tag{33}$$

Additionally, by applying Belghazi et al. (Belghazi et al., 2018, Lemma 1), for any $\epsilon > 0$, there exists a set of neural networks $\{T_{\boldsymbol{\gamma}}\}_{\boldsymbol{\gamma} \in \boldsymbol{\Gamma}}$ on some compact domain $\boldsymbol{\Gamma}$ such that:

$$Eq.\ 31d < \epsilon, \text{ a.s..} \tag{34}$$

Finally, combining Eq. 32, Eq. 33, and Eq. 34, then applying Eq. 29b within Eq. 34, completes the proof of the fairness guarantee.

The proof of the balance guarantee follows similarly, as it also relies on the DirPDV representation of the KL divergence, and is thus omitted. □

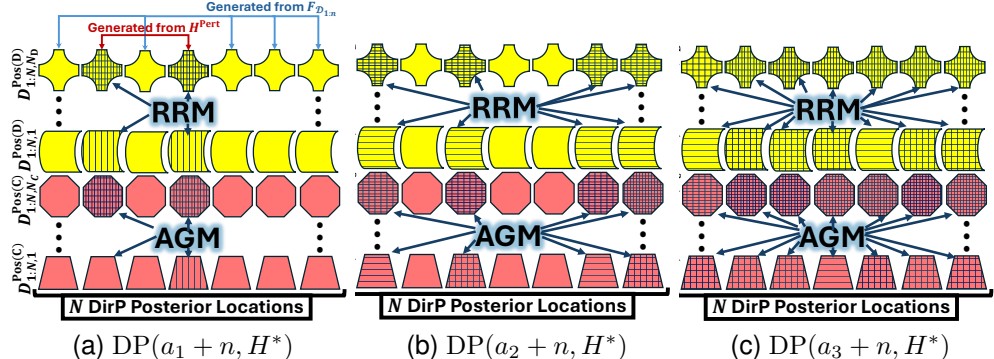

Figure 5: Privacy scheme by DirP posterior sampling with $a_1 < a_2 \ll a_3$. Denser patterns indicate stronger privacy budgets chosen by the operator.

## H FURTHER DISCUSSION ON PRIVACY PRESERVING:

We clarify that the impact of marginal perturbation on data utility is modulated by two key factors: the localized privacy budget and the concentration parameter $a$ in the DirP. Specifically, the posterior base measure is given by:

$$H^* = \frac{a}{a+n} H^{\mathrm{Pert}} + \frac{n}{a+n} F_{\mathcal{D}_{1:n}},$$

where $n$ is the sample size and will be replaced with mini-batch size $n_{mb}$ when implementing Algorithm 2. A larger $a$ increases the influence of the perturbed base measure, enforcing stronger local privacy but potentially reducing utility, while a smaller $a$ shifts weight toward the data, improving utility but reducing local privacy protection. See Figure 5 for visualization. We also note that $a$ directly determines the global privacy budget, as shown in Proposition 1. Thus, higher values of $a$ simultaneously enhance both local and global privacy guarantees.

Eventually, our numerical results highlight the importance of supporting flexible marginal structures and demonstrate that the BNP formulation offers a principled mechanism for balancing privacy and utility. The framework introduces a novel privacy amplification effect, which is beneficial for privacy (Steinke, 2022, Section 6). In this setting, DirP posterior samples are drawn from a mixture of clean samples and perturbed samples generated from $H^{\mathrm{Pert}}$. In contrast, traditional privacy amplification operates by subsampling the data and then applying a Differential Privacy mechanism that perturbs only the selected subset. The proposed framework retains all samples while injecting perturbed samples through posterior sampling, thereby achieving amplification without discarding data.discarding data.

## I ADDITIONAL EXPERIMENTS

All experiments were conducted using PyTorch Lightning with a mini-batch size of 256, running for 1,000 or 1,500 training epochs on the NVIDIA GPU described in Table 11. Further details of the model configurations and dataset-specific training settings are provided in Table 12. This table uses a consistent training sample size across datasets to ensure that running times remain comparable across different dimensional settings. We also note that the runtime of a one-dimensional (1D) convolutional neural network (CNN) differs fundamentally from that of a two-dimensional (2D) CNN due to architectural and computational differences.

Table 11: Hardware specifications of the GPU server used in all experiments.

| | |
|---|---|
| GPU Model | NVIDIA A100-SXM4-40GB |
| CUDA Version | 12.8 |
| Total GPU Memory | 40,960 MiB |

Table 12: Convolutional Neural Network (CNN) architectures and training configurations across all datasets Based on 10,000 Training Samples (equal sample sizes) with Mini-Batch Size 256.

| Dataset | Adult | ProPublica COMPAS | Bank Marketing | Toy Example (SLV) | MNIST |
|---|---|---|---|---|---|
| # of Dimension | 15 | 18 | 21 | 4 | $28 \times 28$ |
| Runtime | 50 min | 50 min | 50 min | 50 min | 400 min |
| Epoch | 1000 | 1000 | 1000 | 1000 | 300 |
| Type of CNN | 1D | 1D | 1D | 1D | 2D |
| # of CNN's parameters | 1.8M | 1.8M | 1.8M | 1.8M | 302M |

## I.1 SIMULATED TOY EXAMPLE BASED ON A SHARED LATENT VARIABLE (SLV)

For a comprehensive evaluation of the proposed method, we begin this section by constructing a four-dimensional toy example designed to analyse the method's behaviour under isolated mechanisms.

### I.1.1 SIMULATING DATA CONSTRUCTION

We simulate data in which one continuous covariate, one multicategory covariate, a binary outcome, and a binary sensitive attribute are all correlated through a shared latent structure. This controlled setup provides a clean environment for examining the factors of interest discussed in this paper, particularly in fairness-related scenarios where a dependency exists between the outcome and the sensitive attribute.

Let

$$Z_1, \ldots, Z_n \overset{\text{i.i.d.}}{\sim} N(0, 1)$$

denote a latent variable that governs the dependence among all observable coordinates. For each observation $i \in [\![n]\!]$, the four attributes

$$(X_{1i}, X_{2i}, Y_i, S_i)$$

are constructed as functions of the shared latent factor $Z_i$, ensuring that the components remain mutually dependent while retaining distinct marginal distributions.

The continuous covariate $X_{1i} \in \mathbb{R}$ is constructed through a smooth nonlinear transformation of the latent factor,

$$X_{1i} = \tanh(Z_i),$$

which produces a bounded continuous attribute while preserving a strictly monotone relationship with the latent variable.

The categorical covariate $X_{2i}$ with three categories is obtained by thresholding the latent variable as

$$X_{2i} = \begin{cases} 1, & Z_i \leq -0.3, \\ 2, & -0.3 < Z_i \leq 0.6, \\ 3, & Z_i > 0.6, \end{cases}$$

which induces an ordinal categorical variable whose distribution reflects the ordering of the latent variable.

The binary outcome $Y_i$ is obtained using a logistic link:

$$\Pr(Y_i = 2 \mid Z_i) = \frac{1}{1 + e^{-2.5Z_i}},$$

so that the probability of assigning category 2 increases sharply as the latent variable $Z_i$ becomes larger.

The sensitive attribute $S_i$ is designed to exhibit very strong dependence with the outcome. After sampling $Y_i$, the sensitive variable is defined by

$$S_i = \begin{cases} Y_i, & \text{with probability } 0.95, \\ 3 - Y_i, & \text{with probability } 0.05. \end{cases}$$

This maintains a realistic binary marginal distribution while ensuring that the mutual dependency between $Y_i$ and $S_i$ is large.

The final toy dataset

$$\{(X_{1i}, X_{2i}, Y_i, S_i)\}_{i=1}^n, \qquad n = 10000,$$

provides a continuous attribute, a multicategory attribute, and two binary attributes that are all coupled through a single latent factor.

In the following subsections, we use these simulated datasets with a mini-batch size of 256 (i.e., setting $n_{\text{mb}} = 256$ in Algorithm 2) as the training data to evaluate the performance of the proposed approach through three below sections

### I.1.2 ISOLATING PRIVACY

We first isolate the privacy component of our conditional generative approach by setting $\lambda_F = 0$ (no fairness) and $\lambda_B = 0$ (no class balance)[2], and by using $n_{\text{iter}} = 1500$ in Algorithm 2. This allows us to evaluate how effectively the proposed method privatizes the dataset. For this purpose, we employ membership-inference attacks (Shokri et al., 2017).

A membership-inference attacker aims to determine whether a specific sample was used in training a classifier, referred to here as the reference model. Given an input data point, the attacker queries the trained reference model and observes its prediction confidence. Reference models typically assign higher confidence to samples seen during training (members) compared to unseen samples (non-members). The attacker exploits this discrepancy to infer membership. In our setting, the members are the synthetic samples generated by the conditional generative model, which are treated as the training dataset for the reference model during the privacy-evaluation phase.

Differential privacy mechanisms are designed to make the reference model's behavior insensitive to the inclusion or exclusion of any single individual. If the mechanism is effective, the reference model's outputs on members and non-members should be statistically indistinguishable. Thus, membership-inference attacks serve as a practical privacy evaluation tool: failure of the attack indicates reduced training-data leakage, whereas success indicates that membership information is still exposed.

We report the effect of our privacy mechanism using two standard evaluation measures, AUC and test accuracy, with definitions following Fawcett (2006). Together, these two measures characterize the privacy–utility trade-off and reveal how the privacy mechanism influences both leakage risk and downstream predictive performance.

- **Attack AUC.** The membership-inference attack produces a confidence score for each sample. When the reference model leaks information, the score distributions of members and non-members differ. The Area Under the ROC Curve (AUC) quantifies the attacker's ability to distinguish these two groups. A high AUC indicates strong privacy leakage, whereas an AUC near 0.5 indicates that members and non-members are indistinguishable. Thus, the AUC directly reflects the privacy provided by the mechanism and captures the trade-off between utility and privacy. For example, across each local privacy budget in Figure 6(a)–(e), increasing the hyperparameter $a$ strengthens privacy while reducing utility. We also note that, for a fixed value of $a$, a stronger local privacy budget results in higher privacy.

- **Test accuracy.** Test accuracy measures the predictive performance of the reference model on a clean, unseen test set. It reflects the utility of the model after privacy noise is applied. As stronger noise is introduced (smaller $\varepsilon$), the training data become less informative, often reducing test accuracy. Thus, test accuracy captures the utility cost of the privacy mechanism.

---

[2]In this example, for simplicity of notation, we write $\lambda_B$ to refer to the coefficient of the KL loss, given by Section 4.4.3, associated with both categorical covariates, $X_{2i}$ and $Y_i$.

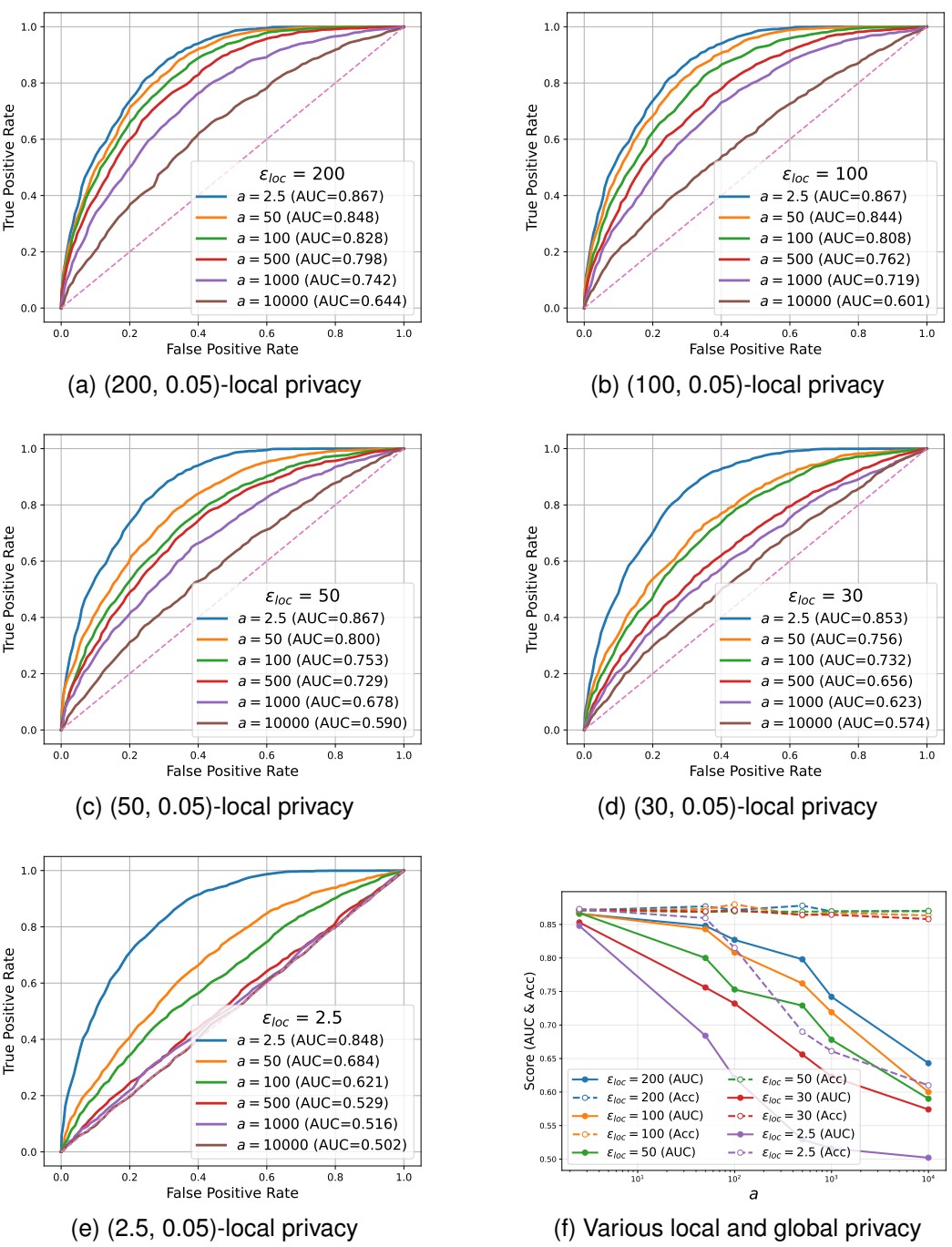

Figure 6: Membership-Inference Attack ROC. AUC values approaching 0.5 indicate stronger privacy.

Figure 6(f) shows accuracy results across different local privacy budgets and confirms the effect of increasing $a$ in reducing utility. We also draw attention to Figure 6(a), which illustrates the role of global privacy in the weakly localized privacy regime (i.e., almost no local noise), demonstrating that the generated data can still achieve meaningful privacy protection while preserving strong utility, as supported by the corresponding accuracy results in Figure 6(f).

To complement the ROC-based evaluation, we also visualize the distributions of prediction confidences for members and non-members using the histograms in Figure 7. Because a membership-inference attacker exploits differences in the reference model's prediction confidence, these histograms provide an intuitive view of the privacy–utility trade-off.

When the model leaks membership information, the histogram for member points shifts to the right, indicating consistently higher confidence, while the histogram for non-members shifts left. Conversely, when the privacy mechanism is effective, the two histograms overlap substantially, and both distributions move left, indicating that the reference model assigns similarly low confidence levels to members and non-members. In this regime, membership inference approaches random guessing, with attack AUC trending toward $0.50$. Thus, the histogram offers a direct visual summary of how the proposed mechanism trades privacy protection against predictive utility.

### I.1.3 ISOLATING PRIVACY AND FAIRNESS

To understand how global privacy simultaneously affects fairness and utility, Figure 8 illustrates the variation of mutual information (MI) against the variation of MMD across different values of $a$, all with the same $(2.5, 0.05)$ local privacy budget and with $95\%$ confidence intervals. These results are obtained using the full optimization schedule of Algorithm 2 with $\lambda_{\mathrm{B}} = 0$, comparing $\lambda_F = 1$ against $\lambda_F = 0$, and using $n_{\mathrm{iter}} = 1500$ total iterations with $n_{\mathrm{fair}} = 800$ fairness-optimization steps. Together, these results form a privacy–fairness–utility trade-off curve.

Figure 8(a) shows that, in the absence of an explicit fairness constraint, increasing $a$ raises the MMD (indicating reduced utility), while the MI remains close to the true MI. This demonstrates that stronger privacy distorts the utility of the generated samples but leaves the MI largely unchanged. This behaviour is expected due to the copula-based construction of $H^{\mathrm{Pert}}$, where the correlation parameter of the Gaussian copula is estimated from the original dataset. As $a$ increases and the posterior base measure $H^*$ converges to $H^{\mathrm{Pert}}$, the conditional generator produces samples whose distribution approaches that of $H^{\mathrm{Pert}}$. Consequently, the MI between the generated outcomes and the sensitive attributes approaches the MI obtained from samples drawn directly from $H^{\mathrm{Pert}}$ (denoted as "Prior MI" in the plot). The slight discrepancy between the true MI and the Prior MI arises from estimation error.

Figure 8(b) shows that, although utility decreases as $a$ increases, the MI between the sensitive attributes and the generated outcomes remains zero when the model is trained with the fairness constraint.

Finally, to further explore whether privacy creates any inherent tension with fairness in the optimization process, we examine how the MI evolves across different epochs through providing Figure 9. Consider the model when it begins by training only for utility during the initial 700 epochs under different values of $a$. As discussed earlier, increasing $a$ causes the MI to approach the Prior MI, which explains why, at the end of utility training, the mean MI estimate is noticeably different from the true value (the navy tilde-pattern line in Figure 9). The MI values are then tracked over the remaining 800 epochs ($n_{\mathrm{iter}} = 1500$, $n_{\mathrm{fair}} = 800$ in Algorithm 2). During the first phase, the model naturally learns the unfair dependence present in the perturbed data. Once the fairness constraint is applied, this dependence is removed across all privacy levels, confirming that fairness can still be achieved even when the data have been perturbed, as already concluded from Figure 8(b).

However, the amount of perturbation affects the behaviour in fairness optimization. Stronger privacy leads to slower MI reduction and causes the MI values to vary more during training. In other words, higher privacy does not prevent fairness, but it makes convergence in fairness optimization slower.

### I.1.4 ISOLATING CLASS BALANCE AND FAIRNESS

As a final investigation, we consider two isolated settings in the absence of privacy ($a = 10^{-6}$):

(i) **Isolating class balance** by setting $\lambda_{\mathrm{B}} = 1$ and $\lambda_{\mathrm{F}} = 0$ (no fairness). This experiment examines whether enforcing class balance affects the underlying relationship between the sensitive attributes and the generated outcomes. To this end, the model is first trained solely for utility during the initial 700 epochs, after which the class-balance constraint is activated for the remaining 300 epochs ($n_{\mathrm{iter}} = 1000$, $n_{\mathrm{balance}} = 300$ in Algorithm 2), while utility updates continue throughout all 1,000 epochs. We record the MI and SP at each epoch

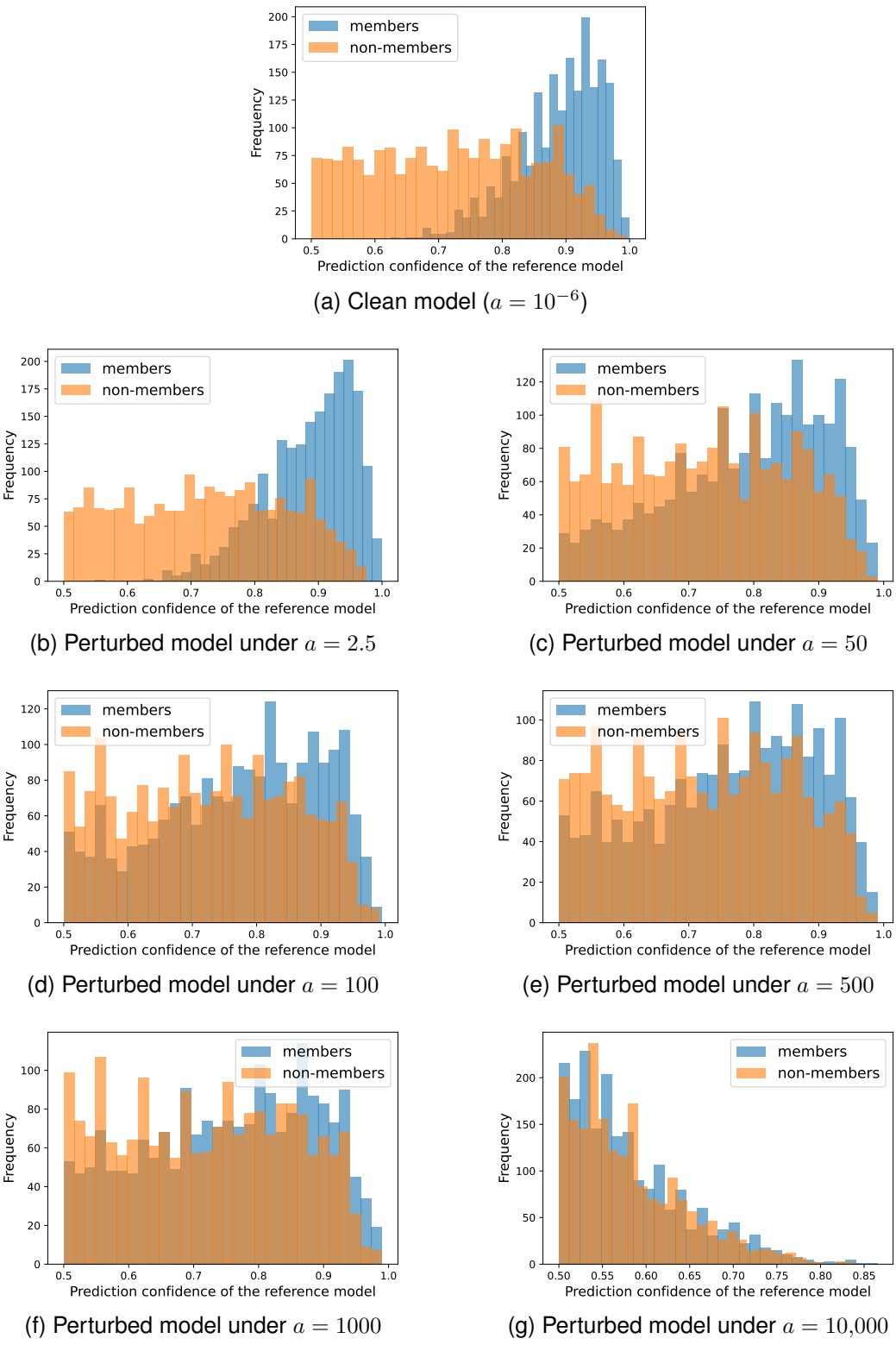

Figure 7: Histograms of prediction confidences for members versus non-members under different choices of $a$, all with the same $(2.5, 0.05)$-local privacy budget.

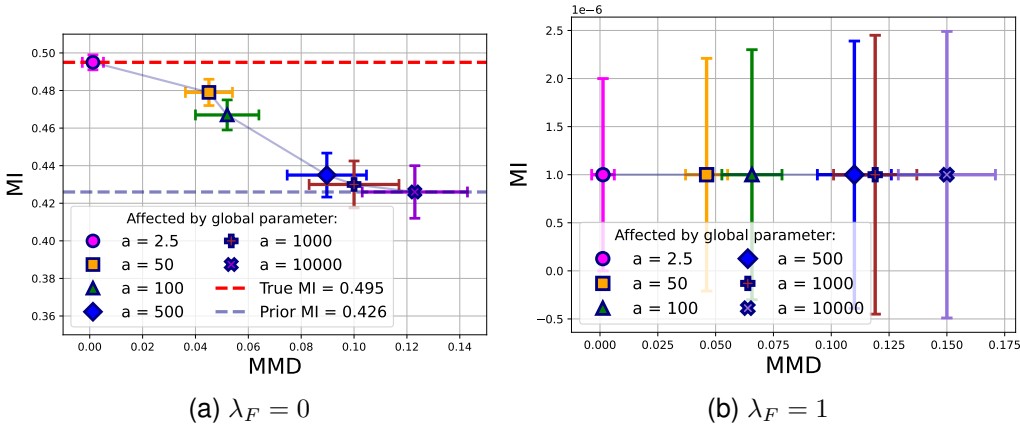

Figure 8: Privacy–fairness–utility trade-off under different choices of $a$, all with the same $(2.5, 0.05)$-local privacy budget.

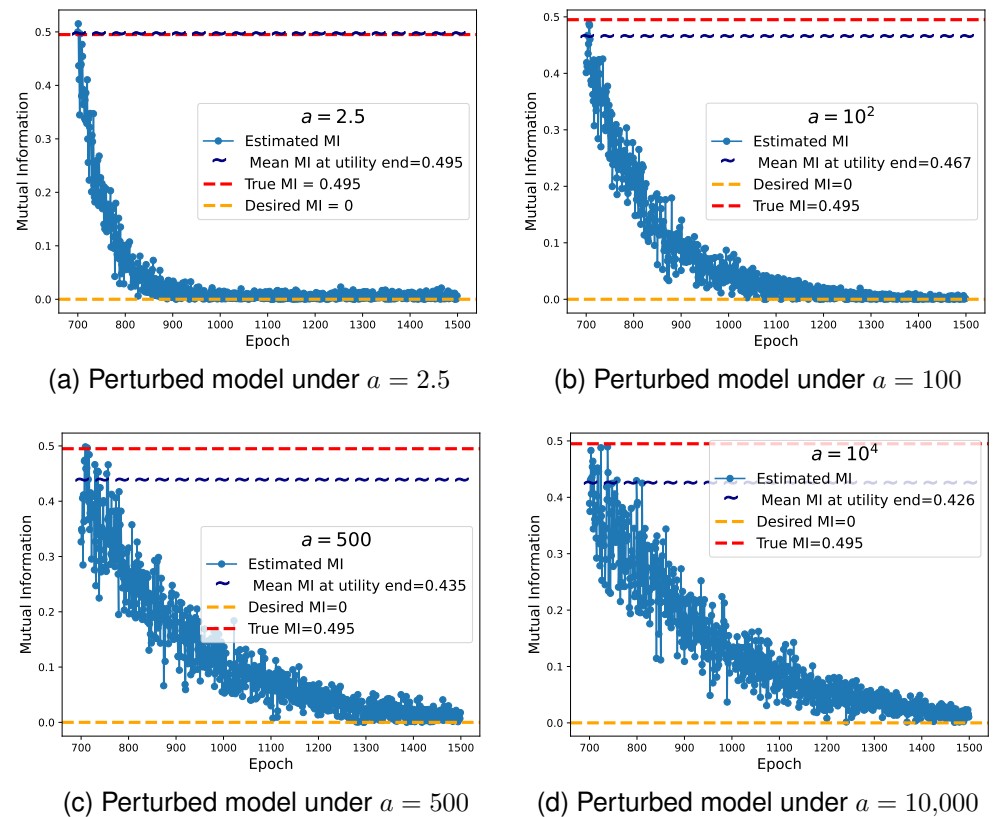

Figure 9: Isolated effects of privacy on fairness optimization ($\lambda_F = 1$) under different choices of $a$, all with the same $(2.5, 0.05)$-local privacy budget.

during these final 300 epochs. As shown in Figure 10(a), the MI and SP remain concentrated around the corresponding true values learned during the first 700 epochs of utility training, indicating that applying class balance alone does not alter the underlying unfairness present in the generated data.

(ii) **Isolating fairness** by setting $\lambda_{\mathrm{F}} = 1$ and $\lambda_{\mathrm{B}} = 0$ (no class balance). Figure 10(b) illustrates the effect of enforcing fairness when the model is first trained for utility during the initial 700 epochs, and the fairness constraint is activated only in the remaining 300 epochs ($n_{\mathrm{iter}} = 1000$, $n_{\mathrm{fair}} = 300$ in Algorithm 2). In contrast to the class-balance-only setting, this experiment highlights the central role of the fairness constraint in removing the dependence between the sensitive attributes and the generated outcomes. After the first 700 epochs, during which the model learns the underlying unfair relationship present in the training data, the fairness constraint effectively eliminates this dependence, even in the absence of class-balance adjustments.

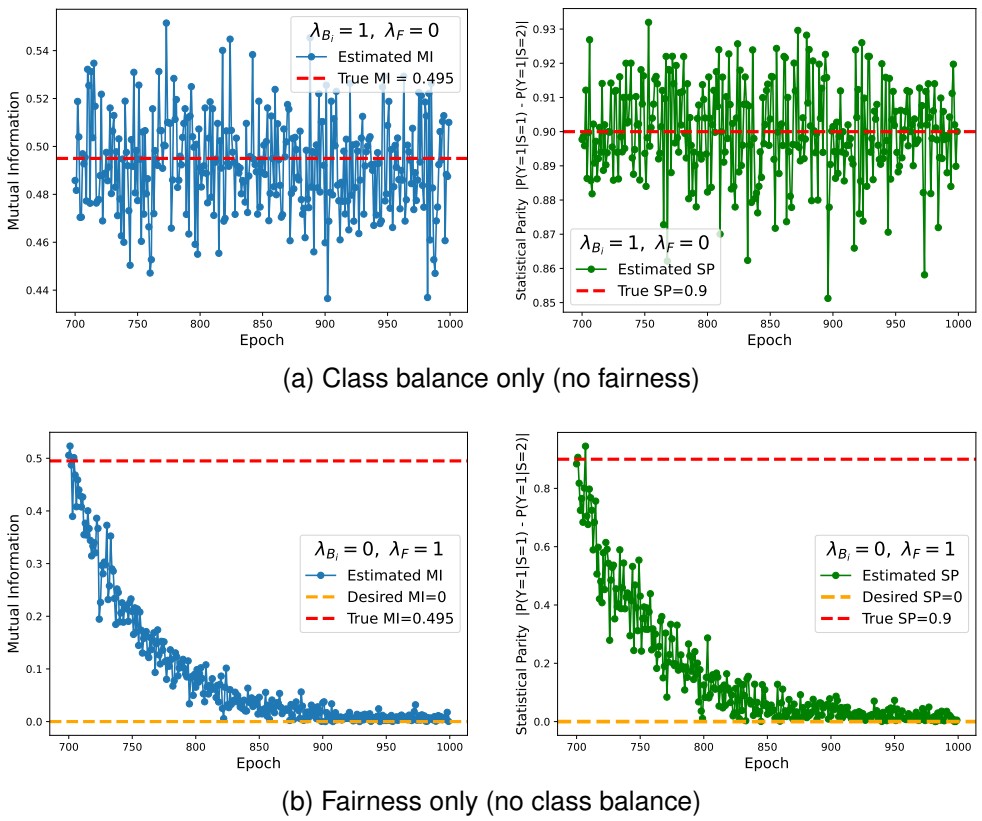

(a) Class balance only (no fairness)

(b) Fairness only (no class balance)

Figure 10: Isolated effects of class balance and fairness on MI and SP.

## I.2  MNIST TOY EXAMPLE WITH MAXIMAL DEPENDENCE.

The MNIST dataset consists of 60,000 handwritten digit images spanning 10 classes (0–9), with each image represented as a $28 \times 28$ pixel grid (784 dimensions) (LeCun, 1998). As an additional illustrative toy experiment, the conditional generative model is trained on the MNIST dataset, where the class label is used as the conditioning attribute. In this setting, the image vectors are treated as the outcomes $\mathbf{Y}$ and the corresponding digit labels as $\mathbf{S}$. In the baseline configuration, the generator is trained so that $\mathbf{S}$ matches $\mathbf{Y}$ with probability 1, creating an extreme case of maximal dependence between the outcomes and the digit attributes. In this configuration, the model simply takes digits $\mathbf{S}$ and produces the corresponding images; see Figure 11.

The proposed fairness mechanism is then applied to gradually reduce this dependence, with the goal of driving the MI between $\mathbf{S}$ and the generated outcomes $\mathbf{Y}$ (i.e., $\widetilde{\mathbf{Y}}$) toward zero. In this setting, the generator receives the digit labels $\mathbf{S}$ but is expected to produce images that are independent of $\mathbf{S}$. Figure 12 shows synthetic outputs for the fixed input sequence $\mathbf{S} = (2, 0, 3, 5, 8, 7, 5, 4, 6, 7, 1, 9)$ under different values of $a$. As expected, there is almost no meaningful correspondence between the inputs and outputs, indicating that the fairness mechanism successfully enforces independence

between $\mathbf{S}$ and $\widetilde{\mathbf{Y}}$. Moreover, for each fixed local privacy budget (isolating the effect of global privacy), increasing $a$ increases the proportion of perturbed outputs. Conversely, when $a$ is fixed (isolating the effect of local privacy), the severity of the distortions becomes more pronounced as the global privacy budget grows. This example illustrates how small values of $a$ can preserve good utility even under a strict local privacy budget, whereas larger values of $a$ induce increasingly strong perturbations.

It is emphasised that this experiment is not intended to represent a realistic fairness scenario, because digit labels do not constitute a meaningful or interpretable sensitive attribute. **This is precisely the reason it is referred to as a toy example.** Its purpose is to illustrate that the proposed method can eliminate even a perfectly enforced dependence between $\mathbf{S}$ and generated $\mathbf{Y}$ and to provide intuition about the behaviour of the local and global privacy mechanisms. This toy example is simply the visual realisation of the posterior sampling diagram shown in Figure 5.

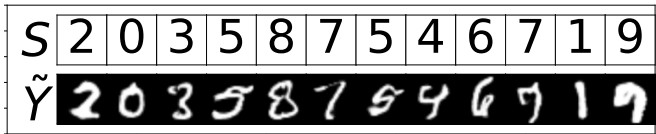

Figure 11: MNIST generation without the fairness mechanism ($\lambda_F = 0$) and without privacy ($a = 10^{-6}$), using input vector $\mathbf{S}$.

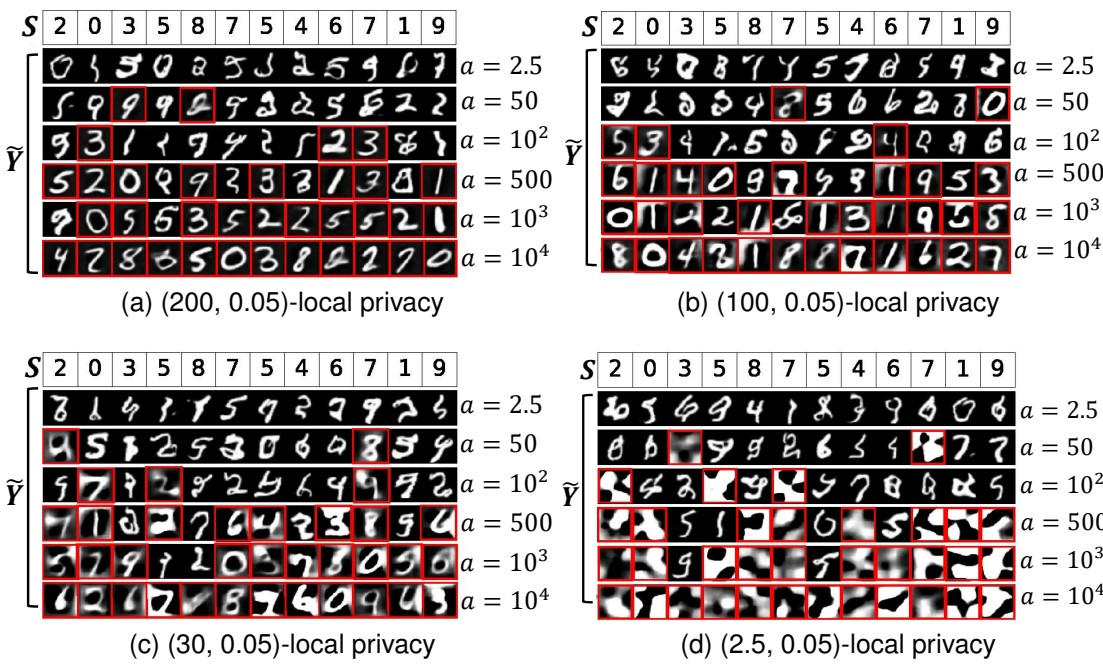

Figure 12: MNIST generation in the presence of the fairness mechanism ($\lambda_F = 1$) with input vector $\mathbf{S}$.

## I.3 ADULT DATASET

### I.3.1 DATA DISCRIPTION

The **Adult dataset** is a well-known benchmark dataset derived from the 1994 US Census Bureau database. It contains demographic and employment information from a sample of 48,842 adults with 15 columns in total. This dataset is widely used for income prediction and for evaluating fairness-aware machine learning methods.

**Continuous Attributes:**

- `age` (A): Age of the individual (in years).
- `fnlwgt` (F): Final weight, representing the number of people the census believes the entry represents.
- `education_num` (EN): Number of years of education.
- `capital_gain` (CG): Capital gain, income from investments.
- `capital_loss` (CL): Capital loss, loss from investments.
- `hours_per_week` (HPW): Hours worked per week.

**Discrete Attributes:**

- `workclass` (W): Employment status, with categories:
  - 'Private', 'Self-emp-not-inc', 'Self-emp-inc', 'Federal-gov', 'Local-gov', 'State-gov', 'Without-pay', 'Never-worked'.
- `education` (E): Level of education, with categories:
  - 'Bachelors', 'Some-college', '11th', 'HS-grad', 'Prof-school', 'Assoc-acdm', 'Assoc-voc', '9th', '7th-8th', '12th', 'Masters', '1st-4th', '10th', 'Doctorate', '5th-6th', 'Preschool'.
- `marital_status` (MS): Marital status, with categories:
  - 'Married-civ-spouse', 'Divorced', 'Never-married', 'Separated', 'Widowed', 'Married-spouse-absent', 'Married-AF-spouse'.
- `occupation` (O): Type of occupation, with categories:
  - 'Tech-support', 'Craft-repair', 'Other-service', 'Sales', 'Exec-managerial', 'Prof-specialty', 'Handlers-cleaners', 'Machine-op-inspct', 'Adm-clerical', 'Farming-fishing', 'Transport-moving', 'Priv-house-serv', 'Protective-serv', 'Armed-Forces'.
- `relationship` (Re): Relationship within family, with categories:
  - 'Wife', 'Own-child', 'Husband', 'Not-in-family', 'Other-relative', 'Unmarried'.
- `race` (Ra): Self-identified race, with categories:
  - 'White', 'Asian-Pac-Islander', 'Amer-Indian-Eskimo', 'Other', 'Black'.
- `sex` (S): Biological sex, with categories:
  - 'Male', 'Female'.
- `native_country` (NC): Country of origin, with categories:
  - 'United-States', 'Cambodia', 'England', 'Puerto-Rico', 'Canada', 'Germany', 'Outlying-US(Guam-USVI-etc)', 'India', 'Japan', 'Greece', 'South', 'China', 'Cuba', 'Iran', 'Honduras', 'Philippines', 'Italy', 'Poland', 'Jamaica', 'Vietnam', 'Mexico', 'Portugal', 'Ireland', 'France', 'Dominican-Republic', 'Laos', 'Ecuador', 'Taiwan', 'Haiti', 'Columbia', 'Hungary', 'Guatemala', 'Nicaragua', 'Scotland', 'Thailand', 'Yugoslavia', 'El-Salvador', 'Trinadad&Tobago', 'Peru', 'Hong', 'Holand-Netherlands'.
- `income` (I): Binary income class, with categories:
  - '$\leq$ 50K', '$>$ 50K'.

### I.3.2 PREPARING DATASET

- The `sensitive` column is moved to the last position to maintain a consistent structure.
- Rows with missing values (`NaN`) and incorrect header rows (where `sensitive` is mistakenly used as a value) are removed.
- Continuous attributes are converted to numeric types to ensure proper mathematical operations during training.

### I.3.3 DATASET LIMITATIONS

Ding et al. (2021) showed that the Adult dataset suffers from a critical limitation arising from its binary target label defined by the $50,000 income threshold. Because this threshold corresponds to the 76th income quantile overall, and the 88th and 89th quantiles for Black individuals and women, respectively, it induces a highly unbalanced label distribution across groups. As demonstrated in Ding et al. (2021), empirical fairness findings become highly sensitive to this extreme threshold: the magnitude of fairness violations, the trade-offs between fairness criteria, and the apparent effectiveness of algorithmic interventions all change substantially as the threshold varies. Consequently, the Adult dataset may not reliably indicate which fairness algorithm is genuinely better or worse, since both similar and contrasting results can arise from this threshold-induced imbalance rather than from true differences in algorithmic performance.

### I.3.4 EVALUATION RESULTS

The trade-off discussed in Appendix H is illustrated in Fig. 2a, particularly in Case 4, where all attributes receive a localized privacy budget of $(2.5, 0.05)$. When the concentration parameter is set to $a = 10$ and $a = 5$, this leads to global budgets of $(7.54, 0)$ and $(29.35, 0.01)$, respectively. The MMD values in these settings illustrate that meaningful utility can be preserved despite significant local perturbation, through appropriate tuning of $a$.

**Fairness Evaluation through Correlation Heatmaps:** To further investigate fairness, we provide heatmaps of the correlation matrices for real and generated samples, considering the sensitive attributes `sex` (S) and `relationship` (Re), as shown in Figures 13 and 14, respectively. In each case, the block representing the correlation between the sensitive attribute and income (I) is highlighted by a green frame, indicating values close to zero when $\lambda_F = 1$. This reflects the direct effect of minimizing the mutual information (MI) between the generated $\mathbf{I}$ and the sensitive attributes, effectively reducing unwanted dependencies.

**Fairness Flexibility for Multi-Category Sensitive Attributes:** The example in Figure 14, which includes 6 categories for the sensitive attribute relationship, demonstrates a significant advantage of our model over baselines designed primarily to minimize SP for binary sensitive attributes. This broader applicability highlights the flexibility of our approach in addressing fairness data synthesis for sensitive attributes with more than two categories.

**A Notion on Confounded Dependencies:** Additionally, some correlations between $\mathbf{I}$ and unprotected attributes (last row of the correlation matrices) are also reduced in the presence of fairness. This effect is particularly pronounced for the correlations between $\mathbf{I}$ and $\mathbf{S}$ as well as $\mathbf{I}$ and $\mathbf{R}e$, given the strong dependency between $\mathbf{S}$ and $\mathbf{R}e$ with a true correlation of $-0.62$. The direct path given by Figure 15a should not lead to the mistaken expectation that no reduction occurs in other attributes, as it can overlook the impact of confounded relationships. For example, when the sensitive attribute is $\mathbf{S}$, undirected connections between $\mathbf{I}$ and other attributes, mediated through $\mathbf{S}$, are shown in Figure 15b. This indicates that minimizing the MI between $\mathbf{I}$ and $\mathbf{S}$ during training reduces not only their direct dependence but also the indirect correlations with other attributes influenced by these confounded relationships, as illustrated in Figures 15d and 15e.

**Consistency Investigation:** Figure 16 illustrates the consistency of our proposed approach as the sample size approaches infinity. The results clearly demonstrate that utility, fairness, and balance are strongly supported with increasing sample size, providing numerical validation for the theoretical guarantees established in Theorem 2.

**Scalability Advantage of the Proposed CBNP Framework:** Based on our tests, running 1000 iterations on the Adult dataset took between 3.5 to 4 hours. This is relatively efficient, especially compared to one of our state-of-the-art baselines, DECAF, which took around 8 days under similar conditions. Notably, DECAF is designed solely for enforcing fairness, whereas our framework simultaneously addresses three key policy requirements, fairness included. Attempting to adapt DECAF for additional guarantees—such as applying DP-SGD to introduce gradient noise for differential privacy—would significantly complicate its implementation and further increase computational

burden. This highlights the scalability advantage of our proposed framework despite incorporating Bayesian nonparametric modeling and fairness-privacy components.

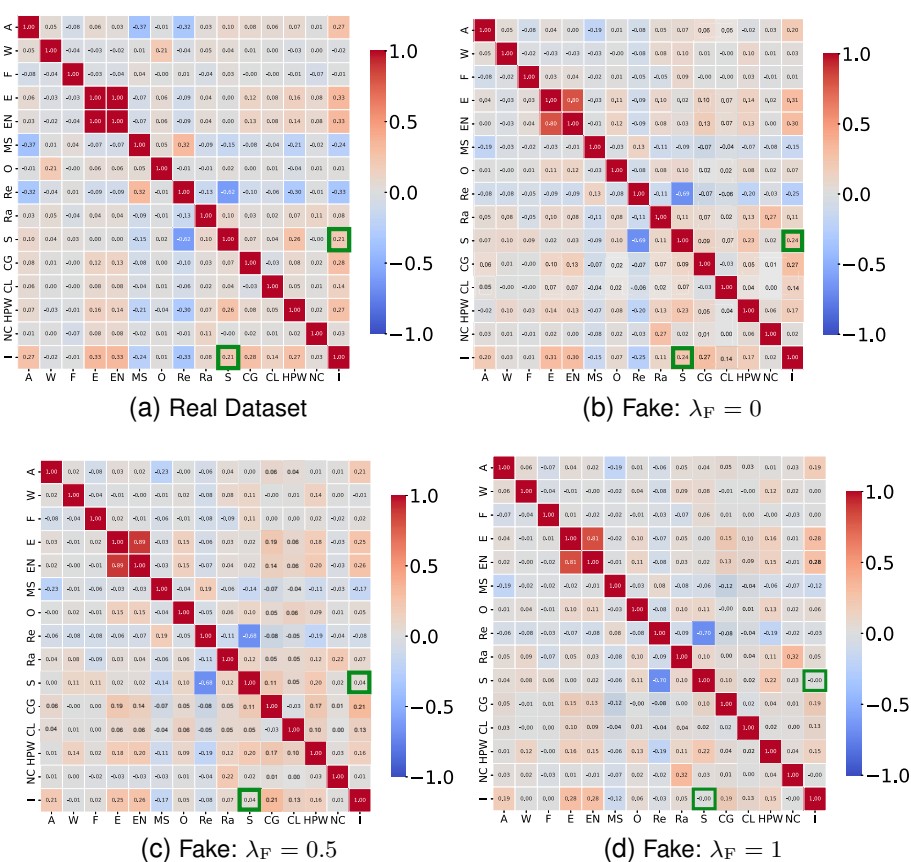

Figure 13: Heatmap of the correlation matrix for the Adult Dataset (real) and synthetic samples (fake), where *sex* is the sensitive attribute and *income* ($>$50K) is the desired outcome.

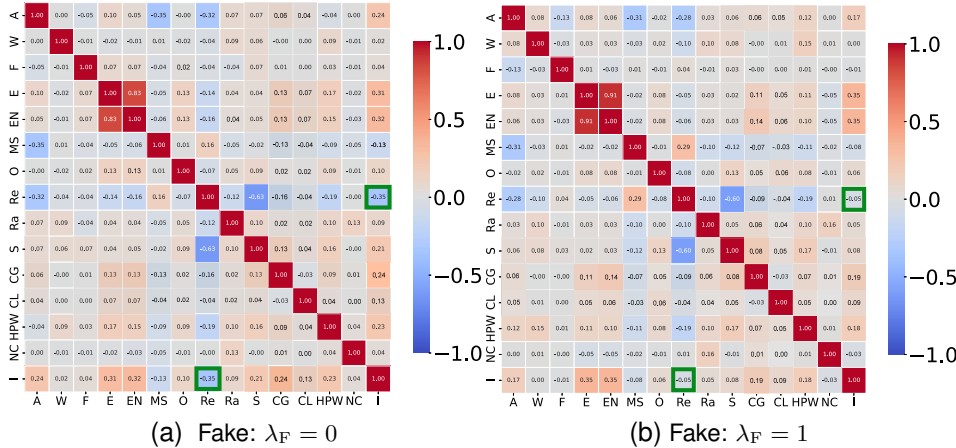

Figure 14: Heatmap of the correlation matrix for the Adult Dataset (real) and synthetic samples (fake), where *relationship* is the sensitive attribute and *income* ($>$50K) is the desired outcome.

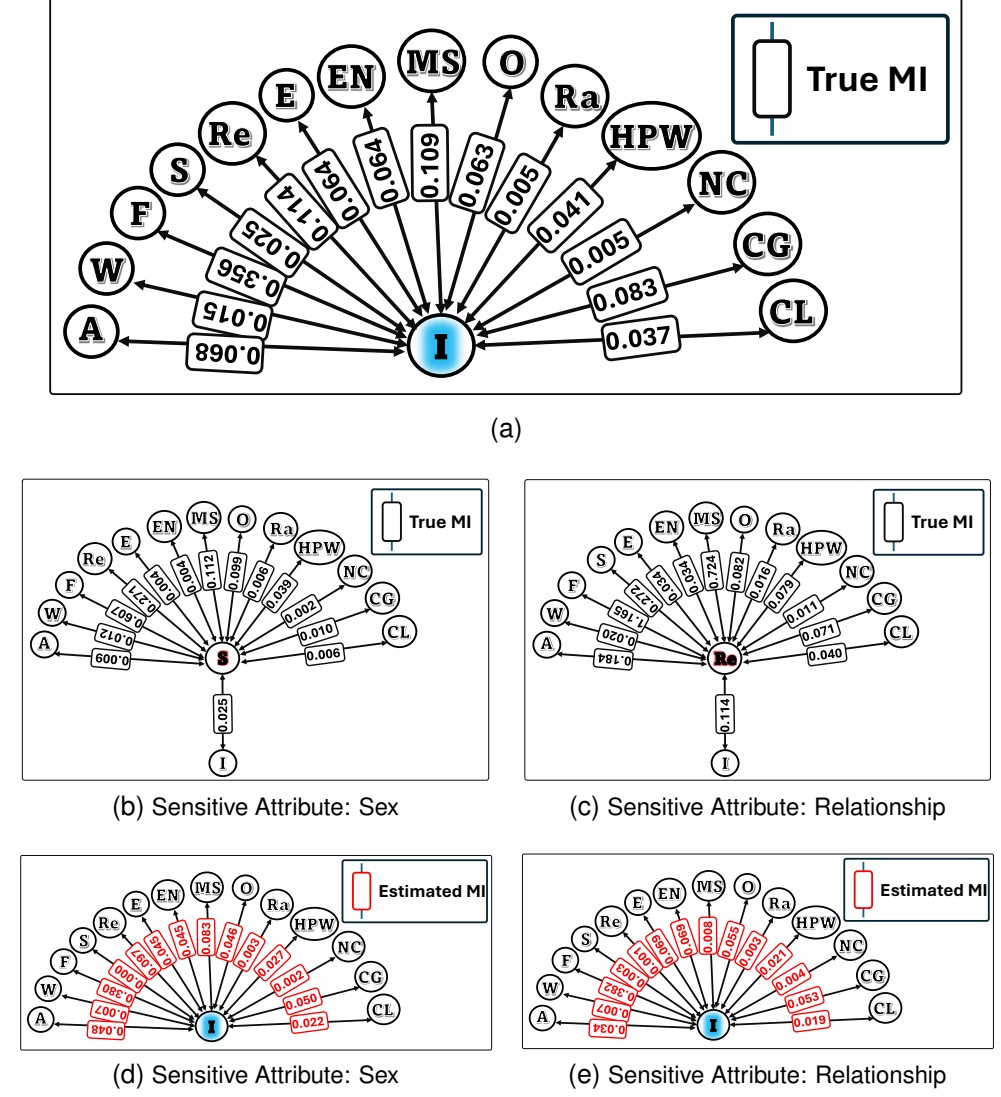

Figure 15: (a) Direct dependency structure between 'I' and related attributes. (b) Dependency structure between 'S' and its related attributes. (c) Dependency structure between 'Re' and its related attributes. (d), (e) Estimated dependency structure between 'I' and related attributes in the presence of fairness, i.e. $\lambda_F = 1$.

## I.4 PROPUBLICA COMPAS DATASET

### I.4.1 DATE DISCRIPTION

We use the **COMPAS** data, downloaded from the ProPublica repository, which contains 60,843 records and 28 attributes. After removing personally identifiable information such as person identifiers, names, and date of birth, the remaining attributes can be shown into two groups:

**Continuous Attributes:**

- Raw_Score: numeric COMPAS risk score before scaling.

- Decile_Score: risk score transformed into a decile from 1 (lowest) to 10 (highest).

- Rec_Supervision_Level: recommended supervision level, recorded as a numeric value.

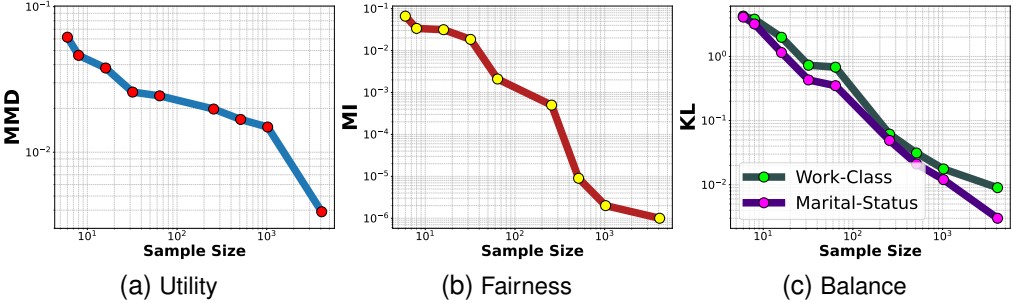

Figure 16: Consistency of the proposed approach as the sample size increases with $\lambda_F = 1$, aiming for balance with respect to **W** and **MS**. All metrics are measured between generated samples and their corresponding true underlying distributions.

**Discrete (Categorical) Attributes:**

- `Agency`: agency conducting the assessment.
- `Sex`: gender of the individual.
- `Race/Ethnicity`: recorded racial or ethnic group.
- `Assessment_Reason`: reason why the assessment was performed.
- `Language`: preferred language for communication.
- `Legal_Status`: the individual's current legal status.
- `Custody_Status`: whether the individual was in custody at the time of assessment.
- `Marital_Status`: marital status of the individual.
- `Screening_Date`: date when the screening was completed.
- `Rec_Supervision_Level_Text`: textual description of supervision level.
- `Scale_Information`: identifiers and names of the COMPAS scales used (`ScaleSet_ID`, `ScaleSet`, `Scale_ID`).
- `Display_Text`: the type of risk being scored (e.g., recidivism, violence).
- `Score_Text`: qualitative risk category (*Low*, *Medium*, or *High*).
- `Assessment_Type`: whether the assessment is initial, a reassessment, or another type.
- `Completion_Flags`: indicators for whether the assessment was completed (`IsCompleted`) or marked as deleted (`IsDeleted`).

### I.4.2 DATA LIMITATION

Bao et al. (2021) argued that the COMPAS dataset contains substantial measurement error, jurisdiction-specific artifacts, and systemic policing disparities that influence both the covariates and the outcome labels. Crucially, key outcomes such as rearrest or failure-to-appear do not reflect ground-truth risk but are shaped by discretionary criminal-justice processes, leading to structurally biased labels. These limitations imply that fairness metrics computed on COMPAS should not be interpreted as evidence of real-world fairness: a reduction in demographic-parity disparity within the dataset does not translate into fairer decision-making in practice. Consequently, although COMPAS may remain useful as a controlled benchmark for comparing algorithmic behavior, it cannot support claims about fairness in real criminal-justice settings.

### I.4.3 EVALUATION RESULTS

In this dataset, the column `Score_Text` represents how the COMPAS system classifies an individual's risk of future crime. It has three possible values: *Low*, *Medium*, and *High*, which correspond to whether the system believes a person is unlikely, somewhat likely, or highly likely to reoffend.

These risk categories matter because they can influence important decisions such as bail amounts, sentencing, or parole recommendations. The column `Ethnic` identifies each person's racial or ethnic group, such as African-American, Caucasian, Hispanic, Asian, and others. Because race is a protected attribute, it should not unfairly affect whether someone is labeled *Low*, *Medium*, or *High* risk.

For clarity, throughout our discussion we use $\mathbf{Y}$ to denote the outcome variable—here, $\mathbf{Y} =$ `Score-Text`, which encodes the risk category assigned by COMPAS—and $\mathbf{S}$ to denote the sensitive attribute—here, $\mathbf{S} =$ `Ethnic`, which represents a person's race or ethnic group. When studying fairness with this dataset, we examine how the distribution of $\mathbf{Y}$ varies across the categories of $\mathbf{S}$. If one group has a disproportionately high probability of being labeled "High" risk compared to others, even when there is no legitimate predictive justification for that difference, it may indicate bias in the risk assessment system.

The conditional distribution is shown in Table 1 and clearly illustrates disparities across racial and ethnic groups. For example, for the outcome $\mathbf{Y} =$ "Low",

$$\Pr\big(\mathbf{Y} = \text{"Low"} \mid \mathbf{S} = \text{"African-American"}\big) \neq \cdots \neq \Pr\big(\mathbf{Y} = \text{"Low"} \mid \mathbf{S} = \text{"other"}\big), \quad (35)$$

indicating that the probability of individuals from different ethnic groups being classified as "Low" risk is not the same, and similarly for the "Medium" and "High" categories. This inequality highlights the presence of group-level disparities that our method seeks to mitigate in the generated data.

Any generative model built to create synthetic data from this unfair dataset should preserve meaningful relationships (e.g., between past behavior and risk category) while reducing or removing spurious dependencies between $\mathbf{S}$ (race) and $\mathbf{Y}$ (risk label). This ensures that the generated data promotes fairness and does not amplify racial disparities present in the original data.

Table 13: **Compas:** Conditional probabilities $\Pr(\mathbf{Y} = $ **"Score Text"** $\mid \mathbf{S} = $ **"Ethnic"**$)$ for each racial or ethnic group in the ProPublica COMPAS dataset, where $\mathbf{Y}$ is the COMPAS risk category and $\mathbf{S}$ is the race/ethnicity.

| Ethnic Group | Score Text | | |
|---|---|---|---|
| | Low | Medium | High |
| African-American | 0.572 | 0.259 | 0.169 |
| Arabic | 0.773 | 0.160 | 0.067 |
| Asian | 0.858 | 0.099 | 0.043 |
| Caucasian | 0.752 | 0.173 | 0.076 |
| Hispanic | 0.797 | 0.149 | 0.054 |
| Native American | 0.639 | 0.215 | 0.146 |
| Oriental | 0.846 | 0.103 | 0.051 |
| Other | 0.837 | 0.128 | 0.034 |

In Section 5, it was demonstrated that the model performs well on this dataset while simultaneously incorporating three key policies: privacy, fairness, and class balancing. As complementary material, bar plots for the attributes `sex`, `marital`, and `language` are presented in Fig. 17, comparing category percentages across the original data, the unbalanced generated data, and the balanced generated data.

In all three plots, the green bars represent the ideal balanced proportions, assigning equal weight to each category as a baseline. The orange bars display the category proportions in the generated dataset produced by the trained model without applying balance constraints on the three attributes.

Table 14: **Compas:** Distribution of racial/ethnic groups in the original COMPAS dataset.

| Ethnic Group | Proportion | Percentage ($S_{rsr}$) |
|---|---|---|
| African-American | 0.444899 | 44.49% |
| Caucasian | 0.358020 | 35.80% |
| Hispanic | 0.143681 | 14.37% |
| Other | 0.042601 | 4.26% |
| Asian | 0.005325 | 0.53% |
| Native American | 0.003599 | 0.36% |
| Arabic | 0.001233 | 0.12% |
| Oriental | 0.000641 | 0.06% |

As expected, the charts show that the generated samples become balanced once the balancing constraints are activated. For example, the proportion of the minority Spanish language group increases toward the ideal balance level.

## I.5 BANK MARKETING DATASET

### I.5.1 DATA DESCRIPTION

We use the **Bank Marketing** dataset from the UCI Machine Learning Repository, which contains 45,211 records and 21 attributes collected from a series of direct marketing campaigns conducted by a Portuguese banking institution between May 2008 and November 2010. No campaigns were run during January and February of each year, and therefore these months do not appear in the dataset. Each record corresponds to a single phone contact with a client, with attributes describing demographic information, previous interactions, economic indicators, and campaign details. Each row corresponds to a single phone contact made to a client as part of a campaign aimed at promoting subscription to a term deposit product. The attributes can be grouped into two classes:

**Continuous Attributes:**

- `age`: age of the client in years.
- `duration`: duration of the last call, in seconds.
- `campaign`: number of contacts performed during this campaign for this client.
- `pdays`: number of days since the client was last contacted from a previous campaign ($-1$ means never contacted).
- `previous`: number of contacts performed before this campaign.
- `emp_var_rate`: employment variation rate (quarterly indicator).
- `cons_price_idx`: consumer price index.
- `cons_conf_idx`: consumer confidence index.
- `euribor3m`: Euribor 3-month rate.
- `nr_employed`: number of employees (economic activity indicator).

**Discrete (Categorical) Attributes:**

- `job`: type of job (e.g., admin., technician, blue-collar).
- `marital`: marital status (married, single, divorced).
- `education`: education level (basic, secondary, tertiary, unknown).
- `default`: whether the client has credit in default (yes, no).

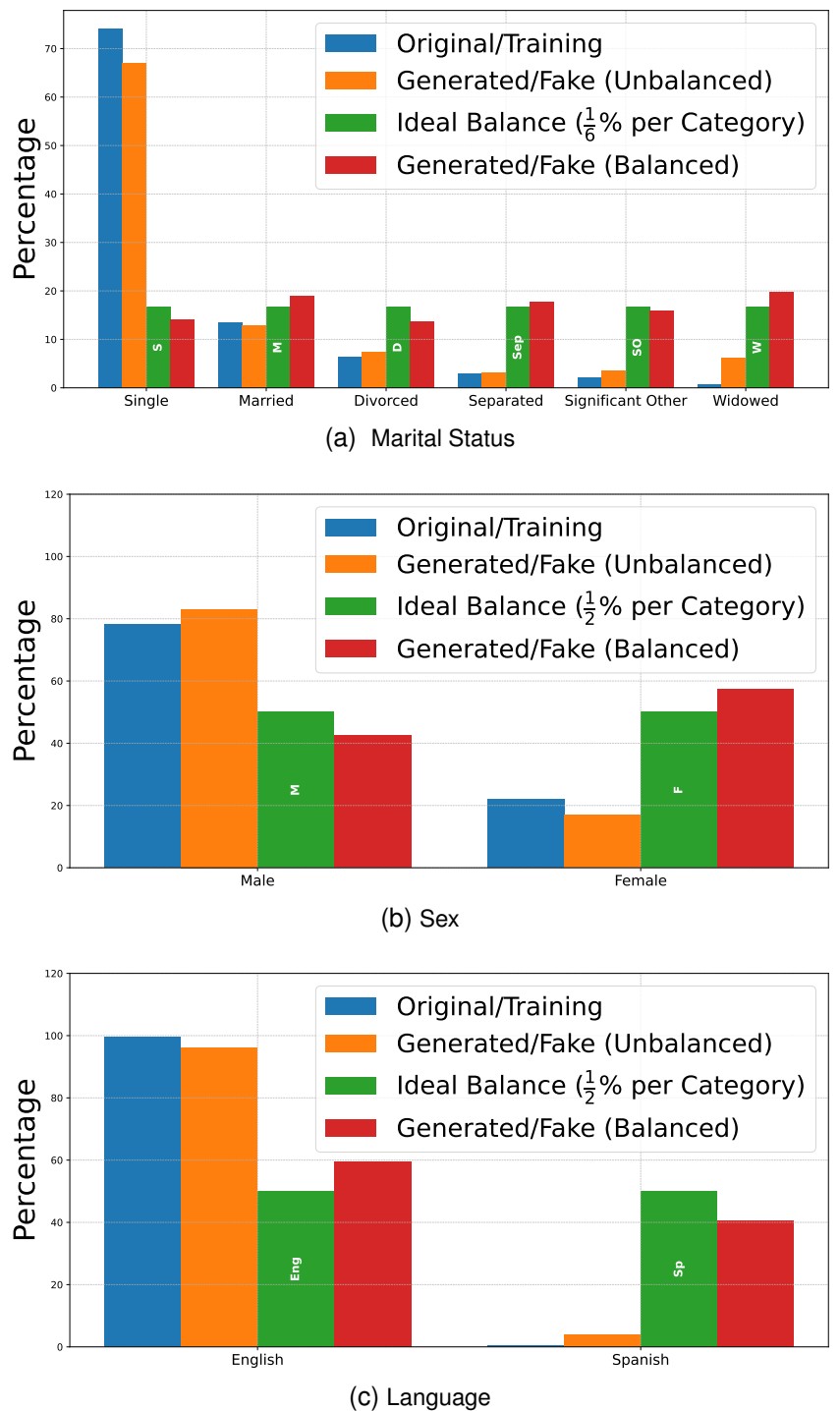

Figure 17: **Compas:** Bar plots of different categories for three categorical unprotectd attributes `Marital_Status`, `Sex`, and `Language`.

- `housing`: whether the client has a housing loan (yes, no).
- `loan`: whether the client has a personal loan (yes, no).
- `contact`: type of communication contact (cellular, telephone).
- `month`: last contact month of the year.
- `day_of_week`: last contact day of the week.
- `poutcome`: outcome of the previous marketing campaign (success, failure, nonexistent).
- `subscribed`: binary target variable indicating whether the client subscribed to a term deposit (`yes` or `no`).

### I.5.2 EVALUATION RESULTS

In this dataset, the column `subscribed` records the outcome of the marketing campaign, specifying whether the client subscribed to a term deposit. The values are binary: *yes* (the client subscribed) or *no* (the client did not). This variable is the primary focus of predictive modeling as it directly reflects campaign success.

The feature `month` indicates the month in which the client was last contacted, with possible values from March to December. January and February are absent because the bank did not conduct campaigns in those months during the collection period. Although `month` is not a demographic attribute, it captures strong seasonal and campaign-intensity effects: some months show markedly higher subscription rates, which are likely driven by marketing schedules rather than inherent differences in client willingness. Relying on such a feature may cause models—and synthetic data generators—to overfit to seasonal trends rather than underlying behavior, potentially skewing analyses or downstream decisions.

Here $\mathbf{Y} = $ Subscribed and $\mathbf{S} = $ Month. Our fairness assessment examines how the distribution of $\mathbf{Y}$ varies across the different months represented by $\mathbf{S}$. When certain months have disproportionately high probabilities of subscription, this reflects a form of campaign bias that may not be desirable to reproduce.

The conditional distribution is given in Table 15, which makes clear that subscription probabilities differ substantially by month. For instance, for the outcome $\mathbf{Y} = $ "Yes",

$$\Pr\big(\mathbf{Y} = \text{"Yes"} \mid \mathbf{S} = \text{"March"}\big) \neq \cdots \neq \Pr\big(\mathbf{Y} = \text{"Yes"} \mid \mathbf{S} = \text{"May"}\big),$$

showing that clients contacted in different months do not have equal likelihoods of subscribing. These differences are likely driven by campaign timing and call strategies, and thus are not necessarily representative of persistent client characteristics.

We aim to generate data to evaluate the model's performance on this dataset and to verify whether it preserves useful patterns while mitigating the month-specific biases present in this additional dataset.

To evaluate our approach, we again employ the tools outlined in Section 5, using the same settings for the local and global privacy budgets and the equal proportions of sensitive attributes described in Section 5. Here, two active attributes, `marital` and `education`, are considered for balancing. This setup allows us to verify that the generated data achieve a balance between utility and fairness while satisfying the specified privacy requirements.

**Utility.** Regarding Table 16, by comparing the non-fairness part (orange cells) with the proportions of the original dataset given in Table 15, it is evident that the model preserves perfect utility in the absence of fairness constraints. We compute the MMD to quantify the similarity between the generated and original data distributions. Lower values indicate closer alignment, which is crucial for maintaining fidelity to the original population. Additionally, predictive performance is assessed through F1-score and overall accuracy (Acc) using a classifier trained on synthetic data and validated on real hold-out data. High scores suggest that relevant relationships between variables are preserved.

**Fairness.** We use Mutual Information (MI) between $\mathbf{S}$ and $\mathbf{Y}$ to measure how much information the month of contact carries about the likelihood of subscription. Negligible MI in Table 17 implies weaker dependence, signaling that the generative model has reduced the influence of month-specific campaign effects. Furthermore, the results in the blue region of Table 16 confirm that the model

Table 15: **Bank Marketing:** Conditional probabilities $\Pr(Y = \text{‘Subscribed”} \mid S = \text{Month”})$ for each contact month in the original dataset, where $Y$ is the subscription outcome and $S$ is the contact month.

| Month | Subscription Outcome | |
|---|---|---|
| | No | Yes |
| March | 0.495 | 0.505 |
| April | 0.795 | 0.205 |
| May | 0.936 | 0.064 |
| June | 0.895 | 0.105 |
| July | 0.910 | 0.090 |
| August | 0.894 | 0.106 |
| September | 0.551 | 0.449 |
| October | 0.561 | 0.439 |
| November | 0.899 | 0.101 |
| December | 0.511 | 0.489 |

Table 16: **Bank Marketing:** Conditional probabilities $\Pr(Y = \text{“Subscribed”} \mid S = \text{“Month”})$ for each contact month in the generated samples, where $Y$ is the subscription outcome and $S$ is the contact month.

| Month | Subscription Outcome ($\lambda_F = 0$) | | Subscription Outcome ($\lambda_F = 1$) | |
|---|---|---|---|---|
| | No | Yes | No | Yes |
| March | 0.501 | 0.499 | 0.863 | 0.137 |
| April | 0.763 | 0.237 | 0.812 | 0.188 |
| May | 0.911 | 0.089 | 0.900 | 0.100 |
| June | 0.904 | 0.096 | 0.854 | 0.146 |
| July | 0.896 | 0.104 | 0.917 | 0.083 |
| August | 0.913 | 0.087 | 0.867 | 0.133 |
| September | 0.498 | 0.502 | 0.827 | 0.173 |
| October | 0.592 | 0.408 | 0.888 | 0.112 |
| November | 0.926 | 0.074 | 0.902 | 0.098 |
| December | 0.497 | 0.503 | 0.886 | 0.114 |

successfully yields nearly equal probabilities across the two categories of the sensitive attribute `month`.

Together, MMD, F1-score, and Acc measure how well utility is retained, while MI reflects fairness improvement. Finally, because $S$ (month) is multi-class, most binary fairness baselines are not applicable, underscoring the importance of approaches—such as ours—that handle multi-category sensitive attributes.

Table 17: **Bank Marketing:** $\mathrm{MI}_{\mathrm{True}}(\mathbf{Y}, \mathbf{S}) = 0.0264$, $\mathrm{F1}_{\mathrm{True}} = 0.540$, and $\mathrm{Acc}_{\mathrm{True}} = 0.884$ over 10 runs.

| $\lambda_{\mathrm{B}} = 1$ | $\lambda_{\mathrm{F}}$ | **MI** | **MMD** | **F1 (DTC)** | **Acc (DTC)** |
|---|---|---|---|---|---|
| None | 0 | 0.0281 | 0.0014 | 0.855 | 0.531 |
|  | 1 | $1e-5$ | 0.0020 | 0.851 | 0.523 |
| Sex Education | 1 | $1e-5$ | 0.0026 | 0.858 | 0.525 |

## J GRAPHICAL OVERVIEW AND CLARIFICATION

**Tabular Data Representation:** A general diagram in Fig. 18 illustrates the tabular data representation, making the concepts in Section 4 easier to follow.

**Contribution of Perturbed Prior $H^{\mathrm{Pert}}$ to Localized Privacy:** A review of the construction of $H^{\mathrm{Pert}}$ is provided in Fig. 19, which depicts how the base measure in the Dirichlet Process is designed to inject localized differential privacy.

**Target Attack Classes:** An overall overview of the fair–private mechanisms in the proposed approach for addressing several types of attackers is presented in Fig. 20, providing a clearer understanding of all mechanisms. For further details on such attackers, see Luo et al. (2024).

## K COMPUTATIONAL ALGORITHM

Algorithm 1 outlines the computational steps for generating samples from $H^{\mathrm{Pert}}$ using a latent Gaussian copula approach. Figure 21 presents an overview of the proposed generative models for synthesizing private, fair, and balanced tabular data.

We clarify that the perturbed base measure $H^{\mathrm{Pert}}$ is constructed **only once**, prior to training. It is **not recomputed at each iteration**. Once built, $H^{\mathrm{Pert}}$ is used to define the posterior base measure $H^*$, from which we repeatedly sample during training. This sampling step is computationally comparable to drawing from any standard distribution and does not incur significant overhead. Because this construction is a **one-time operation**, the method remains scalable even for datasets with dozens of variables. The perturbation of marginal distributions for the base measure is done **upfront**, and the rest of the training proceeds with standard sampling-based techniques.

Algorithm 2 displays the implementation steps of the proposed BNP framework for generating fair, private, and balanced samples, implemented using VAE$\mathcal{C}$GAN, a member of the generator/decoder–based family of generative models. We note that the triple-policy proposed in this paper can also be implemented with other members of this family by applying their respective algorithms.

## L LIMITATIONS

In our model, fairness is enforced by explicitly conditioning the generator on protected attributes and minimizing the MI between outputs and those attributes. This is feasible in CGANs with structured inputs, but in LLMs like ChatGPT, conditioning occurs implicitly via prompts, and protected attributes are not explicitly disentangled, making it difficult to define or minimize MI in a well-posed manner. On the privacy side, our BNP mechanism perturbs structured data distributions using a DirP with a copula-based base measure, which is tractable for tabular data but not directly applicable to token-level LLM training in its current form, especially since LLMs lack access to explicit marginal structures. However, we hope to address this by leveraging DirP mixture models to construct a sequence-level prior over token representations, enabling structured uncertainty at the sequence or span level while preserving coherence and privacy in autoregressive generation. Extending our framework would thus require a fundamental reformulation of how fairness and privacy constraints are represented and enforced in autoregressive architectures.

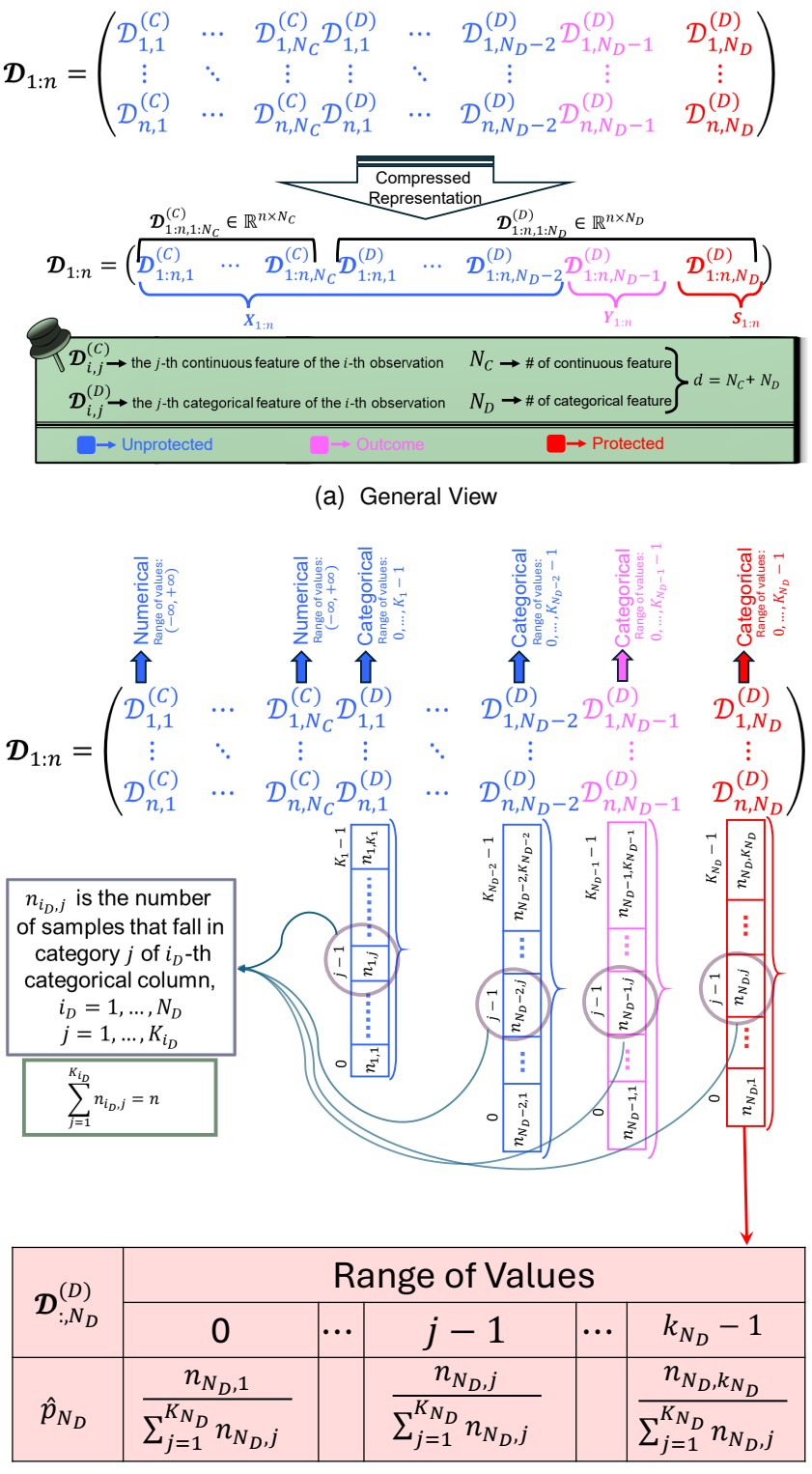

Figure 18: Clarified view of the triple tabular data representation.

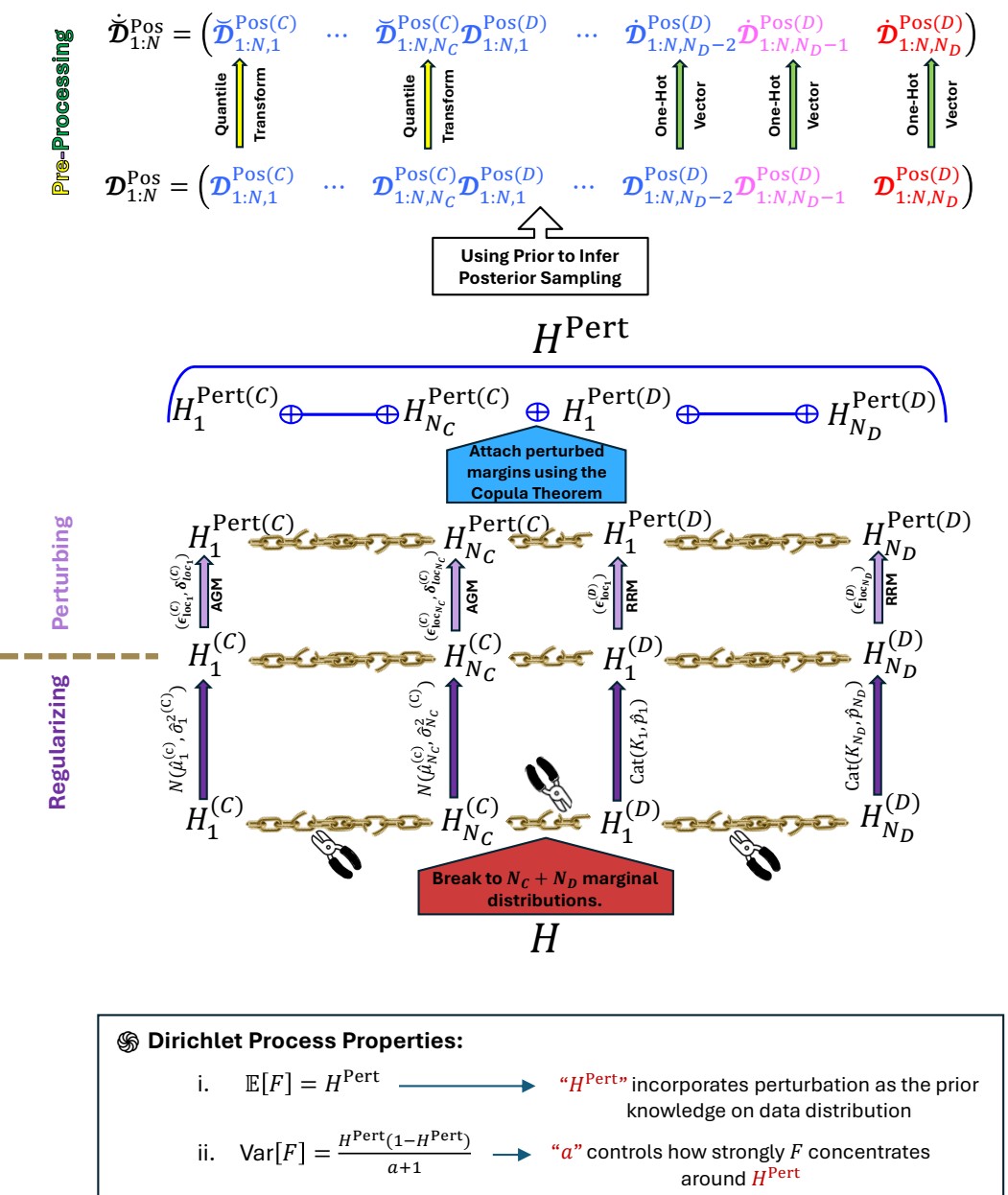

Figure 19: Perturbation scheme for designing $H^{\text{Pert}}$ to inject localized privacy.

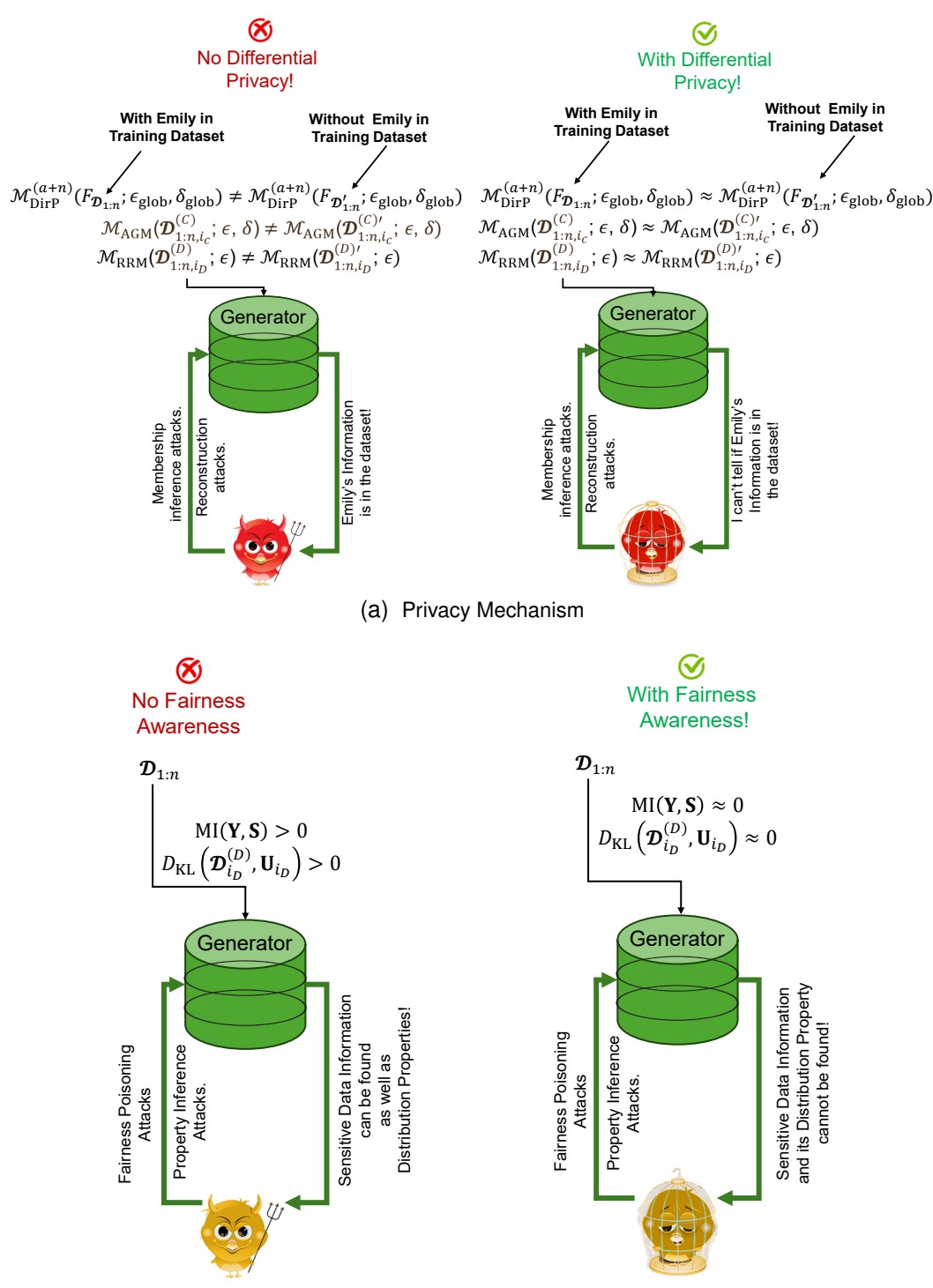

Figure 20: Overview of the triple policies (Fair–Private–Balanced) enforced by our proposed approach to address potential attacks.

---

**Algorithm 1:** Generate a random vector from $H^{\text{Pert}}$ using a latent Gaussian copula approach

---

**Input:**

$\hat{R}$: Estimated correlation matrix of dimension $(d \times d)$

$F_{\text{AGM},i_{\text{C}}}^{-1}$: Inverse of the continuous CDF $F_{\text{AGM},i_{\text{C}}}$

$(p_{i_{\text{D}}1}^{\text{Pert}}, \ldots, p_{i_{\text{D}}K_{i_{\text{D}}}}^{\text{Pert}})$: Probability vector for the categorical variable $\mathcal{M}_{\text{RRM}}(\mathcal{D}_{i_{\text{D}}}^{\text{Pri(D)}}; \epsilon_{\text{loc}_{i_{\text{D}}}}^{\text{(D)}})$

taking values from set $\{1, \ldots, K_{i_{\text{D}}}\}$

$K_{i_{\text{D}}}$: Number of categories for each categorical variable

**Output:** $\mathcal{D}^{\text{Pert}} = (\mathcal{D}_1^{\text{Pert(C)}}, \ldots, \mathcal{D}_{N_{\text{C}}}^{\text{Pert(C)}}, \mathcal{D}_1^{\text{Pert(D)}}, \ldots, \mathcal{D}_{N_{\text{D}}}^{\text{Pert(D)}})$

---

1 **Sample from Multivariate Normal Distribution (Latent Copula Variable):**

2 $\mathbf{Z} = (\overbrace{Z_1, \ldots, Z_{N_{\text{C}}}}^{\text{Number of terms: } N_{\text{C}}}, \overbrace{Z_{N_{\text{C}}+1}, \ldots, \underbrace{Z_{N_{\text{C}}+N_{\text{D}}}}_{d}}^{\text{Number of terms: } N_{\text{D}}}) \sim \mathcal{N}(0, \hat{R});$   // $\mathbf{Z}$ is of size $(1 \times d)$

3 **Continuous Variables**

4     **for** $i_{\text{C}} = 1$ **to** $N_{\text{C}}$ **do**

5         $U_{i_{\text{C}}} \leftarrow \Phi(Z_{i_{\text{C}}});$            // Apply standard normal CDF

6         $\mathcal{D}_{i_{\text{C}}}^{\text{Pert(C)}} \leftarrow F_{\text{AGM},i_{\text{C}}}^{-1}(U_{i_{\text{C}}});$     // Apply inverse of marginal CDF

7 **Discrete Variables**

8     **for** $i_{\text{D}} = 1$ **to** $N_{\text{D}}$ **do**

9         **Creating Thresholds**

10             $\tau_{i_{\text{D}}1} \leftarrow \Phi^{-1}(p_{i_{\text{D}}1}^{\text{Pert}});$ // First threshold from first probability

11             **for** $k = 2$ **to** $K_{i_{\text{D}}} - 1$ **do**

12                 $\tau_{i_{\text{D}}k} \leftarrow \Phi^{-1}\left(\sum_{i=1}^{k} p_{i_{\text{D}}i}^{\text{Pert}}\right);$     // Subsequent thresholds from cumulative sum

13         **if** $Z_{N_{\text{C}}+i_{\text{D}}} \leq \tau_{i_{\text{D}}1}$ **then**

14             $\mathcal{D}_{i_{\text{D}}}^{\text{Pert(D)}} \leftarrow 1$

15         **else if** $\tau_{i_{\text{D}}1} < Z_{N_{\text{C}}+i_{\text{D}}} \leq \tau_{i_{\text{D}}2}$ **then**

16             $\mathcal{D}_{i_{\text{D}}}^{\text{Pert(D)}} \leftarrow 2$

17         **else if** $\ldots$ **then**

18             $\vdots$                 // Continue for other thresholds

19         **else if** $Z_{N_{\text{C}}+i_{\text{D}}} > \tau_{i_{\text{D}}K_{i_{\text{D}}}-1}$ **then**

20             $\mathcal{D}_{i_{\text{D}}}^{\text{Pert(D)}} \leftarrow K_{i_{\text{D}}}$

---

Separately, our fairness term is based on regular mutual information (MI), which directly measures dependence between the sensitive attribute and the generated outcomes. This formulation does not capture fairness notions that are inherently classifier-oriented—such as equalized odds, equality of opportunity, and predictive equality, because these notions require conditioning on the true outcome and therefore do not directly translate to generative models.

An additional extension of interest is *conditional statistical parity*, which is not classifier-based but extends statistical parity by conditioning on observable non-sensitive features. Supporting conditional SP within our framework would require estimating *conditional* MI under a Dirichlet-process formulation, specifically through a conditional DirPMINE estimator. Developing such an estimator is a substantial methodological undertaking and remains outside the scope of the present work.

---

**Algorithm 2:** Training CBNP-VAE$\mathcal{C}$GAN

---

**Input:** Training data $\mathcal{D}_{1:n}$; $n_{\text{iter}}$–iterations; $n_{\text{crit}}$–discriminator iterations; $n_{\text{DB}}$ – number of **D**iscrete variables to be **B**alanced (DB) with respect to their categories; $n_{\text{fair}}$–fair iterations; $n_{\text{balance}}$–balance iterations; $n_{mb}$–mini-batch size, $q = 10$; learning rates: $\alpha_{CG_{\boldsymbol{\omega}'}} = \alpha_{D_{\boldsymbol{\theta}}} = \alpha_{G_{\boldsymbol{\omega}}} = \alpha_{E_{\boldsymbol{\eta}}} = \alpha_{T_{\boldsymbol{\nu}}} = \alpha_{T_{\boldsymbol{\nu}_{i_{\text{D}}}}} = 2\text{e}^{-4}$; penalty $\lambda_p = 10$; Adam optimizer parameters: $(\varepsilon = 1\text{e}^{-8}, \beta_1 = 0.9, \beta_2 = 0.999)$.

**Output:** Optimized parameter of the generator

1   **for** $t = 1$ *to* $n_{\text{iter}}$ **do**

2     Sample $(\mathbf{x}_{1:n_{mb}}, \mathbf{y}_{1:n_{mb}}, \mathbf{s}_{1:n_{mb}}) \sim F$;

3     Sample $(\mathbf{x}_{1:N}^{\text{Pos}}, \mathbf{y}_{1:N}^{\text{Pos}}, \mathbf{s}_{1:N}^{\text{Pos}}) \sim F^{\text{Pos}}$ ;         `//` $F^{\texttt{Pos}} \sim \text{DP}(a + n_{mb}, H^{\texttt{Pert*}})$

4     $(\mathbf{x}_{1:N}^{\text{Pos}\prime}, \mathbf{y}_{1:N}^{\text{Pos}\prime}, \mathbf{s}_{1:N}^{\text{Pos}\prime}) \leftarrow (\mathbf{x}_{1:N}^{\text{Pos}}, \mathbf{y}_{1:N}^{\text{Pos}}, \mathbf{s}_{1:N}^{\text{Pos}})$ ;     `// Apply QT for continuous &` `one-hot for discrete columns`

5     **for** $t = 1$ *to* $n_{crit}$ **do**

6       $\boldsymbol{\xi}_{1:N} \sim N(\mathbf{0}_p, I_p), \boldsymbol{\xi}'_{1:N} \sim N(\mathbf{0}_q, I_q)$;

7       $\widetilde{\boldsymbol{c}}_{1:N} \leftarrow CG_{\boldsymbol{\omega}'}(\boldsymbol{\xi}'_{1:N}), \boldsymbol{c}_{1:N} \leftarrow E_{\boldsymbol{\eta}}(\mathbf{x}_{1:N}^{\text{Pos}}, \mathbf{y}_{1:N}^{\text{Pos}}), \boldsymbol{\ell}_{1:N} \leftarrow (\boldsymbol{\xi}_{1:N}, \boldsymbol{c}_{1:N}, \widetilde{\boldsymbol{c}}_{1:N})$;

8       $(\widetilde{\mathbf{x}}_{1:N}^{\text{Pos}}, \widetilde{\mathbf{y}}_{1:N}^{\text{Pos}}) \leftarrow G_{\boldsymbol{\omega}}(\boldsymbol{\ell}_{1:N}, \mathbf{s}_{1:N}^{\text{Pos}})$;

9       Update discriminator parameters:

10       $\boldsymbol{\theta} \leftarrow \text{Adam}(\nabla_{\boldsymbol{\theta}} \mathcal{L}(D_{\boldsymbol{\theta}} \mid \mathbf{s}^{\text{Pos}'}), \boldsymbol{\theta}, \varepsilon, \beta_1, \beta_2)$;

11     Update generator & encoder parameters for utility:

12     Repeat lines 6-8;

13     $(\boldsymbol{\omega}, \boldsymbol{\eta}) \leftarrow \text{Adam}(\nabla_{\boldsymbol{\omega}, \boldsymbol{\eta}} \mathcal{L}(G_{\boldsymbol{\omega}}, E_{\boldsymbol{\eta}} \mid \mathbf{s}^{\text{Pos}'}), (\boldsymbol{\omega}, \boldsymbol{\eta}), \varepsilon, \beta_1, \beta_2)$;

14     Update MI-DirPDV parameters to estimate MI:

15     Repeat lines 6-8;

16     $\boldsymbol{\nu} \leftarrow \text{Adam}(-\nabla_{\boldsymbol{\nu}} \mathcal{L}_{Fair}(G_{\boldsymbol{\omega}}, T_{\boldsymbol{\nu}} \mid \mathbf{s}^{\text{Pos}'}), \boldsymbol{\nu}, \varepsilon, \beta_1, \beta_2)$;

17     Update KL-DirPDV parameters to estimate KL:

18     Repeat lines 6-8;

19     **for** $i_D = 1$ *to* $n_{\text{DB}}$ **do**

20       $\boldsymbol{\nu}_{i_{\text{D}}} \leftarrow \text{Adam}(-\nabla_{\boldsymbol{\nu}_{i_{\text{D}}}} \mathcal{L}_{Balance}(G_{\boldsymbol{\omega}}, T_{\boldsymbol{\nu}_{i_{\text{D}}}} \mid \mathbf{s}^{\text{Pos}'}), \boldsymbol{\nu}_{i_{\text{D}}}, \varepsilon, \beta_1, \beta_2)$

21     Update code-generator:

22     $\widehat{\boldsymbol{c}}_{1:N} \leftarrow CG_{\boldsymbol{\omega}'}(\boldsymbol{\xi}'_{1:N})$;

23     $\boldsymbol{\omega}' \leftarrow \text{Adam}(\nabla_{\boldsymbol{\omega}'} \mathcal{L}(CG_{\boldsymbol{\omega}'} \mid \mathbf{s}^{\text{Pos}'}), \boldsymbol{\omega}', \varepsilon, \beta_1, \beta_2)$;

24     Update generator for Fairness:

25     **if** $t - n_{\text{fair}} > 0$ **then**

26       Repeat lines 6-8;

27       $\boldsymbol{\omega} \leftarrow \text{Adam}\Big(\nabla_{\boldsymbol{\omega}}(\mathcal{L}(G_{\boldsymbol{\omega}}, E_{\boldsymbol{\eta}} \mid \mathbf{s}^{\text{Pos}'}) + \lambda_F \mathcal{L}_{Fair}(G_{\boldsymbol{\omega}}, T_{\boldsymbol{\nu}} \mid \mathbf{s}^{\text{Pos}'})), \boldsymbol{\omega}, \varepsilon, \beta_1, \beta_2\Big)$;

28     Update generator for class balance:

29     **if** $t - n_{\text{balance}} > 0$ **then**

30       Repeat lines 6-8;

31       **for** $i_D = 1$ *to* $n_{\text{DB}}$ **do**

32         $\boldsymbol{\omega} \leftarrow \text{Adam}\Big(\nabla_{\boldsymbol{\omega}}(\mathcal{L}(G_{\boldsymbol{\omega}}, E_{\boldsymbol{\eta}} \mid \mathbf{s}^{\text{Pos}'}) +$

33                     $\lambda_{B_{i_D}} \mathcal{L}_{Balance}(G_{\boldsymbol{\omega}}, T_{\boldsymbol{\nu}_{i_{\text{D}}}} \mid \mathbf{s}^{\text{Pos}'})), \boldsymbol{\omega}, \varepsilon, \beta_1, \beta_2\Big)$;

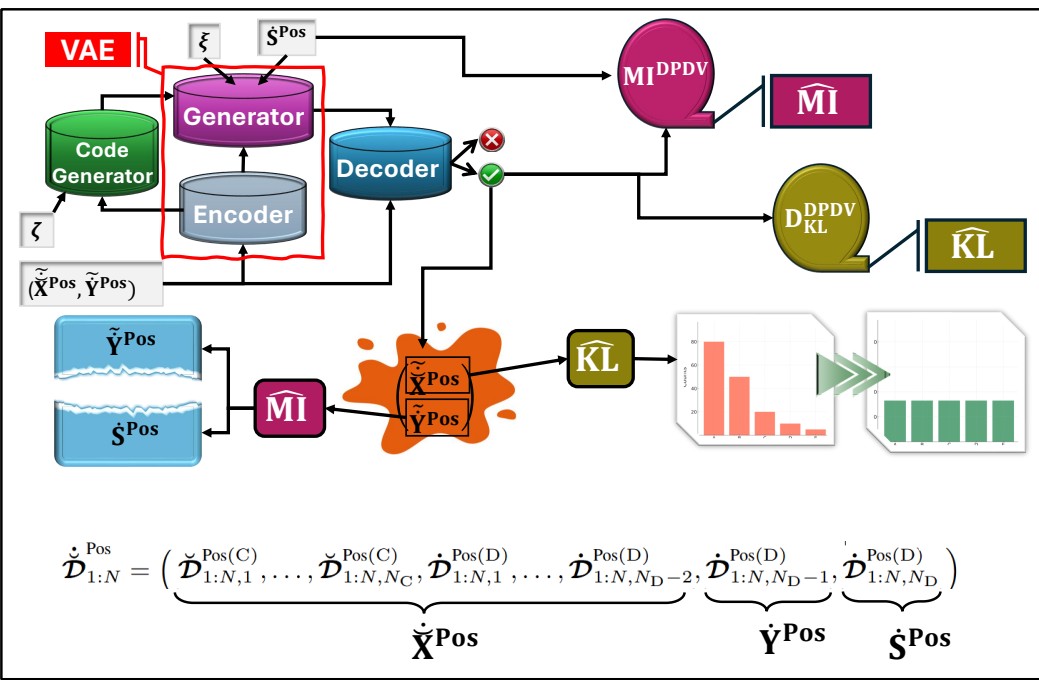

Figure 21: General diagram of the proposed Private–Fair–Balanced framework within BNPL, implemented using VAE$\mathcal{C}$GAN, a member of the family of generative models with a generator/decoder structure.

