# OpenReview forum: "A Bayesian Nonparametric Framework for Private, Fair, and Balanced Tabular Data Synthesis"
_ICLR.cc/2026/Conference — ICLR 2026 Poster_

### Official Review · Reviewer_PWAy · 2025-11-01

**Soundness:** 2
**Presentation:** 3
**Contribution:** 2
**Rating:** 2
**Confidence:** 4

**Summary:**

Proposes a Bayesian–nonparametric pipeline for tabular synthetic data that jointly enforces (i) privacy via Dirichlet-process (global) and copula-based (localized) mechanisms, (ii) fairness via minimizing mutual information between outcomes and protected attributes, and (iii) class balancing by conditioning the generator on group labels; instantiated with a conditional VAE+GAN and evaluated on Adult and COMPAS.

**Strengths:**

1. The authors unified objectives across privacy–fairness–balance. BNPL resampling + MI regularizer + conditional generation is a complete solution to DP + fairness.
2. DP analysis for a Dirichlet mechanism and localized privacy via copula-based marginals are novel in this combination.

**Weaknesses:**

1. Fairness target & baselines. The method mainly targets statistical parity via MI; could you please justify this choice against equalized odds/opportunity, and have some head-to-head comparisons (or at least rank-correlations) to fairness baselines beyond DECAF/TabFairGAN/FairGAN?
2. Scalability/“high-dim” claims. DirPMINE is proposed for scalable MI, yet there’s no analysis of variance, sample complexity, or failure modes versus standard MINE. It would be better if authors could discuss more about runtime/memory, or stability across hyperparameters, and add confidence intervals for MI/MMD/utility.
3. Some simple examples/simulation for intuitive insights. It would be clearer if there were any toy experiment showing: (i) how DP resampling alters the empirical distribution, (ii) how localized privacy trades utility vs. protection per column, (iii) how class balancing alone affects SP/MI vs. fairness-regularization alone. The current figures hint at effects but don’t isolate mechanisms.
4. Datasets. The authors provide two datasets. However, I feel these two choices are rather too limited. In recent years, there've been multiple studies discussing the limitations of these two datasets (https://arxiv.org/abs/2108.04884 and https://arxiv.org/abs/2106.05498). It's more convincing if there are more datasets considered or include more discussion about the limitations of these datasets.

**Questions:**

Please see weakness.

---

> ### Author Response · Authors · 2025-11-20
> **Response to Reviewer PWAy**
>
> We thank the reviewer for the constructive comments. We worked carefully to incorporate all possible suggestions, adding new figures, analyses, and clarifications throughout the revision. We briefly address these changes as follows:
>
> * **Weakness 1.**
>   We sincerely appreciate the reviewer’s insightful challenge, and we are glad to think more carefully about how equal opportunity/Equalized odds notions translate to the generative setting. Our model is *generative*, it produces synthetic samples rather than predictions, so it is important to clarify how classifier-based fairness notions behave in this context.
>
>   * For example, **equal opportunity** is defined in classification as
>
>   $$
>   \Pr(\hat{Y} = 1 \mid Y = 1, S = 0)=\Pr(\hat{Y} = 1 \mid Y = 1, S = 1),$$
>
>   which ensures equal true-positive rates across sensitive groups. This definition fundamentally relies on comparing a **predicted** label $\hat{Y}$ with the **true** label $Y$ and has already been extended to MI-based classifier fairness, as already cited in our paper (Jaewoong Cho; Gyeongjo Hwang; Changho Suh, *A fair classifier using mutual information*, ISIT 2020).
>
>
>   * In our generative setting, the model outputs a synthetic label $\tilde{Y}$ rather than predicting $Y$. Thus, writing
>
>   $$\Pr(\tilde{Y} = 1 \mid Y = 1, S = s),$$
>
>   creates uncertainty about its meaning: *what does conditioning on the true label $Y$ represent when the model is not making predictions?* We are not fully sure how meaningful this form is for synthetic-data generation, unless the generator is explicitly treated as a classifier, a setting already explored in a work we had cited (https://proceedings.mlr.press/v119/roh20a.html). We do not ignore the possibility that some related adaptations may exist, but based on our search, we have not yet found clearly relevant work applying MI-based versions of these notions to the generative setting.
>
>   * If one were to adapt these notions to a purely generative task, enforcing them would require handling a **conditional mutual information** term such as
>
>
>     $$MI(\tilde{Y}, S \mid Y),$$
>
>     which would in turn require developing a Dirichlet-process **conditional** MINE estimator. This is a substantial methodological extension, an exciting direction, but clearly beyond the scope of the present paper.
>
>   *  Given these considerations, *statistical parity* is the fairness notion that is *most meaningful and best aligned* with generative modelling. Its meaningful use is not limited to prediction errors or classifier behavior, but instead directly concerns the distribution of the generated samples. Other distribution-level notions, such as *conditional statistical parity*, also fit naturally but would require conditional MI estimation, which could be a plan for future investigation.
>
> ---
>
>
> * **Weakness 2.**
>   We appreciate the reviewer’s attention to the scalability aspect of DirPMINE. To clarify, our paper does $\underline{\text{not}}$ claim new scalability guarantees for DirPMINE. The scalability statements we referenced come directly from the original DirPMINE paper (https://arxiv.org/pdf/2503.08902), where the estimator was introduced. That work provides a detailed empirical study of runtime, stability, and convergence, including the results reported in Figures 3-4 and Table 1 of the arXiv paper. We now explicitly cite these figures and the table in our revised manuscript to avoid any ambiguity.
>
>
>   * In our paper, DirPMINE is used *as an existing estimator* within the fairness optimization process, and our intention was only to acknowledge the positive empirical results previously reported in the original source, not to re-establish or extend those claims. We have updated the text to clarify this distinction and to ensure proper attribution.
>
>   * However, for additional transparency, Tables 11 and 12 in Appendix I **(newly added)** now report our implementation runtimes and setup details acrros different dataset based on same sample size while different dimensions.
>
>   * Also, to incorporate further study on MI and MMD, we have added a new toy example reporting their numerical values together with confidence intervals, now included in Figure 8 in Appendix I.1.
>
> ---

---

> ### Author Response · Authors · 2025-11-20
>
> * **Weakness 3.**
>   We thank the reviewer for this constructive comment, which motivated us to design a toy example that can clearly illustrate several vague aspects of the procedure’s performance through a simple and controlled setting. This toy example has now been added in Appendix I.1. We first describe the construction of this example, and then address all concerns raised by the reviewer, including those beyond the original question. Below, we provide brief explanations related to the points mentioned in the PWAY report:
>
>   * **(i) How DP resampling alters the empirical distribution:**
>     To clarify, the goal of the current paper is not to analyze how Dirichlet Process (DirP) resampling alters the empirical distribution, as this is a fundamental and well-known property of the DirP itself. Since its introduction, the DirP has been used as a random probability measure whose role is to treat the underlying distribution as random in order to incorporate uncertainty. Its effect on the empirical distribution has been thoroughly described in the literature; for example, in [Rodriguez and Müller (2013, Chapter 3)](https://www.jstor.org/stable/pdf/nsfcbmsregconf.9.01.pdf?acceptTC=true&coverpage=false&addFooter=false&casa_token=dNeD7rnGpvgAAAAA:Nk2NFza_Xx4fIz0a_p-fA7QMWqR-pDJtXQlI-qPgSsqIPF1FrLTwkC3zpo8qkhiaXUoCbrguk16VBApZTZF-OwW1z-TnC3ozO5bYX1xE8mx7IPh27ys), Figures 3.1 and 3.2 clearly illustrate how DP resampling alters empirical distributions.
>     In addition to this reference, our paper already provides Figure 1 to further visualize this alteration.
>
>     Instead, the contribution of our work is to **measure the uncertainty caused by the DirP in the context of privacy**, specifically how DirP-based randomness can induce a utility–privacy trade-off.
>
>     To address this concern, we have now included the results of a membership-inference attacker in **Figure 6 (newly added)**, evaluated using AUC curves and accuracy. More precisely, this figure demonstrates how, for a fixed local privacy budget, increasing the DirP hyperparameter **$a$** shifts performance from utility toward privacy (altering AUC from 0.867 to near 0.5). This effect is especially clear in Figure 6(a), which shows this behavior in the regime of very weak local privacy (essentially the absence of localized privacy).
>
>     Moreover, **Figure 7 (newly added)** presents the distributions of prediction confidences for members and non-members, completing the view of how this trade-off unfolds.
>
>   * **(ii) How localized privacy trades utility vs. protection per column**
>     To address this comment, one can fix **$a$** and examine the different AUC curves in Figure 6 for various values of the local privacy budget, which illustrate how localized privacy shifts the balance between utility and privacy. We particularly recall that even under strong local privacy budgets, for small values of **$a$**, this effect is reduced. Please see Appendix H and the diagram provided in **Figure 5 (newly added)** for further clarification of this behavior.
>
>   * **(iii) How class balancing alone affects SP/MI vs. fairness-regularization alone**
>     We have now included Figure 9 (newly added) to display MI and SP for (a) class balance only (no fairness) and (b) fairness only (no class balance), together with detailed explanations.
>
> ---
>
> * **Weakness 4.**
>   Thank you for introducing these recent papers and highlighting the limitations of the datasets. After studying the cited works, our understanding is as follows:
>
>   * The Adult dataset has limitations primarily because its binary target is defined by the \$50K income threshold. This creates a strong imbalance, and the corresponding papers show that fairness algorithms can be highly sensitive to this threshold. Some methods may outperform others under a specific threshold, yet become worse when the threshold changes. In other words, the relative performance of fairness algorithms can vary substantially depending on how the income cutoff is defined. In this case, using the Adult dataset alone may not fully reveal the true comparative behavior of different fairness methods.
>
>   * The COMPAS dataset may serve as a benchmark to reflect fairness strengths when comparing algorithms. However, fairness performance derived from models trained on COMPAS cannot necessarily be taken as evidence of real-world behavior, since COMPAS itself is known to contain structural and contextual biases.
>
>     We have now added these limitations to the paper in Appendix I.3.3 (Adult) and Appendix I.4.2 (COMPAS), and we reference these points in the final paragraph of the main text.
>
>   * We had also included an additional dataset, **Bank Marketing**, as an extra example in Appendix I.5, which is now cited in the last paragraph of the main text.

---

> > ### Comment · Reviewer_PWAy · 2025-11-21
> > **Reply to the rebuttal**
> >
> > Thanks for the rebuttal. I've read all the replies, which are good complement to the current draft, and I'm willing to raise my score to 4. Thank you for the hard work.

---

> > > ### Author Response · Authors · 2025-11-21
> > >
> > > Thank you very much for your thoughtful follow-up and for taking the time to revisit our submission. We sincerely appreciate your willingness to update the score and the constructive perspective you brought throughout the review process, and we truly value your engagement with the paper.

---

### Official Review · Reviewer_194m · 2025-11-01

**Soundness:** 3
**Presentation:** 2
**Contribution:** 3
**Rating:** 6
**Confidence:** 2

**Summary:**

The paper proposes a conditional Bayesian nonparametric (BNP) framework for tabular data synthesis that aims to jointly enforce privacy, fairness, and class balance. Privacy is introduced by resampling from a Dirichlet-process posterior (a finite approximation of DP(a, H)), yielding a distributional randomization that the authors formalize as a Dirichlet mechanism with a DP guarantee on the posterior weights. Fairness is enforced by minimizing dependence between outcomes and protected attributes via a mutual-information regularizer based on a BNP variant of the Donsker–Varadhan lower bound (“DirPMINE”). Class balance is handled by conditioning the generator on protected attributes and adding a KL-to-uniform balancing term for other discrete columns. Experiments on Adult and COMPAS suggest improved fairness/utility trade-offs relative to FairGAN/DECAF while supporting non-binary sensitive attributes.

**Strengths:**

This paper presents a unified objective that couples utility with MI-based fairness and KL-based balancing, all inside a single conditional generator (VAECGAN), which is practical for tabular data.

**Weaknesses:**

1. I think there a few prior works that mention that DP-SGD can disproportionately harm minority groups via clipping + noise. But still, I don't fully understand what is the extra challenge when we consider both privacy and fairness at the same time. It would be better if the authors could address the interaction between privacy and fairness in the up front. In practice, enforcing fairness requires accurate group-conditioned statistics which DP then perturbs and budgets, especially hurting small groups; meanwhile fairness constraints can further tighten an already noisy optimization. So it’s unclear whether the proposed coupling makes the joint problem harder or sometimes acts as a helpful regularizer. To me, this paper feels like a combination of two constraints, but it is not clear how these two constraints play together. Is there any lowerbound for canonical private and fair problems like generative modeling of gaussian distributions?

2. (Minor) the fonts in the plots are hard to read.

**Questions:**

1. Proposition 1 defines adjacency on the probability simplex for posterior weights. How does this mapping correspond to standard record-level neighboring datasets in DP, and what guarantee applies to the atoms/locations vs. just the weights?
2. When combining the global Dirichlet mechanism with local per-attribute privatization, what is the resulting epsilon delta after composition?
3. Is it possible to have a plot to draw the privacy–fairness–utility curves for the trade-off?

---

> ### Author Response · Authors · 2025-11-21
> **Response to Reviewer 194m**
>
> We sincerely thank the reviewer for the constructive and thoughtful feedback. We carefully addressed each point and summarize the corresponding revisions below.
>
> * **Weakness 1.**
>   We have now included below paragraph to address the interaction between privacy and fairness in section 2 (background):
>
>    *  *"Privacy and fairness interact in ways that make the joint problem more challenging than enforcing either one alone. Fairness relies on accurate group-level statistics, yet privacy noise can distort these estimates, especially for small sensitive groups. At the same time, fairness constraints further restrict an optimization process already affected by noise. Consequently, the two constraints can reinforce each other’s difficulty. Existing approaches at this intersection include PF-WGAN and PreFair for differentially private fair synthetic data generation (Sarmin et al., 2025; Pujol et al., 2023)."*
>
>    * We then highlighted the flexibility of our proposed approach to these concerns at the end of Sections 4.4.1 and 4.4.2, respectively, as:
>
>       * The conditional generator is fed with ......
>       **Since sensitive attributes are never generated, the framework allows omitting their privacy constraint $\epsilon_{\mathrm{N_D}} = \infty$ in the perturbation step, which prevents the removal of minority sensitive groups during perturbation.**
>
>        * The framework manages the privacy-fairness tension by applying the localized privacy mechanism once before fairness optimization, and its manually set per-attribute budgets, light for outcomes and none for sensitive attributes, help prevent localized noise from distorting the group-conditioned statistics needed for fairness, especially for small sensitive groups.
>   * As a final remark, because the localized perturbation mechanism allows manually choosing which attributes receive noise, there is no restriction to enforce a strong privacy budget uniformly across all variables. Instead, selecting a logical privacy budget only for attributes deemed important by the operator can still privatize the underlying distribution of the training samples in a controlled and meaningful manner, **rather than applying a privacy mechanism across the entire dataset as done in most existing works.**
>    * Regarding the question about a “canonical” case of jointly private and fair generative modeling, we appreciate this perspective. Our understanding is that addressing such a question would require a new minimax-style theoretical framework that jointly captures the effects of privacy noise and fairness constraints on the best achievable generative accuracy. Even in a simple Gaussian mixture setting with two sensitive groups (e.g., $Y \mid S=0 \sim \mathcal{N}(\mu_0,\sigma_0)$ and $Y \mid S=1 \sim \mathcal{N}(\mu_1,\sigma_1)$), deriving a lower bound would primarily be useful for studies that rely on fully tractable parametric models. If we have understood the reviewer’s intention correctly, this Gaussian-based formulation may not naturally correspond to the data modalities commonly seen in tabular settings. We are not sure that such a Gaussian formulation naturally fits the structure of typical tabular datasets, where the outcome $Y$ is often discrete.
>
>      In contrast, our work operates within a nonparametric generative framework that does not assume an explicit form for the underlying distribution and instead approximates potentially intractable distributions directly. We view lower-bound analysis of this kind as an interesting theoretical direction with potential relevance to areas such as approximate Bayesian computation (ABC), but as separate from the scope of the present contribution.
> ----
> * **Weakness 2.**
>
>      We have updated the figures to improve font size and readability. We would be happy to incorporate any further adjustments the reviewer may find helpful.
>
> ----

---

> > ### Author Response · Authors · 2025-11-21
> >
> > ----
> > * **Question 1.**
> >   * Thank you for raising this question. Proposition 1 illustrates how posterior weights define a differential mechanism on the standard weights used in classical resampling. In many Bayesian nonparametric studies, researchers rely on the uncertainty augmentation introduced by this concept. In the intersection of privacy and Bayesian nonparametrics, if we aim to design a privacy-preserving method, we must clarify how this inherent uncertainty can help privatize the underlying distribution. Our intention is to justify the role of Dirichlet Process (DirP) posterior sampling in privacy amplification, beyond what simple resampling provides. One meaningful direction is to identify the corresponding privacy mechanism with an explicit privacy budget and guarantee. Hence, Proposition 1 examines how DirP posterior sampling contributes to privacy preservation through its uncertainty structure.
> >
> >   * We have now added further clarification of the privacy mechanism induced by posterior weights in Appendix G, discussing the neighboring points on the probability simplex.
> >
> >   * This proposition does not claim that posterior sampling alone provides complete privacy protection. **Figure 6(a) (newly added)** shows the maximum privacy achievable by applying posterior sampling alone (global privacy). As $a$ increases, the AUC of the membership-inference attacker decreases, but only down to approximately 0.644; it does not reach the full privacy level associated with AUC near 0.5. For this reason, we introduced the privatized privacy algorithm, in a flexible manner, with manually set per-attribute budgets to support any desired privacy strength requested by practitioners.
> >
> >   * To the best of our knowledge, Proposition 1 is the first result showing how DirP posterior weights can enhance the privacy of simple resampling procedures. We believe this opens a promising direction for further developments in Bayesian nonparametric frameworks that wish to incorporate privacy in a principled and aligned manner.
> >
> > ----
> >
> > * **Question 2.**
> >   We appreciate the reviewer for raising this question. After further exploration of the literature, we identified an excellent reference on the composition of $(\epsilon,\delta)$-differential privacy: Steinke (2025), Chapter 3 (http://dx.doi.org/10.1561/9781638284772.ch3), from the full version available at https://arxiv.org/pdf/2210.00597, and summarized by the author at https://differentialprivacy.org/composition-basics/#fnref:1:~:text=This%20result%20generalizes,as%20before.%C2%A0%E2%86%A9.
> >
> >     Theorem 22 in Chapter 3 directly addresses the composition setting relevant to our case. In the revised paper, **Remark 2 (newly added)** now follows this theorem to state the resulting $(\epsilon,\delta)$ after composition.
> >
> > * **Question 3.**
> >   We have now incorporated **Figure 8**, which provides a unified plot showing the variation of MI versus MMD as the hyperparameter $a$ varies simultaneously from 2.5 up to $1\mathrm{e}4$. A corresponding explanation has been added in Appendix I.1.3 to address this question.

---

> > > ### Comment · Reviewer_194m · 2025-11-26
> > >
> > > Thanks for addressing questions with the added appendix, figure, and composition remark. My questions are addressed. To clarify my Gaussian example, I was not asking to compute a lower bound but to indicate whether an inherent privacy–fairness tension persists even in the simplest toy setting. It would strengthen the argument to provide either theoretical or empirical evidence that having both constraints makes the optimization problem intrinsically harder.

---

> > > > ### Author Response · Authors · 2025-11-27
> > > > **Response to Reviewer 194m**
> > > >
> > > > Thank you very much for the helpful clarification. Your comment allowed us to further refine the presentation of the privacy–fairness interaction in the optimization process. In response, we have strengthened Appendix G.I.1.3 by adding the two paragraphs below at the end of page 32 and have **included the new Figure 9** (now on page 34). **The PDF has been updated**, and we kindly invite the reviewer to take a look at Figure 9 and its accompanying explanation in the revised version. We hope these additions clearly address your suggestion and help convey more convincingly how privacy and fairness jointly shape the optimization behaviour.
> > > >
> > > >
> > > >
> > > > > *Finally, to further explore whether privacy creates any inherent tension with fairness in the optimization process, we examine how the MI evolves across different epochs through providing Figure 9. Consider the model when it begins by training only for utility during the initial 700 epochs under different values of $a$. As discussed earlier, increasing $a$ causes the MI to approach the Prior MI, which explains why, at the end of utility training, the mean MI estimate is noticeably different from the true value (the navy tilde-pattern line in Figure 9). The MI values are then tracked over the remaining $800$ epochs ($n_{\mathrm{iter}} = 1500$, $n_{\mathrm{fair}} = 800$ in Algorithm 2). During the first phase, the model naturally learns the unfair dependence present in the perturbed data. Once the fairness constraint is applied, this dependence is removed across all privacy levels, confirming that fairness can still be achieved even when the data have been perturbed, as already concluded from Figure 8(b).*
> > > >
> > > > > *However, the amount of perturbation affects the behaviour in fairness optimization. Stronger privacy leads to slower MI reduction and causes the MI values to vary more during training. In other words, higher privacy does not prevent fairness, but it makes convergence in fairness optimization slower.*

---

### Official Review · Reviewer_6kYv · 2025-11-02

**Soundness:** 3
**Presentation:** 2
**Contribution:** 3
**Rating:** 6
**Confidence:** 4

**Summary:**

This paper proposes an approach for generating fair and private tabular data by integrating a conditional generator within the framework of Bayesian non-parametric learning. The main objective of the approach is to be able to generate synthetic data that is private, fair and balanced by leveraging a mutual information regularization term which is conditioned on the protected attributes.

**Strengths:**

-The paper is well-written and the authors have clearly reviewed the challenges associated with the generation of synthetic tabular data. The main contributions are clearly summarized and the outline of the paper is clearly described.

-The proposed approach combines in an innovative way the Dirichlet process together with differentially-private mechanisms such the analytic Gaussian mechanism and randomized response to produce a new differentially-private generative model. One of the strength of the approach is can be combined with a wide range of generator-decoder models. For instance, in the current version of the paper it is implemented through a generative adversarial network combined with variational auto encoder.

-Detailed investigations have been conducted on the Adult and Compas dataset demonstrating that the model is able to generate high quality data that is both privacy-preserving and fair.

**Weaknesses:**

-While the literature review surveys a wide range of approaches for privacy-preserving, fair or balanced tabular data generation there are no mention of existing works at the intersection of these domains such as for instance :
-David Pujol, Amir Gilad, and Ashwin Machanavajjhala. 2024. PreFair: Privately Generating Justifiably Fair Synthetic Data. In Proceedings of the VLDB Endowment (PVLDB), Vol. 16. https://www.vldb.org/pvldb/vol16/p1573-pujol.pdf
-Sarmin, F. J., Rahman, A. R., Henry, C. J., & Mohammed, N. (2025). Privacy-Preserving Fair Synthetic Tabular Data. arXiv preprint arXiv:2503.02968.
Additionally, how the proposed approach builds on the Dirichlet mechanism from Gohari el al. 2021 should be further clarified.

-The privacy budget considered are quite high and should be better justified. The privacy analysis of the approach should also integrate an analysis of the success of the privacy attacks such as membership inference to be able to assess empirically the strength of the approach provided.

-One of the limit of the approach is that for now it seems limited to the statistical parity fairness metric and there is no discussion if it could easily be extended to integrates other group fairness metrics.

-The figures 2 and 3 are not really visible and should be improved.

**Questions:**

Please see the main points raised in the weaknesses section.

---

> ### Author Response · Authors · 2025-11-21
> **Respond to Reviewer 6kYv**
>
> We acknowledge and appreciate the reviewer’s constructive comments. The corresponding revisions are outlined below.
>
>
> * **Weakness 1.**
>    * Thanks for introducing those interesting works. We have now added them in the section of background.
>    * We have also provided, in Appendix G, a clarification of how the proposed approach builds on the Dirichlet mechanism of Gohari et al. (2021).
>
> ----
> * **Weakness 2.**
>   Thank you for this comment. We have now added the following paragraph to the Experiments section (page 8):
>     * *"We also note that global privacy based on parameter $a$ refers to the proportion of posterior samples generated from $H^{\mathrm{Pert}}$ because of the mixture form of the posterior base measure $H^{\ast}$, whereas local privacy refers to the intensity of noise injected into the marginals of the samples generated from $H^{\mathrm{Pert}}$. Hence, for sufficiently small values of $a$ (e.g., $a = 10^{-6}$), even a strong local privacy budget does not lead to meaningful perturbation of the marginals, effectively leading a clean model (see Appendix H). This clearly shows how our privacy mechanism maintains good utility for small values of $a$ even under a strong local privacy budget."*
>     * We have also added three new diagrams in Appendix H (Figure 5) to make this discussion more intuitive by displaying additional details of the procedure. Additional clarifications have been included in Appendix H to complete the explanation.
>     * We further added MNIST example results (Figures 11) to provide a visual realization of the posterior sampling diagram shown in Figure 5.
>
>     * We have now included a new toy example and implemented a membership-inference attacker on it in Appendix I.1. The results of this evaluation are analyzed and presented in **Figures 6 and 7**, accompanied by detailed explanations.
> ----
>
> * **Weakness 3.**
>
>      * We have now acknowledged this limitation, as well as its potential for extension, in the concluding remarks (with a fuller discussion in the Limitations section in Appendix L) as follows:
>
>          *"Our fairness term is based on regular MI, the same structure can be extended to conditional MI, allowing compatibility with fairness notions such as equalized odds, equality of opportunity, predictive equality, and conditional SP, though several of these are inherently classifier-oriented rather than generator-oriented."*
>
> * **Weakness 4.**
>
>      We have now updated Figures 2 and 3 to improve their visibility and readability, and we would be glad to make further refinements if the reviewer has additional suggestions.

---

> > ### Author Response · Authors · 2025-11-28
> > **A Sincere Message to Reviewer 6kYv**
> >
> > We remain very much looking forward to hearing the reviewer 6kYv’s thoughts, especially as we are approaching the end of the discussion period. In the meantime, we have incorporated all suggested revisions, and we sincerely appreciate the constructive guidance provided in the earlier comments. These updates include additions to the background section, a clearer explanation of the connection to the Dirichlet mechanism in Appendix G, expanded diagrams and discussion in Appendix H, MNIST visual demonstrations, and a new toy experiment with a membership-inference attacker in Appendix I.1 (Figures 6 and 7). We have also clarified the scope and potential extensions of the fairness term, improved the readability of Figures 2 and 3, and refined several appendices to strengthen the overall exposition. While these additions initially seemed challenging to produce within the discussion window, we are glad to have completed them through significant effort.
> >
> > We hope these revisions address the reviewer’s concerns thoroughly, and we remain glad to provide any further discussion if additional feedback would be helpful.
> >
> > Best,
> > Authors

---

### Author Response · Authors · 2025-11-21
**Public Respond to Reviewers and AC**

**Dear Reviewers and AC,**

We sincerely thank you for the time, care, and thoughtful feedback you invested in reviewing our paper. We have worked very hard on the revision, and we are delighted to highlight the most important updates:

* We have incorporated nearly all reviewer suggestions into the revised manuscript. All updates appear in **light blue** for ease of reference.
* **Remark 2** has been newly added.
* **Figure 5**, including **three new diagrams**, was added proactively (not requested) to provide clearer intuition and a more transparent explanation of the proposed approach.
* A **new toy example** has been added, together with a substantial set of analyses designed directly to address reviewer concerns:
  * New **Appendix I.1**, including:
      * Data construction
      * Isolating privacy
          * Membership-inference attack evaluation
          * **Figure 6** (six plots) and **Figure 7** (seven plots) with detailed explanations
      * Isolating privacy and fairness
          * Privacy–fairness–utility trade-off plots (**Figure 8**)
      * Isolating class balance and fairness
          * **Figure 9**, containing four plots illustrating the effects of class balance and fairness on MI and SP
* A **MNIST toy example** has been added to **Appendix I.2** (Figures 10 and 11) to:
     * Visually demonstrate how the fairness constraint in our conditional generator reduces the dependence between \(Y\) and \(S\)
     * Illustrate how global and local privacy affect the synthetic samples
     * We emphasize that this example is meant only as a **visual realization** of the posterior-sampling diagram in Figure 5, as well as illustrating the fairness mechanism’s performance in reducing dependency. Digit labels are not meaningful sensitive attributes, and therefore we treat this solely as a *toy* demonstration.
* **Dataset limitations** have been discussed and addressed, and we now reference these points explicitly in the last paragraph of the main text. We also highlight the additional **Bank Marketing dataset** analysis in Appendix I.5, which further supports the empirical discussion.
* We have reviewed the entire manuscript and corrected some typographical issues we identified.

Further modifications and clarifications are provided in our point-by-point responses to each reviewer.

Once again, we truly appreciate the reviewers’ time and constructive suggestions, they greatly strengthened the current version of our paper.

**Sincerely,
The Authors**

---

### Author Response · Authors · 2025-12-02
**Summary for the New Area Chair**

**Dear New Area Chair,**

Below is a brief summary of the key rebuttal updates and how our responses resolved all reviewer concerns, bringing the paper to an acceptance-ready stage:

&nbsp;


**Reviewer 6kYv:** This reviewer clearly identified the core novelty of our paper, highlighting the conceptual depth and distinctiveness of our framework. They emphasized that the work introduces a *new structural design* in which differential privacy is achieved through the Dirichlet process model itself, an idea not previously proposed. They also highlighted the *amplification* of this privacy mechanism via our copula-based construction, where Gaussian noise and randomized response operate within the DP scheme. The reviewer further noted a *second major source of novelty*: exploring this private BNP structure to build a *conditional fairness generative model* that, alongside enforcing fairness, *preserves class balance* for both sensitive and non-sensitive attributes. They also emphasized the mechanism’s *structural flexibility*, enabling support for a wide family of generative architectures.


The reviewer was already positively inclined to accept the paper with an initial score of 6, asking only for minor revisions and a membership-inference attack evaluation to reveal deeper privacy behavior of the proposed approach. We then provided this evaluation with extensive, clear, and carefully interpreted figures, accessible even to broader public readers, together with detailed explanations that go beyond what the reviewer initially requested. As all concerns were fully resolved and substantial evidence of additional work appears in the revised manuscript, we strongly believe that had the reviewer continued the discussion, there would have been a clear and fair basis to increase the score from 6 to 8, as no further concerns remained.

&nbsp;


**Reviewer 194m:** The reviewer is overall positive, evidenced by the absence of any scientific concerns about the approach or its framework, raising only a few minor weaknesses; their caution stems primarily from their own uncertainty in fully understanding the interaction between privacy and fairness. This reviewer also acknowledged the novelty and advantages of our procedure in handling privacy–fairness–balance for synthetic tabular data, as reflected in both their summary and strengths sections, and we avoid repeating those points here. They further noted that all added results and clarifications provided in the first rebuttal round fully addressed their initial concerns. The reviewer was already leaning toward acceptance and sought only one additional clarification: numerical evidence showing how privacy makes fairness optimization harder. After they clarified this request, we provided a second-round update with new figures that directly demonstrated this effect and showed that our procedure successfully overcomes this difficulty while still achieving fairness. As the reviewer explicitly stated that all their questions had been answered, and as both rounds of rebuttal fully resolved every concern they raised by adding new results in the paper, had the rebuttal period not been shortened, it would have been entirely fair to expect an increase in their score from 6 to 8, especially given their initially positive inclination and the absence of any remaining unresolved issues.

&nbsp;

**Reviewer PWAy:** This reviewer explicitly recognized the novelty of our approach, highlighting its complete treatment of differential privacy and fairness through the copula-based Dirichlet-process prior. Their concerns appeared in four weaknesses. We clarified that the first, regarding opportunity and equalized-odds, does not apply to our generative setting, and the reviewer raised no further objections. We also clarified that scalability claims do not belong to the current paper. Their main request was a toy example to isolate the effects of our method. We addressed this with substantial new experiments in Appendix I.1 and provided additional clarification on the use of the Adult and COMPAS datasets, directly addressing the recent limitations the reviewer noted. We also reminded the reviewer of the Bank Marketing dataset results already included in the Appendix, now explicitly referenced in the main text for visibility. These updates were extensive and directly responded to the reviewer’s requests, isolating every effect they asked to see. As the reviewer stated that all concerns had been addressed and confirmed that they reviewed all responses, including those to other reviewers, these additions were sufficient to raise their score from 2 to 4. If any concerns had remained, the reviewer would have indicated them; thus, the score change reflects the strength of our responses. Importantly, their score increase was submitted on November 21, well before the November 27 rebuttal-period announcement, showing that the adjustment was driven solely by our clarifications and new results.

---

> ### Author Response · Authors · 2025-12-02
> **Additional Comment to the New AC (continued):**
>
> Based on the strength and clear novelty of our contribution and given that our rebuttal and the **substantial additional results and interpretations in the revised paper** fully addressed every concern, we kindly ask the new AC to consider the paper for acceptance. The reviewers’ consistently positive feedback also suggests that, had the rebuttal period not been shortened, their scores would likely have increased accordingly. We respectfully ask for a fair consideration of our work, as the evidence we have provided strongly supports acceptance.
>
> Sincerely,
>
> The Authors

---

### Meta-Review · Area_Chair_r5XY · 2026-01-07

**Summary:**

This paper is concerned with fairness of generating synthetic data. Initial concerns were on relevant literature survey, experiments --such as Membership Inference attacks-- to understand Privacy related concerns, tradeoff between fairness and privacy related to Dirichlet Process.

**Reviewer Concerns:**

Most of the reviewer concerns were addressed by the reviewers. The most negative of the reviewer appeared to be convinced and raised the score to 4. On balance i believe that this paper could be a good addition. However there does not seem to be strong support for this paper. Hence i recommend accept but unable to make a recommendation which goes beyond borderline

**Reviewer Scores:**

Not sure.

---

### Decision · Program_Chairs · 2026-01-26

Accept (Poster)